# No Caption, No Problem: Caption-Free Membership Inference via Model-Fitted Embeddings

**Joonsung Jeon, Woo Jae Kim, Suhyeon Ha, Sooel Son**[*] **& Sung-Eui Yoon**[*]
Korea Advanced Institute of Science and Technology (KAIST)
{mikeraph,wkim97,suhyeon.ha,sl.son,sungeui}@kaist.ac.kr

## Abstract

Latent diffusion models have achieved remarkable success in high-fidelity text-to-image generation, but their tendency to memorize training data raises critical privacy and intellectual property concerns. Membership inference attacks (MIAs) provide a principled way to audit such memorization by determining whether a given sample was included in training. However, existing approaches assume access to ground-truth captions. This assumption fails in realistic scenarios where only images are available and their textual annotations remain undisclosed, rendering prior methods ineffective when substituted with vision-language model (VLM) captions. In this work, we propose **MoFit**, a caption-free MIA framework that constructs synthetic conditioning inputs that are explicitly overfitted to the target model's generative manifold. Given a query image, MoFit proceeds in two stages: (i) **Mo**del-**Fit**ted surrogate optimization, where a perturbation applied to the image is optimized to construct a surrogate in regions of the model's unconditional prior learned from member samples, and (ii) surrogate-driven embedding extraction, where a model-fitted embedding is derived from the surrogate and then used as a mismatched condition for the query image. This embedding amplifies conditional loss responses for member samples while leaving hold-outs relatively less affected, thereby enhancing separability in the absence of ground-truth captions. Our comprehensive experiments across multiple datasets and diffusion models demonstrate that MoFit consistently outperforms prior VLM-conditioned baselines and achieves performance competitive with caption-dependent methods. The code is available at https://github.com/JoonsungJeon/MoFit.

## 1 Introduction

Latent diffusion models (LDMs) (Rombach et al., 2022) have advanced image-generation capabilities of generative models and broadened their applications to various tasks, such as photorealistic facial synthesis (Ergasti et al., 2024), medical CT image generation (Molino et al., 2025), and protein structure generation (Fu et al., 2024). However, there exist growing concerns and evidence that diffusion models can memorize and reproduce high-fidelity training images, posing serious threats to training data privacy (Somepalli et al., 2023; Webster, 2023; Carlini et al., 2023).

Membership inference attacks (MIA) have emerged as a standard empirical approach to assess the risk of training-data exposure in machine learning models (Shokri et al., 2017). MIAs are designed to decide whether a given query sample is used in training the target model, providing concrete metrics for auditing memorization and detecting privacy leakage. Recent studies have adapted MIAs to LDMs, exploiting signals such as differences in conditional training loss, reconstruction error, or denoising consistency to distinguish member samples from non-members (Carlini et al., 2023; Matsumoto et al., 2023; Duan et al., 2023; Fu et al., 2023; Zhai et al., 2024).

Existing MIA studies on text-to-image LDMs assume access to image–caption pairs; the ground-truth caption for a query image is available for inferring its membership. We contend that this assumption is often impractical for auditors. For example, an artist who suspects a generated image

---
[*]Co-corresponding authors

replicates their work typically lacks access to the training captions used by a released target model. Moreover, training-set provenance is frequently undisclosed on public generative-AI platforms.[1]

In this paper, we demonstrate that performing effective MIAs in the caption-free setting is challenging: replacing ground-truth captions with VLM-generated approximations substantially degrades the performance of state-of-the-art MIA approaches based on CLiD (Zhai et al., 2024) (Sec. 3.3).

To address this challenge, we present our novel finding on a systematic difference in how member and non-member (*i.e.*, hold-out) samples respond to mismatched conditioning in their denoising process. Member samples whose captions were used during training exhibit high sensitivity in their conditional denoising loss under alternative or misaligned conditions; hold-out images are relatively less affected (Sec. 3.3). This difference in sensitivity provides an important signal to establish and boost separability between member and non-member groups in the caption-free setting.

Motivated by this observation, we introduce MOFIT, a framework that constructs **Mo**del-**Fit**ted embeddings tailored to the generative manifold of the target model. Given a query image, MOFIT (1) synthesizes a surrogate input that aligns closely with the model's unconditional prior, and (2) extracts an embedding from this surrogate, forming a tightly coupled pair within the model's conditioning space. At inference time, conditioning the original query with this embedding leads to a pronounced increase in conditional loss for member samples, while hold-out samples exhibit relatively minimal changes – thereby enhancing separability in the absence of ground-truth captions.

We evaluate the effectiveness of MOFIT as an alternative to VLM-generated captions in the caption-free setting. Across three fine-tuned text-to-image diffusion models – Pokemon, MS-COCO, and Flickr – MOFIT consistently outperforms prior methods that rely on VLM-generated captions, achieving up to +25% ASR and +30–47% TPR@1%FPR improvements. Notably, on MS-COCO, MOFIT even surpasses prior methods with access to ground-truth captions, highlighting its strong discriminative power without textual supervision.

In summary, our contributions are as follows:

- We introduce the first MIA framework tailored for performing effective membership inference against LDMs in the caption-free setting, reflecting a practical adversary who lacks access to ground-truth captions.
- We present a novel empirical insight: during the denoising process, member samples exhibit larger changes in conditional loss under alternative conditioning than hold-out samples, providing an exploitable feature for separating members from non-members.
- Building on this observation, we propose a two-stage MIA: (1) synthesize caption embeddings explicitly optimized to overfit the target LDM, and (2) exploit those embeddings to condition the original query, thereby exploiting members' selective sensitivity and boosting loss-based separation.
- MOFIT outperforms prior methods conditioned on VLM-generated captions and achieves competitive performance even against state-of-the-art MIAs using ground-truth captions.

## 2 RELATED STUDIES: MEMBERSHIP INFERENCE

A membership inference attack (MIA) is a privacy attack in which an adversary seeks to determine whether a specific data sample is included in the training dataset of a target model. While early studies targeted deep neural networks (Shokri et al., 2017), recent research has extended MIAs to generative models – particularly diffusion models – due to their strong generation fidelity and potential for memorization (Carlini et al., 2023).

Carlini et al. (2023) first examined MIA in unconditional diffusion settings by leveraging multiple shadow models to statistically infer membership. Matsumoto et al. (2023) extended this by directly exploiting the training loss values at specific timesteps. SecMI (Duan et al., 2023) improved attack performance by estimating posterior errors across diffusion trajectories, while PIA (Kong et al., 2023) reduced the query cost by approximating ground-truth trajectories from a single intermediate latent. PFAMI (Fu et al., 2023) introduced a probabilistic fluctuation-based metric to capture differences in generation behavior between member and non-member samples. Most recently, CLiD (Zhai

---

[1]https://civitai.com/

et al., 2024) achieved state-of-the-art performance on multiple datasets by targeting membership inference in text-conditioned LDMs through discrepancies between conditional and unconditional denoising losses.

We emphasize that all prior MIA research on diffusion models has a common threat model in which the adversary already has access to the exact text captions corresponding to query images. However, such ground-truth captions are often inaccessible in practice. For instance, when testing the membership status of facial images of specific individuals, it is unrealistic to assume that the adversary already has the corresponding ground-truth captions to test their membership. In this paper, we adopt a more realistic threat model: a caption-free setting in which only the query image is available.

## 3 PRELIMINARY

### 3.1 LATENT DIFFUSION MODELS

The goal of latent diffusion models is to learn a parameterized reverse process that approximates a given data distribution. In the forward process, Gaussian noise $\epsilon \sim \mathcal{N}(0, \mathbb{I})$ is incrementally added to the latent $z_0$ across timesteps $t = 1, \ldots, T$, yielding a sequence of progressively noisier latents. Each noisy latent is computed as $z_t = \sqrt{\bar{\alpha}_t} z_0 + \sqrt{1 - \bar{\alpha}_t} \epsilon$ where $\bar{\alpha}_t = \prod_{i=1}^{t} \alpha_i$ denotes the cumulative noise schedule.

The reserve process is learned by a denoising model $\epsilon_\theta$, typically a U-Net, trained to predict the added noise at each timestep. For text-conditioned generation, the model is trained on image-caption pairs $(x, c)$ by minimizing a conditional noise prediction objective:

$$\mathcal{L}_{\text{cond}} = \mathbb{E}_{z_0, t, \epsilon \sim \mathcal{N}(0, \mathbb{I})} \left[ \|\epsilon - \epsilon_\theta(z_t, t, c)\|^2 \right], \tag{1}$$

where $\theta$ denotes the denoising model parameters. To enable scalable guidance during inference without requiring an external classifier, the model is usually trained using classifier-free guidance (Ho & Salimans, 2022). Specifically, the condition $c$ is randomly replaced with a null token embedding $\phi_{\text{null}}$ during training, allowing the model to learn both conditional and unconditional denoising:

$$\mathcal{L}_{\text{uncond}} = \mathbb{E}_{z_0, t, \epsilon \sim \mathcal{N}(0, \mathbb{I})} \left[ \|\epsilon - \epsilon_\theta(z_t, t, \phi_{\text{null}})\|^2 \right]. \tag{2}$$

At inference, the model iteratively denoises the latent under the conditioning input over time steps, and the final denoised latent is decoded into an output image.

### 3.2 PROBLEM STATEMENT

We assume a target LDM trained on a dataset $D$ of image-caption pairs $(x, c)$, partitioned into two disjoint subsets: the member set $D_M$ and the hold-out (non-member) set $D_H$. The adversary's objective is to determine whether a query image $x \in D$ is included in the training set $D_M$. Specifically, the target denoising model $\epsilon_\theta$ is trained on $D_M$. Following prior work (Dubiński et al., 2024), $D_H$ is drawn from the same distribution as $D_M$, ensuring a realistic and challenging evaluation setting for membership inference.

Unlike prior research (Kong et al., 2023; Fu et al., 2023), we consider a more practical and challenging setting in which the adversary has access only to the query image $x$, but not to its ground-truth caption $c$. This assumption reflects real-world deployment scenarios where training annotations are often inaccessible. To address the absence of the ground-truth caption, the attacker is allowed to use an alternative condition $\hat{c}$ (*e.g.*, a generated or inferred caption) in place of $c$.

Formally, we define the membership inference attack of a query image $x$ as a binary function:

$$\mathcal{M}(x, \hat{c}) = \begin{cases} 1, & \text{if } x \in D_M \\ 0, & \text{if } x \in D_H. \end{cases} \tag{3}$$

### 3.3 OBSERVATIONS

To better understand the effect of missing ground-truth captions, we empirically evaluate membership inference when the captions are replaced with externally generated alternatives (*e.g.*, VLM

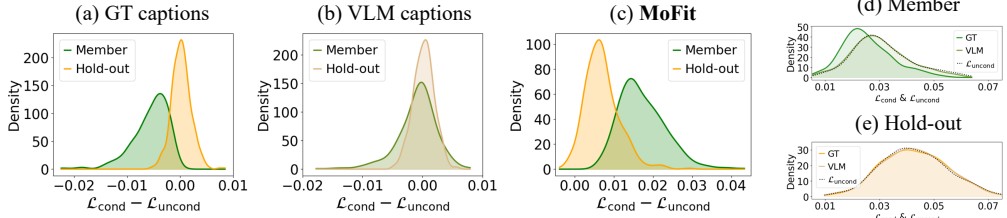

Figure 1: Distribution of membership scores under different condition types: (a) ground-truth captions, (b) VLM-generated captions, and (c) our model-fitted embeddings. In (d), $\mathcal{L}_{\text{cond}}$ values of member samples increase under condition substitution to VLM, whereas hold-out samples remain relatively stable in (e). Dotted lines denote $\mathcal{L}_{\text{uncond}}$, and all distributions are estimated using Gaussian kernel density estimation.

outputs). Motivated by CLiD (Zhai et al., 2024), we contrast the distributional distinctness of the membership scores for $D_M$ and $D_H$ under two attack scenarios: (i) ground-truth captions available, and (ii) ground-truth captions replaced by alternative captions.

**Setup.** CLiD (Zhai et al., 2024) scores membership of a query image by the difference between conditional and unconditional noise-prediction losses:

$$\mathcal{L}_{\text{CLiD}} = \mathcal{L}_{\text{cond}} - \mathcal{L}_{\text{uncond}} = \mathbb{E}_{t,\epsilon}\left[\|\epsilon - \epsilon_\theta(z_t, t, c)\|^2\right] - \mathbb{E}_{t,\epsilon}\left[\|\epsilon - \epsilon_\theta(z_t, t, c_{\text{null}})\|^2\right], \quad (4)$$

where $c$ is the caption paired with image $x$ during training and $c_{\text{null}}$ denotes the unconditional setting (*i.e.*, no-text condition). To establish a caption-free setting where ground-truth captions are unavailable, we replace the caption $c$ with an externally generated description $\hat{c}$ using CLIP-Interrogator[2], a vision-language model (VLM), and substitute them into CLiD (Zhai et al., 2024) framework as conditioning input. For the target diffusion model, we adopt *SD-Pokemon*[3], a Stable Diffusion v1-4 model[4] fine-tuned on the Pokémon dataset (LambdaLabs, 2022). To ensure a fair comparison, we reuse the same noise $\epsilon \sim \mathcal{N}(0, \mathbb{I})$ when calculating Eq. 4 and follow all other settings from the original CLiD framework.

**Observations.** We find a clear degradation in MIA performance when VLM-generated captions $\hat{c}$ are used instead of ground-truth captions $c$, despite their apparent semantic alignment with the images. Under ground-truth conditioning, CLiD yields clearly separable distribution patterns of $\mathcal{L}_{\text{cond}} - \mathcal{L}_{\text{uncond}}$ for members and hold-out samples (Fig. 1(a)). However, conditioning on VLM outputs substantially reduces this separation and incurs largely overlapping score distributions (Fig. 1(b)). Quantitative evaluation results for this degradation are presented in Sec. 5.2.

| Metric | (d) Member | (e) Hold-out |
|---|---|---|
| Mean ± Std (GT) | 0.0253 ± 0.0091 | 0.0433 ± 0.0125 |
| Mean ± Std (VLM) | 0.0300 ± 0.0104 | 0.0432 ± 0.0125 |
| KS Test - Statistic ↑ | 0.2284 | 0.0264 |
| KL Divergence ↑ | 0.7126 | 0.2796 |

Table 1: Quantitative comparison of $\mathcal{L}_{\text{cond}}$ distribution under ground-truth vs. VLM captions for (d) member and (e) hold-out samples in Fig. 1. Arrows indicate the direction of larger deviation.

We attribute the performance degradation to a *difference in sensitivity* to conditioning between members and hold-out samples. As Fig. 1(d) shows, member samples incur a large increase in $\mathcal{L}_{\text{cond}}$ when conditioning is replaced by VLM-generated captions; by contrast, hold-out samples (Fig. 1(e)) exhibit only a modest increase. Meanwhile, $\mathcal{L}_{\text{uncond}}$ stays approximately unchanged for both groups. This asymmetric sensitivity produces a systematic upward shift in the membership score (*i.e.*, $\mathcal{L}_{\text{cond}} - \mathcal{L}_{\text{uncond}}$) for members. Appendix A.3 provides results on additional datasets.

The sensitivity of member and hold-out samples is further supported by the results in Tab. 1, where the mean absolute difference, Kolmogorov–Smirnov (KS) statistic, and Kullback–Leibler (KL) divergence all indicate greater sensitivity of member samples compared to hold-out samples in response to changes in conditioning. Such behavior is intuitive: member samples, having been explicitly exposed to ground-truth captions during training, exhibit increased sensitivity of $\mathcal{L}_{\text{cond}}$ to

---

[2]https://huggingface.co/spaces/pharmapsychotic/CLIP-Interrogator
[3]https://huggingface.co/lambdalabs/sd-pokemon-diffusers
[4]https://huggingface.co/CompVis/stable-diffusion-v1-4

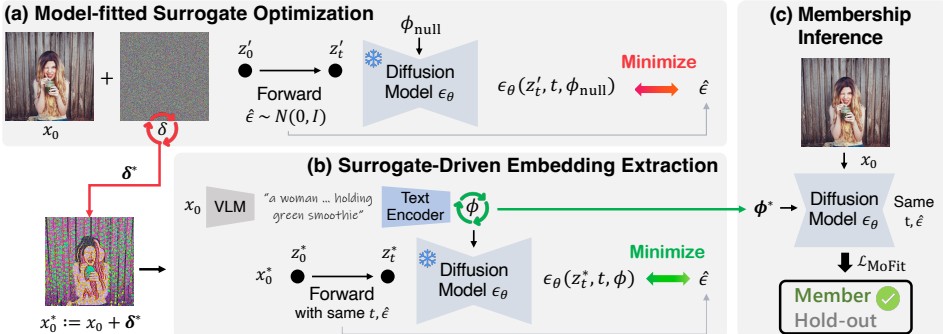

Figure 2: Overview of our proposed method. (a) Given a query image $x_0$, we first optimize a perturbation $\delta$ to overfit to the learned representation from the model. (b) From the resulting surrogate image $x_0 + \delta^*$, we extract a model-fitted embedding $\phi^*$, which is then used as a synthetic condition to amplify the disparity between member and hold-out samples in (c).

condition changes. In contrast, hold-out samples, absent from the training distribution, demonstrate less condition-sensitive behavior. Based on these findings, we summarize the following two key observations:

1. *Member samples are highly sensitive to conditioning*: $\mathcal{L}_{\text{cond}}$ increases consistently when using alternative captions.

2. *Hold-out samples are relatively less affected by conditioning variations*: $\mathcal{L}_{\text{cond}}$ exhibit only minimal changes across different captions.

**Intuition** We propose to exploit the observed sensitivity difference to improve membership inference in the caption-free setting. Specifically, we generate conditioning embeddings that significantly increase $\mathcal{L}_{\text{cond}}$ for member samples while inducing only minimal changes for hold-out samples.

Additionally, we observe that $\mathcal{L}_{\text{uncond}}$ tends to be lower for member samples than for hold-out samples. This is expected, as member samples are directly involved in minimizing $\mathcal{L}_{\text{uncond}}$ during training (Ho & Salimans, 2022). Consequently, embeddings from MOFIT amplify the difference $\mathcal{L}_{\text{cond}} - \mathcal{L}_{\text{uncond}}$ (Eq. 4) for members – via elevated $\mathcal{L}_{\text{cond}}$ and relatively low $\mathcal{L}_{\text{uncond}}$ – while producing less amplification for hold-out samples, which exhibit only modest increases in $\mathcal{L}_{\text{cond}}$ and maintain relatively high $\mathcal{L}_{\text{uncond}}$, thereby reinstating a reliable separability signal (see Fig. 1(c)).

## 4 METHODOLOGY

We propose MOFIT, a caption-free membership-inference framework that leverages ***Mo**del-**Fit**ted* embedding to selectively increase the conditional loss $\mathcal{L}_{\text{cond}}$ for member samples.

In a caption-free setting, one alternative – other than relying on VLMs –is to recover the paired caption of a query image $x_0$ by directly optimizing an embedding for the clean $x_0$ with respect to $\mathcal{L}_{\text{cond}}$ (Eq. 1). However, since the membership status of $x_0$ is unknown, the optimization aligns embeddings for both member and hold-out samples. Therefore, at inference time, such embeddings produce uniformly low $\mathcal{L}_{\text{cond}}$ for both members and non-members, eroding the discriminative signal (see Sec. 5.5 for details).

To address this challenge, we first transform each query image into a surrogate that is strongly overfitted to the target model's learned distribution. Given a query image $x_0$, MOFIT first constructs a *model-fitted* surrogate, *i.e.*, $x_0^* = x_0 + \delta^*$, where $\delta^*$ is a tightly optimized perturbation such that $x_0^*$ appears more coherent with the model's internal distribution when perceived by the target LDM. From this surrogate, we derive its paired embedding $\phi^*$ by minimizing the conditional loss $\mathcal{L}_{\text{cond}}$ (Eq. 1), forming a overfitting pair $(x_0^*, \phi^*)$. For membership inference, MOFIT then conditions the original query $x_0$ with the model-fitted embedding $\phi^*$, *i.e.*, $(x_0, \phi^*)$. Intuitively, given $x_0$, MOFIT constructs a surrogate–embedding pair $(x_0^*, \phi^*)$ that is not only tightly aligned in the model's conditioning space, but also deliberately overfitted to the target model's internal distribution. Thus, conditioning the original query $x_0$ on $\phi^*$ elicits asymmetric sensitivity: member samples incurs

pronounced $\mathcal{L}_{\text{cond}}$ responses while hold-out samples exhibit relatively modest changes. An overview of MoFIT is depicted in Fig. 2.

We note that $x_0$ is not the **exact** counterpart of $\phi^*$; this mismatch $(x_0, \phi^*)$ in inference induces a misalignment between the image and its conditioning. Accordingly, member samples exhibit heightened sensitivity and produce larger $\mathcal{L}_{\text{cond}}$ responses than when conditioned with VLM-generated captions $(x_0, \phi_{\text{VLM}})$, as observed in Sec. 3.3.

## 4.1 MODEL-FITTED SURROGATE OPTIMIZATION

MoFIT constructs a surrogate image that is explicitly optimized to resemble training samples, thereby producing a variant that is intensively adapted to the model's unconditional prior for a given query image. Concretely, it injects a perturbation $\delta$ into the query image $x_0$, *i.e.*, $x_0' = x_0 + \delta$. This surrogate image $x_0'$ is then forwarded to a specific timestep $t$ in the forward process using a single sampled noise vector $\hat{\epsilon} \sim \mathcal{N}(0, \mathbb{I})$, *i.e.*, $z_t' = \sqrt{\bar{\alpha}_t} z_0' + \sqrt{1 - \bar{\alpha}_t} \hat{\epsilon}$, where $z_0'$ denotes the latent encoding of the perturbed image $x_0'$.

In the absence of captions, we use the null conditioning $\phi_{\text{null}}$ and optimize $\delta$ to make the model's unconditional prediction match the sampled noise $\hat{\epsilon}$. Formally, we solve:

$$\delta^* := \arg\min_{\delta} \mathcal{L}_{\text{uncond}} = \arg\min_{\delta} \mathbb{E}_{z_0', t, \hat{\epsilon}} \left[ \|\hat{\epsilon} - \epsilon_\theta(z_t', t, \phi_{\text{null}})\|^2 \right]. \tag{5}$$

To promote strong convergence toward the model's learned manifold, we fix $\hat{\epsilon}$ and $t$ during optimization, thereby stabilizing the direction of perturbation. This *model-fitted* surrogate image $x_0^* = x_0 + \delta^*$ then serves as the input for optimizing an embedding that is tightly coupled with the surrogate.

## 4.2 SURROGATE-DRIVEN EMBEDDING EXTRACTION

Given the model-fitted surrogate $x_0^*$, we aim to extract an embedding $\phi^*$ that reflects the model's response to this model-aligned input. To this end, we treat $\phi$ as an optimizable parameter and minimize the conditional denoising loss (Eq. 1) under the same noise $\hat{\epsilon}$ and timestep $t$ used in the previous surrogate optimization stage:

$$\phi^* := \arg\min_{\phi} \mathbb{E}_{z_0^*, t, \hat{\epsilon}} \left[ \|\hat{\epsilon} - \epsilon_\theta(z_t^*, t, \phi)\|^2 \right]. \tag{6}$$

We initialize $\phi$ with the embedding of a VLM-generated caption, which may serve as a suitable starting point for the optimization. Consequently, $\phi^*$ becomes a conditioning embedding optimized to best describe $x_0^*$, and it is used as the condition in the membership inference stage. Conditioning the original image $x_0$ on $\phi^*$ creates a deliberate image-condition mismatch that elicits a large increase in $\mathcal{L}_{\text{cond}}$ for member samples, while hold-out samples exhibit only a modest change.

## 4.3 MEMBERSHIP INFERENCE WITH AMPLIFIED $\mathcal{L}_{\text{COND}}$ DISPARITY

Given the optimized embedding $\phi^*$ paired with the model-fitted surrogate $x_0^*$, we perform membership inference by computing the discrepancy between conditional and unconditional losses on the original image $x_0$. Since $\phi^*$ is intensively optimized to align with the model-fitted surrogate $x_0^*$ under a fixed $t$ and $\hat{\epsilon}$, but does not correspond to the original image $x_0$, both member and hold-out samples receive a mismatched condition. However, only member samples – sensitive to misaligned conditions – respond with significantly elevated $\mathcal{L}_{\text{cond}}$ values under $\phi^*$, whereas hold-out samples remain less affected by the mismatch. This behavior difference forms the basis of our inference signal.

Accordingly, we define our membership score as the difference between conditional and unconditional denoising losses:

$$\mathcal{L}_{\text{MoFIT}} = \mathbb{E}_{z_0, t, \hat{\epsilon}} \left[ \|\hat{\epsilon} - \epsilon_\theta(z_t, t, \phi^*)\|^2 \right] - \mathbb{E}_{z_0, t, \hat{\epsilon}} \left[ \|\hat{\epsilon} - \epsilon_\theta(z_t, t, \phi_{\text{null}})\|^2 \right]. \tag{7}$$

To further enhance the discriminative power of our attack, we incorporate auxiliary losses that have shown utility in prior work (Zhai et al., 2024). In particular, we consider unconditional loss $\mathcal{L}_{\text{uncond}}$ as well as the CLiD score based on VLM-generated captions. The latter is computed as follows:

$$\mathcal{L}_{\text{VLM}} = \mathbb{E}_{z_0, t, \hat{\epsilon}} \left[ \|\hat{\epsilon} - \epsilon_\theta(z_t, t, \phi_{\text{VLM}})\|^2 \right] - \mathbb{E}_{z_0, t, \hat{\epsilon}} \left[ \|\hat{\epsilon} - \epsilon_\theta(z_t, t, \phi_{\text{null}})\|^2 \right] \tag{8}$$

| Methods | Condition | Pokemon | | | MS-COCO | | | Flickr | | |
|---------|-----------|------|------|----------|------|------|----------|------|------|----------|
| | | ASR | AUC | TPR@1%FPR | ASR | AUC | TPR@1%FPR | ASR | AUC | TPR@1%FPR |
| CLiD | GT | 96.52 | 99.17 | 90.14 | 86.50 | 90.27 | 68.80 | 91.10 | 95.13 | 77.20 |
| Loss | | 72.27 | 78.99 | 4.81 | 63.70 | 67.88 | 4.80 | 61.60 | 64.24 | 5.40 |
| SecMI | | _78.51_ | _86.22_ | 6.97 | 57.30 | 58.07 | 4.20 | 54.00 | 52.38 | 2.00 |
| PIA | VLM | 71.79 | 76.76 | 10.82 | 66.00 | 69.70 | 6.60 | 61.00 | 64.05 | 5.00 |
| PFAMI | | 74.43 | 81.25 | 6.01 | 80.40 | _87.50_ | 29.40 | 76.90 | 84.99 | 24.80 |
| CLiD | | 77.55 | 83.43 | _19.23_ | _80.90_ | 86.53 | **50.80** | _79.00_ | _85.16_ | _40.60_ |
| **MOFIT** | $\phi^*$ | **94.48** | **97.30** | **50.48** | **88.00** | **94.17** | _47.00_ | **86.00** | **91.32** | **53.20** |

Table 2: Comparison of membership inference performance under the caption-free setting, where baseline methods are conditioned using either ground-truth or VLM-generated captions. Bold numbers denote the best, and underlined numbers indicate the second-best results.

where $\phi_{\text{VLM}}$ denotes the embedding of VLM-generated caption. We then formulate the final membership decision rule as a weighted combination of the normalized losses, following the robust-scaling strategy introduced in (Zhai et al., 2024). The corresponding membership prediction is computed as:

$$\mathcal{M}(x, \phi^*) = \mathbf{1}\left[\gamma \cdot \mathcal{R}\left(\mathcal{L}_{\text{MOFIT}}\right) + (1 - \gamma) \cdot \mathcal{R}\left(-\mathcal{L}_{\text{aux}}\right) > \tau\right], \tag{9}$$

where $\mathcal{L}_{\text{aux}} \in \{\mathcal{L}_{\text{uncond}}, \mathcal{L}_{\text{VLM}}\}$, and the negation on $\mathcal{L}_{\text{aux}}$ reflects the inverted loss dynamics of our score function, whereby member samples attain higher scores than hold-out samples. $\mathcal{R}(\cdot)$ denotes the robust scaler defined as $\mathcal{R}(w_i) = (w_i - \tilde{w})/IQR$, with $\tilde{w}$ as the median and $IQR$ as the interquartile range. The hyperparameter $\gamma \in [0, 1]$ controls the balance between our proposed score and auxiliary loss, and $\tau$ is the decision threshold.

## 5 EXPERIMENTS

### 5.1 EXPERIMENTAL SETUP

**MOFIT & Baselines.** We iteratively update the perturbation $\delta$ along the gradient sign direction, employing a step size initialized at 0.15 and linearly decayed in proportion to the iteration count throughout optimization. The resulting model-fitted surrogate is then used to extract an embedding via the Adam optimizer with a learning rate of 0.06. Throughout both optimization stages, the diffusion timestep is fixed at $t = 140$, within a total schedule of $T = 1000$. Optimal hyperparameters are selected based on search results presented in Fig. 5 of Appendix A.1.

We consider five prior methods – Loss-based inference (Matsumoto et al., 2023), SecMI (Duan et al., 2023), PIA (Kong et al., 2023), PFAMI (Fu et al., 2023), and CLiD (Zhai et al., 2024) – as baselines, each conditioned on VLM-generated captions to simulate the caption-inaccessible setting. For real-world datasets (i.e., MS-COCO and Flickr), we generate captions using BLIP-2 (Li et al., 2023), while for the stylized dataset (i.e., Pokemon), we employ CLIP-Interrogator. Additional details are provided in Appendix A.1.

**Target Models.** We evaluate our method on Stable Diffusion v1.4 fine-tuned on three datasets – Pokemon, MS-COCO (Lin et al., 2014), and Flickr (Young et al., 2014) – as well as the pre-trained Stable Diffusion v1.5 [5] model. However, as noted in Dubiński et al. (2024) and further demonstrated in Tab. 6 of Appendix A.1, existing methods perform near chance level on the LAION-mi split – a member/hold-out partition specifically constructed for Stable Diffusion – due to the model's strong generalization. To clearly assess the performance difference between VLM-captioned baselines and our approach, we replace the original member set with 431 verified memorized samples (Webster, 2023), while retaining the LAION-mi hold-out set for evaluation.

**Evaluation Metrics.** We report Attack Success Rate (ASR), AUC, and True Positive Rate at 1% False Positive Rate (TPR@1%FPR), following the standard metrics used in our baselines. For MS-COCO and Flickr, we evaluate on 500 randomly sampled images from each of the member and hold-out sets, while all available images are used for the Pokemon dataset.

---

[5] https://huggingface.co/stable-diffusion-v1-5/stable-diffusion-v1-5

## 5.2 EVALUATION ON FINE-TUNED LDMS

We evaluate the membership inference performance of various methods on latent diffusion models (LDMs) fine-tuned with three distinct datasets – Pokemon, MS-COCO, and Flickr – under a practical threat model where only query images are accessible to the auditor. In Tab. 2, we first report the performance of CLiD (Zhai et al., 2024) when access to ground-truth (GT) captions is granted, representing the upper-bound case (first row). In the caption-free setting, baseline methods may utilize captions generated by VLMs as alternatives (second row). However, this substitution leads to a substantial drop across all evaluation metrics. CLiD's ASR on the Pokemon dataset decreases by nearly 29% when GT captions are replaced by VLM-generated alternatives. This result underscores a critical limitation: while VLMs can generate semantically relevant descriptions, they cannot replicate the exact ground-truth captions, thereby failing to recover the same conditioning effect.

MOFIT conditions each query image $x_0$ using its model-fitted embedding $\phi^*$, which is extracted from a surrogate image $x_0^*$ specifically optimized to align with the model's learned distribution. Crucially, because $\phi^*$ is highly tailored to $x_0^*$, conditioning the query $x_0$ with $\phi^*$ during inference results in a pronounced misalignment. This effect amplifies membership-specific responses, thereby improving inference accuracy. As reported in Tab. 2, MOFIT significantly outperforms prior methods conditioned on VLM-generated captions across both ASR and AUC. Remarkably, it even surpasses CLiD with ground-truth captions on the MS-COCO dataset, indicating that surrogate-based misalignment can serve as a competitive alternative to original training captions.

## 5.3 EVALUATION ON STABLE DIFFUSION

We assess MOFIT on the pre-trained Stable Diffusion v1.5 (SD v1.5) using the modified LAION-mi benchmark (see Sec. 5.1 for details). As demonstrated in Tab. 3, prior methods suffer notable performance degradation when conditioned on VLM-generated captions. In contrast, MOFIT outperforms all VLM-conditioned baselines and even surpasses the GT-captioned CLiD in ASR. Although its AUC is slightly lower, it achieves the highest TPR@1%FPR, exceeding the second-best result

| Methods | Condition | Stable Diffusion v1.5 | | |
|---|---|---|---|---|
| | | ASR | AUC | TPR@1%FPR |
| CLiD | GT | 77.38 | 77.83 | 49.65 |
| Loss | | 68.21 | _74.73_ | 4.87 |
| SecMI | | 55.10 | 54.57 | 8.12 |
| PIA | VLM | 63.92 | 66.74 | 6.03 |
| PFAMI | | 72.85 | **78.15** | _19.26_ |
| CLiD | | 58.12 | 59.28 | 4.18 |
| **MOFIT** | $\phi^*$ | **77.61** | 71.03 | **41.30** |

Table 3: Evaluation on SD v1.5.

by over 20%, demonstrating strong discriminative power in high-precision regimes. We include additional experiments on SD v2.1 (Sec. A.9.2) with a different text encoder and SD v3 with a fundamentally different architecture (Sec. A.9.3).

## 5.4 UNDERSTANDING THE SEPARABILITY OF MOFIT

We attribute the performance gain of MOFIT to the distinct sensitivity characteristics between member and hold-out samples. Interestingly, this disparity is further amplified when conditioning on the model-fitted embedding $\phi^*$. As observed in the MS-COCO dataset (Fig. 3(b, top)), member samples exhibit a pronounced increase in $\mathcal{L}_{\text{cond}}$, indicating a strong sensitivity to the misaligned condition. In contrast, hold-out samples exhibit only a modest increase – their $\mathcal{L}_{\text{cond}}$ distribution closely aligned with that under other conditions (Fig. 3(b, bottom)).

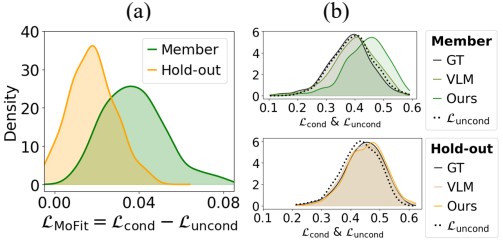

Figure 3: (a) Membership score distributions when conditioned on the model-fitted embedding $\phi^*$. (b) $\mathcal{L}_{\text{cond}}$ and $\mathcal{L}_{\text{uncond}}$ for member and hold-out samples under varying conditions.

As a result, when computing the final discrepancy score $\mathcal{L}_{\text{MoFit}}$ (Eq. 7) in Fig. 3(a), member samples predominantly exhibit positive values due to the pronounced gap between $\mathcal{L}_{\text{cond}}$ and $\mathcal{L}_{\text{uncond}}$. In contrast, hold-out samples yield scores near zero, suggesting minimal impact from the change in conditioning. Results of Pokemon and Flickr datasets are depicted in Appendix A.2.

## 5.5 EFFECTIVENESS OF MODEL-FITTED SURROGATE $x_0^*$

To further evaluate the effectiveness of $x_0 + \delta^*$, we perform an ablation by varying the input image used to extract the embedding $\phi^*$. Specifically, we compare four configurations: (i) the original query image $x_0$ – corresponding to the alternative described in Sec. 4, (ii) the query image with

| Input | Condition | Pokemon | | | MS-COCO | | | Flickr | | |
|---|---|---|---|---|---|---|---|---|---|---|
| | | ASR | AUC | TPR@1%FPR | ASR | AUC | TPR@1%FPR | ASR | AUC | TPR@1%FPR |
| $x_0$ | | 75.63 | 81.64 | 11.06 | 78.00 | 85.59 | 31.00 | 75.50 | 82.75 | 27.20 |
| $x_0 + \delta$ | $\phi$ | 93.99 | 96.42 | 10.34 | 81.70 | 89.76 | 29.20 | 79.60 | 86.63 | 28.60 |
| $x_0 + \delta_{\text{MAX}}$ | | 75.87 | 81.79 | 7.45 | 78.00 | 85.43 | 34.00 | 75.00 | 82.32 | 28.00 |
| **MOFIT** | $\phi^*$ | **94.48** | **97.30** | **50.48** | **88.00** | **94.17** | **47.00** | **86.00** | **91.32** | **53.20** |

Table 4: Quantitative comparison of MIA performance under input image variations.

random noise $x_0 + \delta$, (iii) an adversarial variant $x_0 + \delta_{\text{MAX}}$ optimized to *maximize* Eq. 5, and (iv) our proposed surrogate $x_0 + \delta^*$, which minimizes Eq. 5. For configuration (ii), we add uniformly sampled noise drawn from the range $[-\varepsilon, \varepsilon]$ to the query image, and for each dataset, we sweep $\varepsilon \in \{0.1, 0.2, \ldots, 0.9\}$, reporting the results at the best-performing noise level (0.5 for *Pokemon*, 0.8 for *MS-COCO*, and 0.6 for *Flickr*). Importantly, for (iii), while MOFIT constructs a model-fitted pair $(x_0^*, \phi^*)$ that is tightly coupled and mutually adapted to the target model, $\delta_{\text{MAX}}$ forces the query $x_0$ to explicitly deviate from the model's learned distribution.

As shown in Tab. 4, MOFIT outperforms all alternative input types, underscoring the importance of extracting $\phi^*$ from a surrogate $x_0^*$ that is carefully aligned with the model's learned representation. This pairing forms a mutually adapted structure: $x_0^*$ is tailored to tightly conform to the model, and $\phi^*$ captures this overfitted signal with high fidelity. While random noise $\delta$ offers modest discriminative signals in specific cases (e.g., Pokemon), its performance lacks consistency across different datasets. In contrast, MOFIT achieves stable and superior results in all evaluation metrics and datasets, demonstrating robust generalization to variations in training data. Membership score distributions for each dataset are provided in Appendix A.4, and additional ablation studies are detailed in Appendix A.1.

## 5.6 DISCUSSIONS

### 5.6.1 OVERFITTING DEGREE OF SURROGATE–EMBEDDING PAIRS

To assess whether the two-stage optimization in MOFIT effectively overfits both the surrogate image $x^*$ and the corresponding embedding $\phi^*$, we examine the distributions of $\mathcal{L}_{\text{cond}}$ and $\mathcal{L}_{\text{uncond}}$ computed by the target model. Specifically, we compare these loss values for ground-truth image-caption pairs $(x, c)$ and our model-fitted pairs $(x^*, \phi^*)$.

Fig. 4(a) illustrates the loss distributions of $(x, c)$ from the Pokemon dataset. In Fig. 4(b, left), we present the $\mathcal{L}_{\text{uncond}}$ values of $x^*$ where it corresponds to the outcome of the surrogate optimization stage (Sec. 4.1) in MOFIT. Notably, the $\mathcal{L}_{\text{uncond}}$ values of $x^*$ are significantly lower than those of $x$, indicating that the surrogate has been strongly overfitted to the model's unconditional prior.

In Fig. 4(b, right), we observe a further decrease in $\mathcal{L}_{\text{cond}}$ following the embedding optimization stage

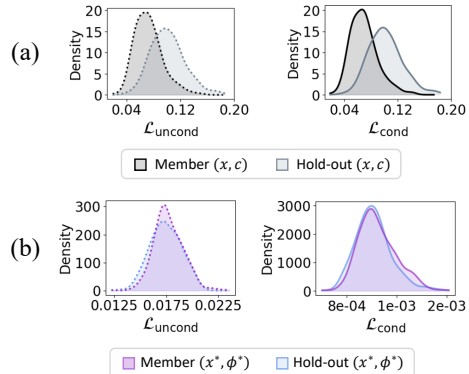

Figure 4: Distributions of (left) $\mathcal{L}_{\text{uncond}}$ and (right) $\mathcal{L}_{\text{cond}}$ of member and hold-out pairs from (a) Pokemon dataset and (b) model-fitted pairs of MOFIT.

(Sec. 4.2), where the embedding $\phi^*$ is specifically tailored to pair with the surrogate $x^*$. Compared to the $\mathcal{L}_{\text{cond}}$ distribution of $(x, c)$ in Fig. 4(a, right), the loss values of $(x^*, \phi^*)$ are not only substantially lower but also more concentrated – exhibiting significantly reduced variance. This suggests that $\phi^*$ is highly aligned with the surrogate $x^*$ on the model's learned manifold. This strong alignment is expected to induce substantial mismatch when $\phi^*$ is applied to the original query image $x$, thereby amplifying the sensitivity of member samples during membership inference. Corresponding loss distributions for other datasets are provided in Appendix A.5.

This uniform overfitting serves as a critical foundation: once the surrogate and its embedding is overfitted and tightly aligned, the embedding can more effectively separate members from hold-outs, as evidenced by the improved performance of MOFIT in Tab. 4.

| Methods | Condition | (a) Gaussian Blur | | | (b) JPEG Compression | | | (c) LoRA | | |
|---------|-----------|-----|-----|-----------|-----|-----|-----------|-----|-----|-----------|
| | | ASR | AUC | TPR@1%FPR | ASR | AUC | TPR@1%FPR | ASR | AUC | TPR@1%FPR |
| CLiD | GT | 89.10 | 92.27 | 70.40 | 85.50 | 89.59 | 61.00 | 59.00 | 52.05 | 1.00 |
| Loss | | 64.10 | 68.17 | 4.20 | 63.40 | 66.53 | 3.80 | 54.50 | 49.26 | 0.00 |
| SecMI | | 60.10 | 62.15 | 5.20 | 54.00 | 54.59 | 2.80 | 58.50 | 53.50 | 1.00 |
| PIA | VLM | 63.10 | 65.28 | 4.40 | 64.90 | 67.18 | 5.00 | 58.50 | 53.50 | 0.00 |
| PFAMI | | 80.60 | _87.51_ | 27.72 | 78.04 | 84.39 | _36.03_ | 73.50 | 77.50 | 1.00 |
| CLiD | | _81.00_ | 87.17 | _44.60_ | _78.80_ | _85.21_ | **39.20** | 59.00 | 53.11 | 1.00 |
| **MOFIT** | $\phi^*$ | **88.70** | **94.92** | **54.20** | **82.80** | **89.75** | 26.00 | 58.50 | 54.35 | 0.00 |

Table 5: Membership inference performance under fine-tuning with data augmentations: (a) Gaussian blur and (b) JPEG compression. (c) Fine-tuning with Low-Rank Adaptation (LoRA) (Hu et al., 2022) also shows potential as a defense.

### 5.6.2 POTENTIAL DEFENSIVE METHOD

**Data Augmentation.** To evaluate resilience against defender-side strategies, we fine-tune SD v1.4 on MS-COCO with two augmentations: (a) Gaussian blur ($3 \times 3$ kernel, $\sigma \in [0.1, 2.0]$) and (b) JPEG compression (quality = 60). MOFIT is then evaluated using embeddings from the non-augmented base model, simulating a challenging setting where the attack is unaware of input-space changes during fine-tuning. Tab. 5(a,b) shows that all methods exhibit comparable or slightly degraded performance under augmentation. Two trends remain: (i) baselines degrade notably when switching from GT to VLM-generated captions; and (ii) MOFIT consistently outperforms all baselines under VLM captions, maintaining ASRs above 82.80% even with augmentation.

**LoRA.** We observe that Low-Rank Adaptation (LoRA) (Hu et al., 2022), which updates only a small set of additional parameters instead of the full U-Net, significantly degrades the performance of MOFIT and existing baselines. As shown in Tab. 5(c), both MoFit and most baselines drop to near-random performance when evaluated on 100 training samples from the LoRA-adapted SDv1.4[6], under both ground-truth and VLM-generated captions. We attribute this robustness to LoRA's minimal footprint, which retains most original weights and reduces memorization capacity (Amit et al., 2024). Additional insights into PFAMI's robustness are provided in Appendix A.7.

### 5.6.3 LIMITATION: RUNTIME AND EARLY STOPPPING

One limitation of MOFIT may be the runtime, taking 7 to 9 minutes per image to optimize the surrogate and extract its embedding (see Appendix A.8.1). Thus, we conduct early stopping strategy that terminates the optimization process once a predefined loss threshold is reached. We evaluate on 100 images each from the member and hold-out splits of MS-COCO, and compare with the state-of-the-art baseline CLiD – performing 83.50% ASR and 87.37 AUC in the same setting. In Tab. 9 of Appendix A.8.2, MOFIT outperforms CLiD when the optimization is stopped at 0.08 during surrogate optimization or at 0.007 during embedding extraction, saving 336.64 and 75.93 seconds, respectively. This suggests that an adversary can strategically balance efficiency and effectiveness by choosing an appropriate early stopping criterion. Additional details are provided in Appendix A.8.2.

## 6 CONCLUSION

We present MOFIT, a novel membership inference framework that operates under a practical caption-free setting. MOFIT enforces separability between member and hold-out samples through a model-fitted surrogate with its tightly optimized embedding, which amplify the conditional loss for member samples while keeping hold-out samples relatively stable. Extensive experiments across multiple benchmarks demonstrate that MOFIT substantially outperforms prior state-of-the-art methods conditioned on VLM-generated captions and, in some cases, even surpasses caption-dependent baselines. We believe this work broadens the scope of membership inference in generative models and underscore the need for stronger safeguards against MIA attacks.

---

[6]https://huggingface.co/sr5434/sd-pokemon-model-lora

## ACKNOWLEDGEMENTS

We would like to thank the reviewers for their constructive comments and suggestions. This work was supported by the National Research Foundation of Korea (NRF) grant funded by the Korea government (MSIT) (No. RS-2023-00208506) and the Institute of Information & Communications Technology Planning & Evaluation (IITP) grant funded by the Korea government (MSIT) (No. RS-2020-II200153, Penetration Security Testing of ML Model Vulnerabilities and Defense). Prof. Sung-Eui Yoon and Prof. Sooel Son are co-corresponding authors.

## ETHICS STATEMENT

This work does not involve human subjects, personally identifiable information, or sensitive user data. All experiments were conducted using publicly available datasets: MS-COCO, Flickr-8k, LAION-mi, Webster (2023), and a released fine-tuned Pokemon diffusion model. Pokemon dataset has been taken down due to copyright issues, so we used the dataset released by the authors of SecMI (Duan et al., 2023).[7]

Our proposed method, MOFIT, is intended to study and evaluate privacy vulnerabilities in generative diffusion models. While membership inference can reveal training data exposure, our findings are strictly aimed at understanding the privacy risks of large-scale generative models and informing the design of more robust, privacy-preserving architectures.

We do not release any models, tools, or datasets that could be directly used to exploit privacy vulnerabilities in deployed systems. We follow responsible disclosure principles and emphasize that our contributions are for defensive and diagnostic purposes only. No personally identifiable data or private user content was used in this research.

## REPRODUCIBILITY STATEMENT

To ensure reproducibility, we provide comprehensive implementation details of our proposed framework MOFIT in the main paper (Sec. 5.1) and include additional training configurations, hyperparameters, and architectural descriptions in the appendix. Dataset used (MS-COCO, Flickr-8k, LAION-mi, Webster (2023), and the released SD-Pokemon model) are publicly available, and we describe the dataset splits and preprocessing steps in Appendix A.1. For key components such as surrogate optimization, embedding extraction, and evaluation metrics (e.g., CLiD score), algorithmic procedures are formalized in the main text, and corresponding ablation and sensitivity analyses are included in Appendix A.1.

## THE USE OF LLMS

The author(s) used ChatGPT for minor grammatical refinements of the manuscript. These modifications have been manually reviewed and finalized by the author(s).

---

[7]https://github.com/jinhaoduan/SecMI-LDM

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

APPENDIX CONTENTS

# A APPENDIX

## A.1 IMPLEMENTATION DETAILS

We provide additional implementation details to improve the reproducibility and clarity of our experimental setup.

**Optimization Details & Ablation Study.** We iteratively optimize the perturbation $\delta$ in the direction of the gradient sign. Specifically, the perturbation is uniformly sample from the range $[-0.3, 0.3]$ and is updated via following equation:

$$x'_{i+1} = x'_i - \alpha \cdot \text{sign}(\nabla_{x'_i}\mathcal{L}_{\text{uncond}}), \tag{10}$$

where $\mathcal{L}_{\text{uncond}}$ denotes the unconditional loss defined in Eq. 5, and $x'_i$ represents the surrogate being optimized at iteration $i$. The step size $\alpha$ is initialized to 0.15 and proportionally decayed with the iteration count.

To justify the choice of $\alpha$, we conduct a hyperparameter analysis across multiple values: $\eta \in \{0.025, 0.05, 0.1, 0.2, 0.3\}$. We perform this analysis on the Pokemon dataset by randomly sampling 100 images from both the member and hold-out sets. As shown in Fig. 5, increasing $\alpha$ generally improves attack performance up to a certain point, with both ASR and AUC peaking at $\alpha = 0.15$. Based on this observation, we use $\alpha = 0.15$ throughout all experiments.

The resulting model-fitted surrogate is then used to extract an embedding via the Adam optimizer with a learning rate of 0.06. The number of optimization steps varies by dataset: 200 steps for Pokemon, 300 steps for MS-COCO and Flickr, and 1000 steps for Stable Diffusion v1.5. The increased iteration count for Stable Diffusion is due to its strong generalization, which requires more steps to be overfitted.

For a single query image, the overall runtime for its two-stage optimization process varies by dataset: approximately 7 minutes for Pokemon, 8 minutes for MS-COCO and Flickr, and 9 minutes for Stable Diffusion v1.5, all measured on a single NVIDIA GeForce RTX 4090 GPU. Please refer to Sec. A.8.1 for more details.

Figure 5: ASR and AUC for different initial step size and timestep $t$.

Both optimization procedures are performed at a fixed diffusion timestep of $t = 140$, within the full denoising schedule of $T = 1000$. This choice is guided by ablation results presented in Fig. 5, where ASR and AUC are evaluated across varying timesteps ranging from $t = 50$ to $t = 700$. The results indicate that performance peaks within the range $t \in [100, 200]$. Accordingly, we adopt $t = 140$ for all experiments.

**Membership Inference.** For the auxiliary loss $\mathcal{L}_{\text{aux}}$ in Eq. 9, we use $\mathcal{L}_{\text{uncond}}$ for Pokemon and Stable Diffusion v1.5, and use $\mathcal{L}_{\text{VLM}}$ for MS-COCO and Flickr. The balancing hyperparameter $\gamma$ is increased by 0.05 within the range $[0, 1]$. Threshold $\tau$ is selected to maximize ASR for each $\gamma$.

**Baselines.** All baseline methods are evaluated using their default settings. For the Naive method (Matsumoto et al., 2023), membership is determined based on $\mathcal{L}_{\text{cond}}$. For SecMI (Duan et al., 2023) and PIA (Kong et al., 2023), we adopt the SecMI-stat and the default PIA, respectively – both rely on threshold-based inference without training a classification model. When ground-truth captions are available, CLiD (Zhai et al., 2024) proposes several caption-splitting strategies (*e.g.*, simple clipping, random noise, and word importance calculation). Among them, we adopt simple clipping. Unlike CLiD, MoFIT follows the settings of SecMI and PIA, assuming access to a subset of member and hold-out samples for threshold calibration; hence, we do not train a shadow model to determine $\gamma$ or $\tau$.

**Target Models.** We evaluate our method on a range of text-to-image diffusion models trained on diverse datasets. We begin with SD-Pokemon, introduced in Sec. 3.3, which fine-tunes Stable

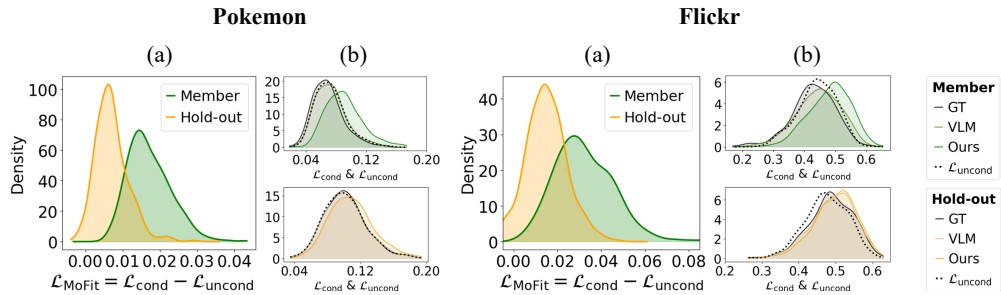

Figure 6: Score distributions on the Pokemon and Flickr datasets. (a) $\mathcal{L}_{\text{MoFit}}$ (Eq. 7) score of member and hold-out samples. (b) $\mathcal{L}_{\text{cond}}$ and $\mathcal{L}_{\text{uncond}}$ under condition variations.

Diffusion v1.4 on 416/417 member/hold-out image-caption pairs for 15,000 steps. Using the same base model, we further fine-tune on 2,500/2,458 splits of MS-COCO and 2,500/2,500 splits of Flickr, each with 150,000 steps, following the experimental setups in prior studies (Duan et al., 2023; Zhai et al., 2024; Kong et al., 2023).

| Methods | Condition | LAION-mi | | |
|---|---|---|---|---|
| | | ASR | AUC | TPR@1%FPR |
| Loss | | 55.02 | 50.72 | 1.60 |
| SecMI | | 53.55 | 53.34 | 2.90 |
| PIA | GT | 54.80 | 49.58 | 2.00 |
| PFAMI | | 52.75 | 51.67 | 0.75 |
| CLiD | | 56.90 | 58.62 | 4.60 |
| CLiD | VLM | 53.80 | 52.09 | 3.2 |

Table 6: Quantitative comparison on SDv1.5 for LAION-mi.

To evaluate our method on a widely-used large-scale model, we additionally consider the pre-trained Stable Diffusion v1.5 [8]. For membership evaluation, we adopt the LAION-mi benchmark (Dubiński et al., 2024), which provides curated member and hold-out splits for Stable Diffusion. However, as discussed in Sec. 5.1, existing methods perform near chance level due to the model's high generative capacity. We evaluate all baseline methods using 500 images each from the member and hold-out sets of LAION-mi. As demonstrated in Tab. 6, ASR of prior methods is almost the same as random guessing, even in the condition of ground-truth captions (Please refer to Appendix A.9.1 for evaluation of MoFit). Accordingly, to enable a more discriminative evaluation, we substitute the member set with verified memorized images, identified using the reproduction methodology in (Webster, 2023). This curated benchmark allows for a more discernible comparison between VLM-captioned baselines and our approach, enabling clearer assessment of each method's effectiveness. Among the 500 image URLs, we utilize the 431 images that were successfully downloaded.

**VLMs.** Since BLIP-2 is designed to generate natural language descriptions for real-world images, we use it to produce alternative captions for the MS-COCO, Flickr, LAION-mi, and Webster (2023) datasets. For the stylized Pokemon dataset, we instead employ CLIP-Interrogator, which is commonly used to infer the underlying text prompts that may have been used to generate a given synthetic image. This choice is further motivated by the fact that BLIP (Li et al., 2022) was previously used to caption the original Pokemon training set.

## A.2 MoFit on Pokemon and Flickr

In addition to Fig. 3, we present the corresponding score distributions for the Pokemon and Flickr datasets in Fig. 6. For both datasets, the discrepancy scores in (a) exhibit clear separability between member and hold-out samples. Furthermore, in (b), the $\mathcal{L}_{\text{cond}}$ values notably increase under the model-fitted embedding condition of MoFit, while the $\mathcal{L}_{\text{uncond}}$ values remain largely unchanged. As discussed in Sec. 5.2, this differential response plays a key role in restoring separability and accounts for the superior performance of MoFit reported in Tab. 2.

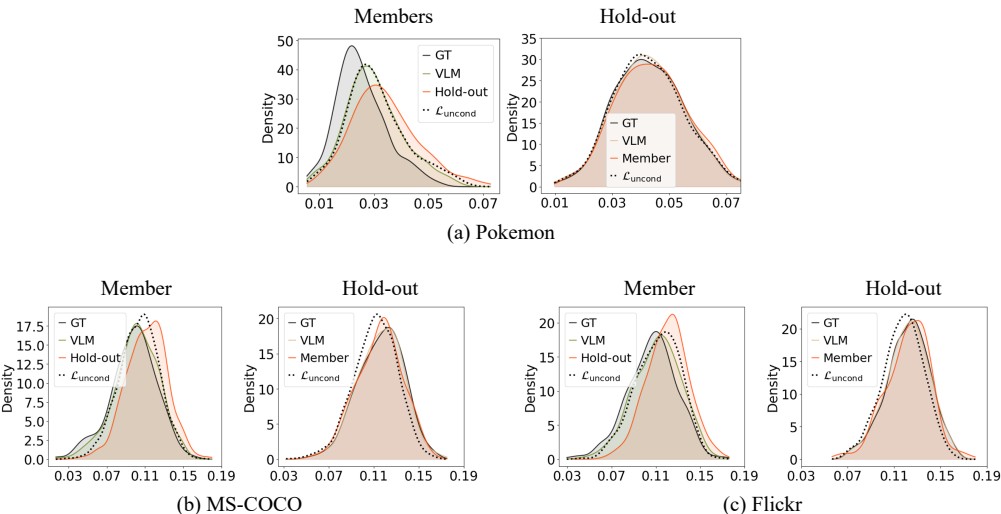

Figure 7: Responses of $\mathcal{L}_{\text{cond}}$ and $\mathcal{L}_{\text{uncond}}$ across multiple datasets under different conditioning schemes.

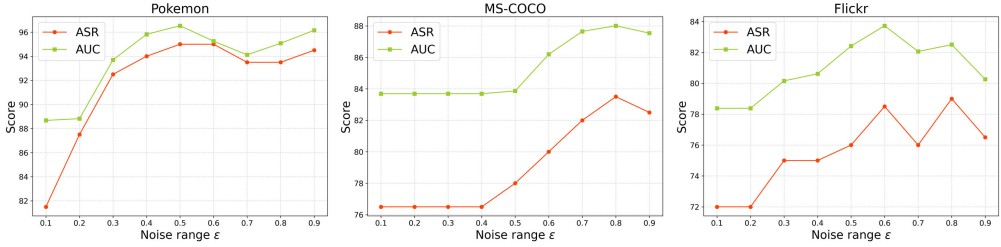

Figure 8: ASR and AUC when embeddings are extracted from perturbed images $x_0 + \delta$ under varying noise levels $\varepsilon$.

## A.3 RESPONSE OF MEMBER AND HOLD-OUT SAMPLES ACCORDING TO THE CONDITION

As observed in Sec. 3.3, member and hold-out samples exhibit differing response to condition changes. We further investigate this deviation by expanding both the dataset and the types of conditioning. In addition to the Pokemon dataset, we include MS-COCO and Flickr to examine whether similar patterns emerge. We also introduce a new type of conditioning: while VLM-generated captions semantically describe the given image, we simulate non-descriptive conditioning by using captions from the opposite group – *i.e.*, member images are conditioned on captions from the hold-out set, and vice versa.

Fig. 7 shows the distributional changes of $\mathcal{L}_{\text{cond}}$ and $\mathcal{L}_{\text{uncond}}$ across all datasets. Member samples consistently exhibit increased $\mathcal{L}_{\text{cond}}$ values when transitioning from ground-truth to VLM-generated captions, with a further increase under non-descriptive captions (red lines). In contrast, hold-out samples remain less affected, even under condition of captions from the member set. These results reinforce the two key observations discussed in Sec. 3.3.

## A.4 INPUT IMAGE VARIATIONS

**Noise level $\varepsilon$ for each dataset in Sec. 5.5.** In the experiment described in Sec. 5.5, embeddings are extracted from perturbed query images of the form $x_0 + \delta$, where $\delta$ is uniformly sampled from $[-\varepsilon, \varepsilon]$. We sweep $\varepsilon \in \{0.1, 0.2, \ldots, 0.9\}$ to identify the optimal noise magnitude for each dataset. Fig. 8 reports ASR and AUC values as functions of $\varepsilon$, computed using 100 member and 100 hold-out

---

[8] https://huggingface.co/stable-diffusion-v1-5/stable-diffusion-v1-5

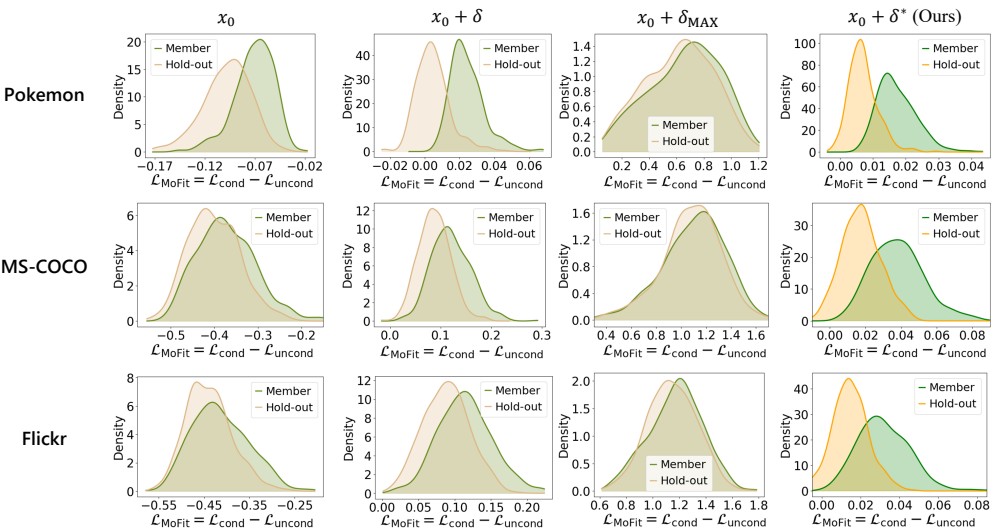

Figure 9: $\mathcal{L}_{\text{MoFit}}$ score distribution according to the input image variations.

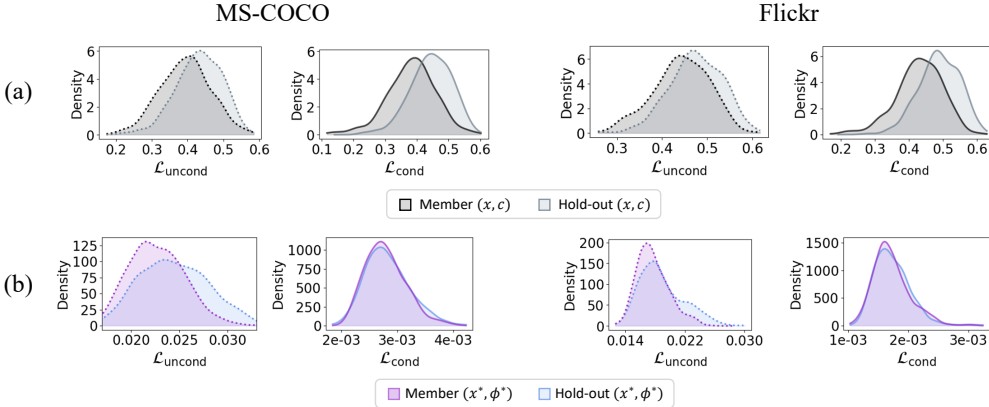

Figure 10: Distributions of $\mathcal{L}_{\text{uncond}}$ (dotted lines) and $\mathcal{L}_{\text{cond}}$ for member and hold-out samples from (a) MS-COCO and Flickr datasets, and (b) model-fitted pairs used in MoFit.

samples per dataset. Based on these results, we select $\varepsilon = 0.5$ for *Pokemon*, $\varepsilon = 0.8$ for *MS-COCO*, and $\varepsilon = 0.6$ for *Flickr*. The corresponding evaluation results on the full datasets are summarized in Tab. 4.

To further support the superior results of MoFit reported in Tab. 4, we present the $\mathcal{L}_{\text{MoFit}}$ score distributions under different input variations: the original image $x_0$, a randomly perturbed image $x_0 + \delta$, an adversarial variant $x_0 + \delta_{\text{MAX}}$, and the model-fitted surrogate $x_0 + \delta^*$ used by MoFit. As shown in Fig. 9, conditioning on $x_0 + \delta^*$ yields the greatest separability, demonstrating the effectiveness of MoFit in leveraging model-fitted surrogates.

## A.5 HANDLING UNKNOWN MEMBERSHIP VIA SURROGATE OVERFITTING

To infer membership status when the original caption is unavailable, an alternative approach – other than VLM-generated captions – is to directly optimize a text embedding for the clean query image $x$ using the conditional loss $\mathcal{L}_{\text{cond}}$ (as discussed in Sec. 4). However, as proven in Tab. 4, this strategy

| Methods | Condition | Pokemon | | | MS-COCO | | | Flickr | | |
|---|---|---|---|---|---|---|---|---|---|---|
| | | ASR | AUC | TPR@1%FPR | ASR | AUC | TPR@1%FPR | ASR | AUC | TPR@1%FPR |
| Loss | | 80.07 | 87.50 | 8.65 | 72.16 | 78.21 | 10.20 | 66.70 | 70.14 | 4.60 |
| SecMI | | 81.99 | 89.53 | 6.49 | 59.20 | 60.47 | 5.00 | 55.00 | 54.66 | 2.40 |
| PIA | GT | 80.43 | 86.90 | 19.95 | 78.28 | 84.64 | 16.40 | 70.31 | 74.94 | 5.80 |
| PFAMI | | 75.75 | 81.06 | 47.10 | 84.80 | 91.41 | 44.60 | 83.00 | 90.58 | 41.20 |
| CLiD | | 96.52 | 99.17 | 90.14 | 86.50 | 90.27 | 68.80 | 91.10 | 95.13 | 77.20 |

Table 7: Performance comparison of membership inference methods when the ground-truth captions are accessible.

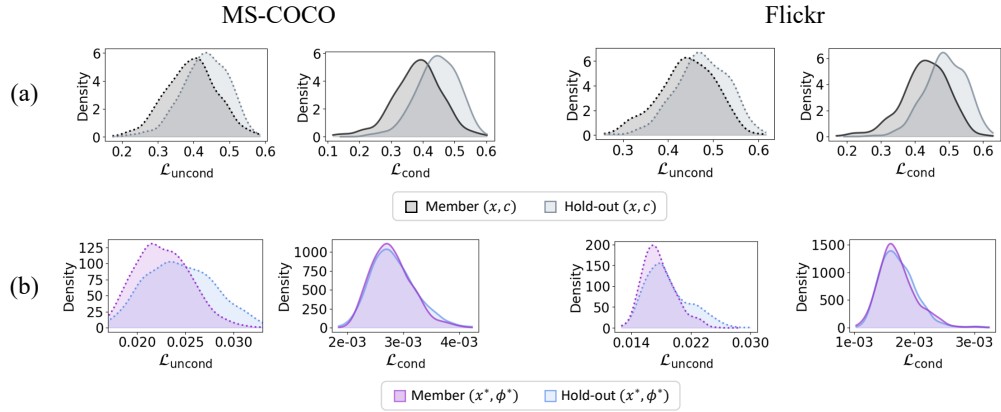

Figure 11: $\mathcal{L}_{\text{uncond}}$ (dotted lines) and $\mathcal{L}_{\text{cond}}$ for member and hold-out samples from (a) MS-COCO and Flickr datasets, and (b) model-fitted pairs used in MOFIT.

results in a clear degradation in performance, suggesting that the clean image alone does not yield sufficiently discriminative embeddings.

In contrast, MOFIT circumvents this limitation by consistently overfitting both member and hold-out query images to the model. Fig. 4(b) demonstrates that both $\mathcal{L}_{\text{cond}}$ and $\mathcal{L}_{\text{uncond}}$ distributions of the model-fitted pairs $(x^*, \phi^*)$ exhibit substantial overlap between member and hold-out samples. This observation implies that MOFIT effectively constructs pairs aligned with the model's learned representation, irrespective of the sample's true membership status.

Building upon this analysis, Fig. 10 presents the $\mathcal{L}_{\text{uncond}}$ and $\mathcal{L}_{\text{cond}}$ distributions for two additional datasets – MS-COCO and Flickr – alongside the corresponding distributions for the model-fitted pairs in MOFIT. For both datasets, the distributions of $(x^*, \phi^*)$ exhibit lower magnitude and higher density compared to those of the original pairs $(x, c)$. These results indicate that MOFIT effectively maps and intensively overfits both the surrogate and their coupled embeddings onto the model's generative manifold, regardless of the dataset.

### A.6 RESULTS OF BASELINES WITH GROUND-TRUTH CAPTIONS

In Tab. 7, we present the performance of baseline methods evaluated with access to ground-truth captions. CLiD is included in the main paper (Sec. 5.2) as it demonstrates the best performance in this setting. Notably, MOFIT performs competitively with CLiD under ground-truth conditions on the MS-COCO dataset.

### A.7 MEMBERSHIP INFERENCE AGAINST LORA-ADAPTED LDMS

In Tab. 5(c), we report the degradation of membership inference methods against LoRA-adapted Stable Diffusion v1.4 fine-tuned on the Pokemon dataset. While Luo et al. (2025) report vulnerabilities in LoRA-adapted LDMs, their analysis focuses on early MIA methods (e.g., the "Loss" baseline) and does not cover recent approaches.

| (a) Surrogate | VRAM (MB) | Runtime (sec) | (b) Embedding | VRAM (MB) | Runtime (sec) |
|---|---|---|---|---|---|
| Adam | 18053.79 | 333.24 | Adam (**Ours**) | 14799.11 | 38.28 |
| SGD | 18052.38 | 333.19 | SGD | 14797.22 | 38.21 |
| RMSProp | 18050.83 | 333.66 | RMSProp | 14803.43 | 38.15 |
| LION | 18049.72 | 333.57 | LION | 14803.64 | 38.15 |
| **Ours** | 20667.50 | 358.96 | | | |

Table 8: Comparison of GPU VRAM usage and runtime overhead across optimizers during (a) surrogate optimization and (b) embedding extraction.

| Threshold | (a) Surrogate Optimziation | | | | Threshold | (b) Embedding Extraction | | | |
|---|---|---|---|---|---|---|---|---|---|
| | ASR | AUC | TPR@1%FPR | Runtime | | ASR | AUC | TPR@1%FPR | Runtime |
| 0.10 | 81.50 | 88.53 | 32.00 | 10.71 | 0.009 | 81.00 | 88.17 | 35.00 | 29.11 |
| 0.09 | 82.00 | 89.67 | 39.00 | 16.33 | 0.008 | 82.00 | 89.62 | 33.00 | 34.37 |
| **0.08** | **84.00** | **90.15** | **42.00** | **22.32** | **0.007** | **84.00** | **90.50** | **31.00** | **40.45** |
| 0.07 | 85.00 | 90.68 | 49.00 | 33.62 | 0.006 | 85.00 | 91.57 | 37.00 | 48.07 |
| 0.06 | 86.50 | 91.00 | 39.00 | 55.41 | 0.005 | 86.00 | 92.97 | 43.00 | 58.19 |
| Total | | | | 358.96 | Total | | | | 116.38 |

Table 9: Performance of MoFit under an early stopping regime applied to both surrogate optimization and embedding extraction.

**PFAMI robustness.** Unlike other baselines, PFAMI remains relatively robust under LoRA. We attribute this to its distinct metric formulation.

Baselines other than PFAMI rely on *cross-query* comparisons – typically based on the relative ranking of diffusion losses across different queries. The poor performance of these baselines under LoRA, as shown in Tab. 5(c), suggests that the loss values of member and non-member queries become increasingly similar. This makes it difficult for cross-query methods to distinguish between them based on relative loss ordering.

However, PFAMI uses *within-query* scores by applying multiple augmentations (*i.e.*, crop) to a single query image and computing the loss for each. The membership score is based on the relative variation within these losses. Since it does not rely on comparisons across different queries, PFAMI remains robust even when LoRA causes the overall loss values of member and non-member samples to become similar. The relative differences within a single query are preserved, making it less sensitive to such global shifts.

## A.8 RUNTIME OF MoFit AND EARLY STOPPING STRATEGY

We report VRAM usage and runtime of MoFit in Appendix A.8.1. While MoFit incurs high time cost, our focus is on achieving strong membership inference performance without relying on VLM-generated captions – demonstrating the first such result in the *caption-free setting*. Given this goal, we prioritized accuracy over efficiency. Nevertheless, recognizing the importance of cost-efficiency, we further explore the trade-off via early stopping in Appendix A.8.2.

### A.8.1 OPTIMIZER-SPECIFIC VRAM USAGE AND RUNTIME

We perform ablation studies comparing lightweight optimizers – SGD, RMSProp, and LION (Chen et al., 2023).

**(i) Model-fitted surrogate optimization.** MoFit updates the perturbation using a single-step, first-order update that directly applies the sign of the loss gradient – $\text{sign}(\nabla \mathcal{L}_x)$ – to the input image. As evaluated in Tab. 8(a) for 1,000 steps (default setting), this design results in higher GPU consumption (up to 2,618 MB VRAM) and slightly longer runtime (up to 25.77s) than optimizer-based alternatives. However, this single-step method avoids the need for multiple optimization trials of hyperparameter tuning (*e.g.*, learning rate or momentum), which can be costly and dataset-dependent. We view this trade-off as a design choice left to the adversary, depending on their resource constraints and deployment scenario.

**(ii) Surrogate-driven embedding extraction.** For the embedding extraction step from the model-fitted surrogate, we employ the Adam optimizer by default. In Tab. 8(b), while the optimizers

were evaluated using 200 optimization steps (default setting for SD v1.4 fine-tuned with Pokemon dataset), we find that all optimizers perform consistently, with only a subtle difference. Given its stability and wide applicability across diverse tasks, we find Adam to be a practical default.

### A.8.2 EARLY STOPPING STRATEGY

To reduce the runtime of MOFIT, we perform additional experiments using an early stopping strategy. Specifically, we terminate the optimization once a predefined loss threshold is reached, applying this strategy to both (i) surrogate optimization and (ii) embedding extraction. All experiments are conducted using the SD v1.4 model fine-tuned on the MS-COCO dataset. This setup allows us to assess whether competitive performance can be achieved with reduced computational cost.

**(i) Model-fitted surrogate optimization.** We first average the final loss (Eq. 5) from full optimization runs, which yield 0.02446 for members and 0.02259 for hold-outs. We then re-run the optimization and terminate early when the loss reaches a higher predefined threshold, choosing from 0.1, 0.09, 0.08, 0.07, 0.06. When a threshold is met, we save the corresponding surrogate and proceed to extract the embedding using the same setup as in the main experiments (300 iterations for MS-COCO).

In Tab. 9(a), we report membership inference results and average GPU runtime when applying early stopping to the surrogate optimization stage of MOFIT. We evaluate on 100 member and 100 hold-out images from the MS-COCO dataset, using an NVIDIA RTX 4090. Notably, when compared to our baseline CLiD (83.50% ASR and 87.37 AUC in the same setting), MOFIT reaches comparable or superior performance even when stopped early – for instance, at a loss threshold of 0.08, it achieves competitive results to CLiD while only taking 22.32 seconds per image (originally takes 358.96 sec). This suggests that an adversary can strategically balance efficiency and effectiveness by choosing an appropriate early stopping criterion.

**(ii) Surrogate-driven embedding extraction.** We follow the same procedure and compute the average final loss of Eq. 6 after embedding extraction, obtaining values of 0.00232 for members and 0.00229 for hold-outs. We then re-run full surrogate optimization (1,000 steps as default) for all samples, extract embeddings, and apply early stopping when the loss in Eq. 6 reaches a preset threshold: 0.009, 0.008, 0.007, 0.006, 0.005.

As shown in Tab. 9(b), we again observe a trade-off between runtime and attack performance, mirroring the pattern in (i). Notably, MOFIT reaches comparable performance to CLiD at a threshold of 0.007, while reducing runtime to 75.93 seconds per image. This demonstrates that early-stopping can be effectively applied at both optimization and embedding stages to flexibly adjust the cost-performance balance of MOFIT.

### A.9 EVALUATION AGAINST STABLE DIFFUSION V1.5, V2.1, AND V3

We implement additional experiments on three publicly available pre-trained diffusion models with increasing scale and diversity: (1) Stable Diffusion v1.5 (SD v1.5), (2) Stable Diffusion v2.1 [9] (SD v2.1), which differs in the text encoder with SD v1 (switch from CLIP to OpenCLIP (Cherti et al., 2023)) and (3) Stable Diffusion v3 [10] (SD v3), which adopts a different architecture based on Transformers instead of U-Net.

### A.9.1 STABLE DIFFUSION V1.5

We evaluate on SD v1.5 using the LAION-mi split (Dubiński et al., 2024), a standard member/hold-out partition. We also increase the optimization timestep from $t = 140$ to $t = 350$, which improves MOFIT's performance on large-scale models. As shown in Tab. 10(a), although the task remains challenging, MOFIT consistently outperforms VLM-captioned baselines and even remains competitive with CLiD under GT captions. These results confirm MOFIT 's strong transferability to large-scale public models in realistic, caption-free settings.

---

[9]https://github.com/Stability-AI/stablediffusion
[10]https://huggingface.co/stabilityai/stable-diffusion-3-medium-diffusers

| Methods | Condition | (a) SD v1.5 | | | (b) SD v2.1 | | | (c) SD v3 | | |
|---|---|---|---|---|---|---|---|---|---|---|
| | | ASR | AUC | TPR@1%FPR | ASR | AUC | TPR@1%FPR | ASR | AUC | TPR@1%FPR |
| CLiD | GT | 60.00 | 58.13 | 1.00 | 58.00 | 55.82 | 2.00 | 67.50 | 71.64 | 5.00 |
| Loss | | 52.00 | 50.48 | 0.60 | 55.50 | 52.47 | **1.00** | 53.00 | 46.27 | 0.00 |
| SecMI | | 52.50 | 50.28 | **2.00** | 57.50 | 54.37 | 0.00 | 62.50 | 65.06 | **4.00** |
| PIA | VLM | 51.50 | 49.75 | 1.60 | 59.00 | 57.53 | 0.00 | 55.00 | 50.84 | 0.00 |
| PFAMI | | 54.00 | 52.58 | 0.75 | 51.50 | 39.63 | **1.00** | 68.50 | 69.81 | 3.00 |
| CLiD | | 56.50 | 52.78 | 1.00 | 53.50 | 51.90 | **1.00** | 67.50 | 71.59 | **4.00** |
| MOFIT | $\phi^*$ | 60.00 | **58.23** | 2.00 | **61.34** | **58.99** | 0.00 | **70.00** | **73.42** | 2.00 |

Table 10: Evaluation against three large-scale pre-trained models: SD v1.5 (LAION-mi train vs. test), SD v2.1 (LAION-mi train vs. CC3M), and SD v3 (COCO vs. CC3M).

### A.9.2 STABLE DIFFUSION V2.1

**Defining Member/Hold-out Splits for SD v2.1.** Unlike SD v1.5, which allows controlled evaluation via the curated LAION-mi split (Dubiński et al., 2024), evaluating membership inference on SD v2.1 presents a unique challenge. SD v2.1 is trained on LAION-5B, a superset that contains both the member and hold-out images of LAION-mi. Thus, LAION-mi cannot be reused directly for testing generalization performance on SD v2.1.

| LAION-mi member vs. | COCO | Flickr | CC3M | LAION-mi hold-out |
|---|---|---|---|---|
| FID | 53.6041 | 61.8270 | **16.0582** | 8.8673 |

Table 11: FID scores between LAION-mi member set and several datasets.

To address this challenge, we construct a distributionally aligned evaluation split, following best practices in LAION-mi that recommend closely matched real-world distributions for member and hold-out samples. We compute FID scores between the LAION-mi member set and candidate datasets (MS-COCO (Lin et al., 2014), Flickr (Young et al., 2014), CC3M (Sharma et al., 2018)) in Tab. 11, and find that CC3M yields the lowest FID (16.06), closely matching the intra-LAION-mi FID (8.87). Accordingly, we use LAION-mi members as the member set and CC3M as the hold-out set for SDv 2.1 evaluation.

**Evaluation on SDv2.1 using real-world split.** We perform membership inference attacks using 100 images from each split and evaluate MOFIT alongside all baselines. In Tab. 10(b), under VLM-generated captions – a realistic setting where the adversary lacks GT annotations – all baselines exhibit notable drops in ASR and AUC. In contrast, MOFIT consistently outperforms them, achieving a +2.34% ASR gain over the second-best method under these restricted conditions.

### A.9.3 STABLE DIFFUSION V3

**Defining Member/Hold-out Splits for SD v3.** SD v3 provides no publicly available details about its training dataset aside from the number of training images, making it challenging to construct reliable member/hold-out splits.

Empirically, as in Fig. 12, we observed that COCO samples yield significantly lower conditional and unconditional loss values (Eq. 1 and Eq. 2 in the main paper) compared to CC3M. Based on this, we selected COCO as the member set and CC3M as the hold-out set for evaluating MOFIT on SDv3.

**Evaluation on Pre-trained SDv3.** We conducted membership inference using 100 images from each split, running all methods at a fixed diffusion timestep (t = 328.1250, i.e., step 140). As shown in Tab. 10(c), MOFIT outperforms all baselines, in-

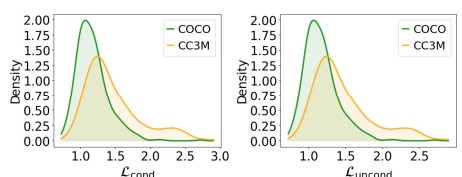

Figure 12: $\mathcal{L}_{\text{cond}}$ and $\mathcal{L}_{\text{uncond}}$ of SD v3 for COCO and CC3M.

cluding CLiD under ground-truth conditions. These results further confirm that MOFIT can match or even exceed methods that rely on full image-caption supervision, consistent with our findings on SD v1.4 (See Tab. 2.)

### A.10 EVALUATION AGAINST STABLE DIFFUSION FINE-TUNED ON MEDICAL DATASET

| Methods | Condition | (a) SD v1.5 | | | (b) SD v2.1 | | | (c) SD v3 | | |
|---|---|---|---|---|---|---|---|---|---|---|
| | | ASR | AUC | TPR@1%FPR | ASR | AUC | TPR@1%FPR | ASR | AUC | TPR@1%FPR |
| CLiD | MoonDream2 | 74.91 | 80.20 | 13.22 | 77.50 | 83.41 | 31.00 | 75.20 | 81.62 | 33.40 |
| CLiD | Molmo | 78.00 | 81.40 | 8.00 | 72.00 | 71.65 | 17.00 | 68.50 | 69.23 | 21.00 |

Table 13: Evaluation of two recent VLMs: MoonDream2 and Molmo Flux captioner.

We additionally evaluated MOFIT on a domain-specific Latent Diffusion Model (LDM) trained on medical data. Specifically, we used Prompt2MedImage[11], a Stable Diffusion v1.4 model fine-tuned on the ROCO dataset (Pelka et al., 2018), which contains radiology image-caption pairs. The training and validation splits were used as member and hold-out sets, respectively. Additionally, because general-purpose VLMs cannot reliably caption domain-specific medical images, we employ a BLIP model fine-tuned on the ROCO dataset[12] to generate the required captions in the caption-free setting.

| Methods | Condition | Prompt2MedImage | | |
|---|---|---|---|---|
| | | ASR | AUC | TPR@1%FPR |
| CLiD | GT | 60.50 | 60.30 | 0.00 |
| Loss | | 54.00 | 47.27 | 0.00 |
| SecMI | | 53.00 | 46.20 | 0.00 |
| PIA | VLM | 51.50 | 44.25 | 0.00 |
| PFAMI | | 53.00 | 50.22 | 1.00 |
| CLiD | | 55.00 | 49.78 | 1.00 |
| **MOFIT** | $\phi^*$ | **57.00** | **54.44** | **2.00** |

Table 12: Evaluation against Prompt2MedImage, a SD v1.4 fine-tuned on ROCO dataset.

As shown in Tab. 12, MOFIT consistently outperforms all VLM-based baselines across all metrics, successfully distinguishing member samples from hold-outs even in this specialized medical domain. These results confirm that MOFIT generalizes across domains, highlighting its effectiveness on in-the-wild LDMs beyond general-purpose text-to-image settings.

## A.11 RECENT VISION-LANGUAGE MODELS (VLMS)

In Tab. 13, we conduct experiments for MoonDream2 [13] and Molmo Flux captioner (Deitke et al., 2025) to explore caption generation using recent vision-language models (VLMs). We condition CLiD with the VLM-generated captions and evaluate on three fine-tuned SD v1.4 models (Pokemon, MS-COCO, and Flickr).

Compared to Tab. 2, we still observe notable performance degradation when using VLM-generated captions. These results highlight that even highly descriptive captions from powerful VLMs fail to recover the original image-caption supervision signal, underscoring the need for research in caption-free settings as our approach.

| | ASR | AUC | TPR@1%FPR |
|---|---|---|---|
| Random 1 | 94.50 | 96.96 | 18.00 |
| Random 2 | 94.00 | 96.23 | 25.00 |
| Random 3 | 94.00 | 95.85 | 18.00 |
| Random 4 | 95.50 | 96.72 | 41.00 |

Table 14: Performance of MOFIT under four random target noise settings.

## A.12 STABILITY OF MOFIT AGAINST RANDOM TARGET NOISE $\hat{\epsilon}$

To evaluate MOFIT's sensitivity to hyper-parameters, we conducted an additional experiment where the target noise $\hat{\epsilon}$ is randomized. Specifically, we sample four random noise vectors from a standard normal distribution and set each as the target noise. This evaluation helps confirm that MOFIT remains robust to the choice of target noise, suggesting that any noise drawn from a normal distribution is sufficient for our method. The experiment was conducted on SD v1.4 fine-tuned with the Pokemon dataset, using 100 member and 100 hold-out samples. As shown in Tab. 14, MOFIT performs consistently across all metrics when implemented with random noise.

## A.13 FAILURE CASES OF MOFIT

In this section, we analyze extreme member and hold-out cases observed under MOFIT, highlighting their distinctive characteristics. Specifically, using the two fine-tuned SD v1.4 models (MS-COCO and Flickr), we select the top-20 samples from each of the following groups:

[11] https://huggingface.co/Nihirc/Prompt2MedImage
[12] https://huggingface.co/Siddartha01/blip-medical-captioning-roco
[13] https://moondream.ai/

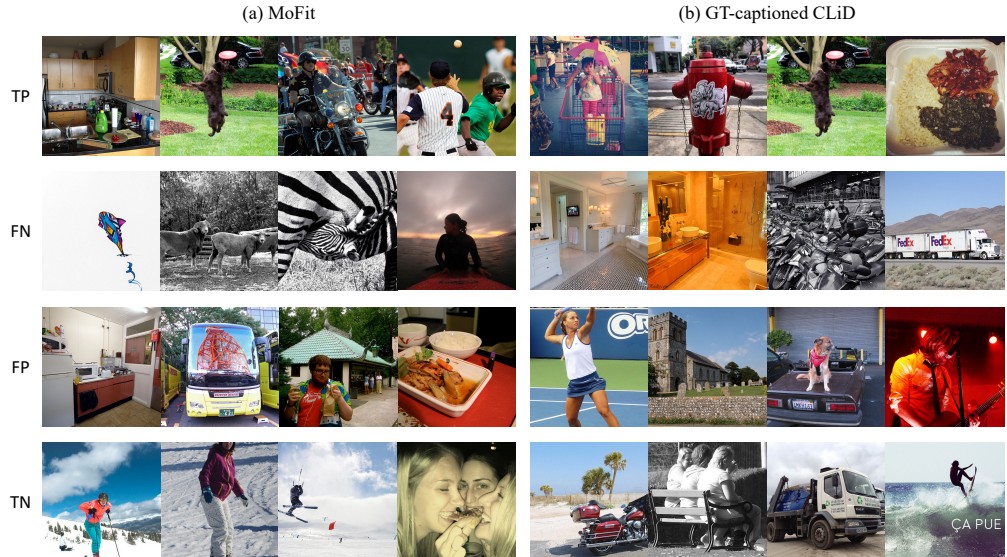

Figure 13: Visualization of extreme samples from (a) MOFIT and (b) GT-captioned CLiD settings.

| Methods | MS-COCO | | | | Flickr | | | |
|---|---|---|---|---|---|---|---|---|
| | TP | FN | FP | TN | TP | FN | FP | TN |
| Colorfulness | 135.27 | 122.65 | 147.64 | 132.54 | 146.22 | 138.56 | 147.51 | 147.95 |
| Entropy | 15.22 | 13.70 | 15.11 | 14.27 | 15.55 | 15.05 | 14.92 | 15.33 |
| Keypoint | 500.00 | 471.95 | 499.25 | 492.20 | 500.00 | 494.55 | 499.65 | 497.75 |

Table 15: TP, FN, FP, and TN statistics using color diversity and image complexity metrics for two fine-tuned SD 1.4 models.

- TP (True Positives): members predicted as members
- FN (False Negatives): members predicted as hold-outs
- FP (False Positives): hold-outs predicted as members
- TN (True Negatives): hold-outs predicted as hold-outs

We then examine the color diversity and image complexity across these extreme samples. To quantify color diversity, we used the colorfulness score (Hasler & Suesstrunk, 2003), and to assess image complexity, we employed Shannon entropy (Shannon, 1948) and ORB keypoint count (Rublee et al., 2011). Higher colorfulness indicates richer chromatic variation, while higher entropy and keypoint counts reflect greater structural complexity.

Interestingly, as shown in Tab. 15, we observed that samples predicted as members (TP and FP) consistently exhibit higher values across all metrics compared to their hold-out counterparts (FN and TN). This suggests that the model is more likely to associate colorful and structurally rich images with training membership. We believe this finding may offer an interesting direction for future work on understanding what types of images diffusion models are more likely to retain.

**Visualization of extreme samples.** We first visualize representative extreme cases (TP, FN, FP, TN) of MS-COCO from our MOFIT setup in Fig. 13(a). Among member samples, FN images tend to exhibit low visual complexity and monotonous color schemes (*e.g.*, plain black-and-white scenes or sky/ocean backgrounds), whereas TP samples show vibrant colors (*e.g.*, green hues) and diverse semantic components (*e.g.*, kitchen items, road scenes). A similar pattern is observed in hold-out samples: FP images often include vivid colors (*e.g.*, green, red) and rich details (*e.g.*, reflection of a tower, food on the table), while TNs are visually simpler (*e.g.*, snowy landscapes, minimal objects).

We additionally include Fig. 13(b) to visualize representative extreme cases under the presence of GT captions, using samples extracted from the GT-captioned CLiD setup. While the extreme

| Methods | Condition | (a) 500 images | | | (b) 1,000 images | | |
|---|---|---|---|---|---|---|---|
| | | ASR | AUC | TPR@1%FPR | ASR | AUC | TPR@1%FPR |
| CLiD | GT | 80.50 | 83.35 | 55.00 | 83.00 | 86.76 | 60.00 |
| Loss | | 64.50 | 68.05 | 5.00 | 64.50 | 67.72 | 4.00 |
| SecMI | | 62.00 | 62.31 | 4.00 | 58.00 | 59.49 | 4.00 |
| PIA | VLM | 66.50 | 70.71 | 7.00 | 69.50 | 71.83 | 9.00 |
| PFAMI | | 83.00 | 87.26 | 23.00 | 83.50 | **90.45** | **31.00** |
| CLiD | | 76.00 | 80.32 | 23.00 | 78.50 | 82.55 | 29.00 |
| **MOFIT** | $\phi^*$ | **86.00** | **91.89** | **27.00** | **88.00** | 90.35 | 30.00 |

Table 16: Performance against LDMs fine-tuned with only few training samples: (a) 500 or (b) 1,000 images.

| Methods | Condition | Pokemon | | | MS-COCO | | | Flickr | | |
|---|---|---|---|---|---|---|---|---|---|---|
| | | ASR | AUC | TPR@1%FPR | ASR | AUC | TPR@1%FPR | ASR | AUC | TPR@1%FPR |
| ReDiffuse | GT | 52.40 | 49.53 | 0.72 | 54.20 | 52.41 | 1.20 | 60.00 | 48.23 | 0.20 |
| ReDiffuse | VLM | 52.16 | 49.75 | 0.72 | 55.30 | 53.37 | 1.60 | 51.30 | 48.73 | 0.20 |

Table 17: Evaluation of ReDiffuse under GT and VLM-generated captions.

samples are not fully identical, the same trend holds: FNs are visually plain, whereas FPs exhibit greater colorfulness and complexity. This consistency suggests that the observed behaviors of FPs and FNs may reflect a general property of latent diffusion models, offering a promising direction for future work on understanding which types of images are more likely to be retained or less trained in the model.

## A.14 LIMITED NUMBER OF TRAINING SAMPLES

We assume a case where a Stable Diffusion model is fine-tuned using a limited number of training images. To simulate this low-resource scenario, we fine-tune SD v1.4 using only 500 or 1,000 images from the MS-COCO dataset over 30,000 or 60,000 steps, respectively – maintaining the same steps-to-image ratio used in our main experiments (2,500 images for 150,000 steps), and following the setting introduced in CLiD.

Tab. 16 reports the membership inference results for all baseline methods and MOFIT, evaluated on 100 member and 100 hold-out images. As expected, prior methods show a noticeable performance drop when moving from ground-truth captions to VLM-generated captions, highlighting the difficulty of caption-free MIA in this sparse data regime. In contrast, MOFIT continues to outperform baselines even under the VLM-generated caption setting, demonstrating strong performance despite the reduced training data. These results indicate that MOFIT generalizes well to low-resource LDMs, suggesting broader applicability and robustness in practical scenarios where only a limited number of private images are available for fine-tuning.

## A.15 COMPARISON WITH A BLACK-BOX METHOD: REDIFFUSE

We additionally evaluate a recent black-box membership inference method, ReDiffuse (Li et al., 2024). ReDiffuse proposes a black-box attack that infers membership based on the reconstruction error between a query image and multiple generated samples. While ReDiffuse is black-box with respect to model architecture and parameters, it assumes access to ground-truth (GT) captions during inference. Thus, in the caption-free setting, it still relies on VLM-generated captions, similar to prior baselines.

While ReDiffuse was originally evaluated under an easier protocol where member and hold-out distributions differ, we re-assess it under our more realistic, distributionally aligned setting. Comparing Tab. 17 with Tab. 2, its performance drops significantly under both GT and VLM-generated captions. This suggests that while ReDiffuse is effective when member and hold-out distributions are disjoint, its ability to infer membership deteriorates in more realistic scenarios where the distributions are matched.

