# OpenReview forum: "No Caption, No Problem: Caption-Free Membership Inference via Model-Fitted Embeddings"
_ICLR.cc/2026/Conference — ICLR 2026 Poster_

### Official Review · Reviewer_edeD · 2025-10-17

**Soundness:** 3
**Presentation:** 4
**Contribution:** 3
**Rating:** 6
**Confidence:** 3

**Summary:**

The authors are tackling the task of membership inference attack without access to ground truth captions. The idea is to find a way to distinguish between samples involved in the training of the generative diffusion model and those who don't. The problem become more feasible when not relying on ground truth captions since it is more applicable in real-life. The idea is to find cases of privacy violation of copyrights. The main conceptual idea of the method is that the difference between the conditional loss (conditioned on the caption, commonly referred as guidance-free generation), and the uncoditional one is at its highest on hold-out samples.

**Strengths:**

* The problem is stated explicitly and in a very understandable and accessible manner.
* The motivation of the method, and the intuition behind the structural choices is very clear and detailed, make it very accessible even for people which are less expert in this field.
* The experimental part is very impressive, the results of MoFit are better with a large margin then the competitors.

**Weaknesses:**

* The authors claim to be the first to tackle the caption-free MIA technique on generative diffusion models whereas [1], which not cited, explicitly claim to be the first (and already published). The authors should mention it and differentiate themselves from it, and to tone down their primacy in the field if it was already done previously.
* The evaluation time is super high - so the method is very slow and perhaps not feasible for set of images which is more than a few (7-9 minutes for a single image is reported in the Supp).
* I didnt see (correct me if Im wrong) any evaluation on a wide range of diffusion processes. Im not talking about the same architecture trained on several different datasets. I think that if you show that your approach is architecture-agnostic, namely when a novel diffusion mechanism will arrive your approach will highly likely work on it as well.
* I think it would be nice to analyze several fail cases, to better understand on which cases the method fails, even a short discussion on it (help follow-up works to understand the limitation of the method).


refs:

[1] Towards Black-Box Membership Inference Attack for Diffusion Models. Li et al. ICML 25.

**Questions:**

* for $\lambda$ optimization, in Eq. 5. The min is over all $\lambda$s, when there is no $\lambda$ in the formula itself. I assume that $z_t$ can be written as a function of $x_0$, it is somehow should be stated clearly, since minimizing on a variable which is not argument in the formula is confusing.
* it is not stated explicitly based on which LDM table 2 was produced?


I think that despite that caption-free methods were probably published earlier than this work, the paper is well written, and it seems novel to me. The paper flows nicely, there is a lot of elaboration on the intuition behind empirical decisions, which make it much more intuitive for the reader. The results of the paper are impressive, and I think that the authors cover relevant and timely topic with nice academic progress.

---

> ### Author Response · Authors · 2025-11-22
>
> ## **Overview of All Reviewer Comments (1/2)**
>
> To thoroughly address the reviewers’ questions and concerns, we conducted a comprehensive set of additional experiments. For clarity and convenience, we summarize all relevant tables below. We kindly invite you to review this summary and refer to the corresponding comments (Comment A to F) for detailed explanations.
> **Note.** T@1F in the table refers to  the TPR@1%FPR metric.
>
> ---
>
> ### **| W1. Novelty of MoFit in the caption-free setting**
>
> |  |  | Pokemon |  |  | MS-COCO |  |  | Flickr |  |  |
> | ----- | ----- | :---: | ----- | ----- | :---: | ----- | ----- | :---: | ----- | ----- |
> |  |  | ASR | AUC | T@1F | ASR | AUC | T@1F | ASR | AUC | T@1F |
> | GT | CLiD | 96.52 | 99.17 | 90.14 | 86.50 | 90.27 | 68.80 | 91.10 | 95.13 | 77.20 |
> | GT | **ReDiffuse** | 52.40 | 49.53 | 0.72 | 54.20 | 52.41 | 1.20 | 60.00 | 48.23 | 0.20 |
> | VLM | Loss | 72.27 | 78.99 | 4.81 | 63.70 | 67.88 | 4.80 | 61.60 | 64.24 | 5.40 |
> | VLM | SecMI | 78.51 | 6.22 | 6.97 | 57.30 | 58.07 | 4.20 | 54.00 | 52.38 | 2.00 |
> | VLM | PIA | 71.79 | 76.76 | 10.82 | 66.00 | 69.70 | 6.60 | 61.00 | 64.05 | 5.00 |
> | VLM | PFAMI | 74.43 | 81.25 | 6.01 | 80.40 | 87.50 | 29.40 | 76.90 | 84.99 | 24.80 |
> | VLM | CLiD | 77.55 | 83.43 | 19.23 | 80.90 | 86.53 | 50.80 | 79.00 | 85.16 | 40.60 |
> | VLM | **ReDiffuse** | 52.16 | 49.75 | 0.72 | 55.30 | 53.37 | 1.60 | 51.30 | 48.73 | 0.20 |
> | Emb. | Ours | 94.48 | 97.30 | 50.48 | 88.00 | 94.17 | 47.00 | 86.00 | 91.32 | 53.20 |
>
> Table 1\. Evaluation of the black-box MIA method ReDiffuse \[1\]. The results extend Table 2 from the main paper.
>
> We thank the reviewer for bringing attention to the recent black-box MIA method, ReDiffuse \[1\]. We first want to clarify that while ReDiffuse is black-box with respect to the model’s architecture and parameters, it still ***assumes access to ground-truth (GT) captions***. Moreover, as shown in Table 1, ReDiffuse shows limited applicability in practical cases where members and hold-outs originate from the same distribution (see **Comment A** for details).
>
> We also agree that the caption-free setting has been considered in prior works. However, these methods – including ReDiffuse – rely solely on VLM-generated captions to circumvent the absence of GT captions, which leads to ***suboptimal performance*** (Table 1). In contrast, our approach bypasses VLM dependency entirely, offering a new direction for membership inference without captions.
>
> We will revise our contribution in the main paper and more carefully position our work as ***an alternative approach to VLM-generated captions in the caption-free setting***.
>
> ---
>
> ### **| W2. Early stopping to reduce runtime**
>
> We thank the reviewer for highlighting runtime concerns in our two-stage optimization. As noted in **Comment B**, we report optimizer-specific VRAM usage and runtime. We will further quantify the efficiency–performance trade-off using *early stopping* in a future comment.
>
> ---
>
> \[1\] Li, Jingwei, et al. "Towards black-box membership inference attack for diffusion models." arXiv preprint arXiv:2405.20771 (2024).

---

> ### Author Response · Authors · 2025-11-22
>
> ## **Overview of All Reviewer Comments (2/2)**
>
> ### **| W3. Evaluation on Stable Diffusion v2 and v3**
>
> |  |  | (a) SD v2.1 |  |  | (b) SD v3 |  |  |
> | ----- | ----- | :---: | ----- | ----- | :---: | ----- | ----- |
> | **Condition** | **Method** | ASR | AUC | T@1F | ASR | AUC | T@1F |
> | GT | CLiD | 58 | 55.82 | 2.00 | 67.50 | 71.64 | 5.00 |
> | VLM | Loss | 55.50 | 52.47 | 1.00 | 53.00 | 46.27 | 0.00 |
> | VLM | SecMI | 57.50 | 54.37 | 0.00 | 62.50 | 65.06 | 4.00 |
> | VLM | PIA | 59.00 | 57.53 | 0.00 | 55.00 | 50.84 | 0.00 |
> | VLM | PFAMI | 51.5 | 39.63 | 1.00 | 68.50 | 69.81 | 3.00 |
> | VLM | CLiD | 53.5 | 51.90 | 1.00 | 67.50 | 71.59 | 4.00 |
> | $\\phi^\*$ | **Ours** | 61.34 | 58.99 | 0.00 | 70.00 | 73.42 | 2.00 |
>
> Table 2\. Evaluation against two settings of SD v2.1 (LAION-mi train vs. CC3M) and the SD v3 (COCO vs. CC3M).
>
> In Table 2, we evaluate MoFit on two large-scale models of different architecture from Stable Diffusion v1.5: Stable Diffusion v2.1 (SD v2.1) \[1\] of architecturally different in the text encoder and Stable Diffusion v3 \[2\] which is different in the noise predictor. Across all models, ***MoFit consistently outperforms VLM-captioned baselines across different models***, indicating robustness across different architecture designs. Please refer to **Comment C** for details.
>
> ---
>
> ### **| W4. Lack of failure case analysis**
>
> |  |  |  | LoRA |  |
> | ----- | ----- | ----- | :---: | ----- |
> | **Condition** | **Method** | ASR | AUC | T@1F |
> | GT | CLiD | 59.00 | 52.05 | 1.00 |
> | VLM | Loss | 54.50 | 49.26 | 0.00 |
> | VLM | SecMI | 58.50 | 53.50 | 1.00 |
> | VLM | PIA | 58.50 | 53.50 | 0.00 |
> | VLM | PFAMI | 73.50 | 77.50 | 1.00 |
> | VLM | CLiD | 59.00 | 53.11 | 1.00 |
> | **$\\phi^\*$** | **Ours** | 58.50 | 54.35 | 0.00 |
>
> Table 3\. Evaluation against LoRA-adapted LDM fine-tuned with the Pokemon dataset.
>
> We thank the reviewer for highlighting the importance of analyzing failure cases. As an initial step, we explored a potential defensive strategy against MoFit. Interestingly, as shown in Table 3, the effectiveness of MoFit ***substantially degrades when LDMs are trained with LoRA***.
>
> We further analyzed MoFit’s extreme cases by examining *color diversity* and *image complexity* across TP, FN, FP, and TN samples. Notably, samples predicted as members (TP and FP) tend to exhibit higher color diversity and greater structural complexity compared to their hold-out counterparts (FN and TN, respectively).
>
> Please refer to **Comment D-E** for experimental results and details.
>
> ---
>
> ### **| Q1. Clarification of Eq.(5)**
>
> We appreciate the reviewer’s concern regarding the clarity of Eq.(5). To address this, we will revise the manuscript to explicitly state that $z’\_t$ *denotes the latent representation of the perturbed input* $x\_0 \+ \\delta$ *at timestep* $t$.
>
> ---
>
> ### **| Q2. Lack of information on the model used for fine-tuning**
>
> We thank the reviewer for pointing this out. Unless otherwise stated, all fine-tuning experiments in the main paper are conducted using Stable Diffusion v1.4 (SD v1.4) as the base model. We will revise the manuscript to clearly specify this in the setup description (Section 4.1).
>
> ---
>
> ### **| Notes on Implementation Details and Revisions**
>
> We clarify that the optimization previously referred to as PGD was actually implemented as standard Gradient Descent (GD), with all reported results unaffected. Additionally, we extend the perturbation-level analysis in Section 5.5, further confirming MoFit’s superiority over alternative surrogate types. Please refer to **Comment F** for details.
>
> ---
>
> \[1\] [https://github.com/Stability-AI/stablediffusion](https://github.com/Stability-AI/stablediffusion)
> \[2\] [https://huggingface.co/stabilityai/stable-diffusion-3-medium-diffusers](https://huggingface.co/stabilityai/stable-diffusion-3-medium-diffusers)

---

> ### Author Response · Authors · 2025-11-22
>
> ## **Comment A \- W1: Difference between ReDiffusion and MoFit**
>
> ### **(1) ReDiffuse**
>
> We appreciate the reviewers for pointing out recent studies on membership inference against text-to-image diffusion models. ReDiffuse \[1\] proposes a black-box attack that infers membership based on the reconstruction error between a query image and multiple generated samples.
>
> While ReDiffuse is black-box with respect to model architecture and parameters, ***it assumes access to ground-truth (GT) captions during inference***. In more realistic settings without GT captions, it still relies on VLM-generated captions – similar to prior baselines.
>
> Additionally, while recent studies \[2, 3\] highlight the importance of evaluating in settings where members and hold-outs are drawn from the same distribution, ReDiffuse adopts a less challenging protocol where the two groups differ in distribution, making them easier to distinguish.
>
> We evaluated ReDiffuse under our more challenging, distributionally aligned setting. As shown in Table 1, its performance drops significantly under both GT and VLM-generated captions. This suggests that while ReDiffuse is effective when member and hold-out distributions are disjoint, ***its ability to infer membership deteriorates in more realistic scenarios where the distributions are matched***.
>
> ---
>
> ### **(2) Novelty of MoFit**
>
> We would like to first clarify the positioning of our work in relation to prior MIA studies. Among the baselines evaluated in our study, SecM , PIA, and CLiD consider the caption-free scenario, where ground-truth (GT) captions are unavailable during inference. However, we emphasize that ***these methods – including ReDiffuse – typically address this by leveraging captions generated by vision-language models (VLMs)***, which, as shown in Tables 2 and 3 in the main paper, results in significant performance degradation compared to GT captions.
>
> Given the notable performance degradation observed with VLM-generated captions, ***our work explores an alternative approach for membership inference in the caption-free setting***. Specifically, motivated by our observation that member samples exhibit greater sensitivity to condition changes than hold-outs, we design a surrogate-driven method that does not rely on VLM-generated captions. To the best of our knowledge, **t*his work is the first work to improve performance in the caption-free setting*** by exploring alternatives to the commonly adopted – but often suboptimal – VLM-generated captions.
>
> We thank the reviewer again for raising the question about novelty. We will revise the paper to more clearly position our contribution as ***a novel alternative to VLM-generated captions in the caption-free setting***.
>
> ---
>
> \[1\] Li, Jingwei, et al. "Towards black-box membership inference attack for diffusion models." arXiv preprint arXiv:2405.20771 (2024).
> \[2\] Zhai, Shengfang, et al. "Membership inference on text-to-image diffusion models via conditional likelihood discrepancy." Advances in Neural Information Processing Systems 37 (2024): 74122-74146.
> \[3\] Dubiński, Jan, et al. "Towards more realistic membership inference attacks on large diffusion models." Proceedings of the IEEE/CVF Winter Conference on Applications of Computer Vision. 2024\.

---

> ### Author Response · Authors · 2025-11-22
>
> ## **Comment B \- W2: Large-Scale Applicability**
>
> ### **(1) Computational cost and real-world applicability**
>
> We acknowledge the reviewer’s concern regarding the runtime of our two-stage optimization, which sequentially optimizes the model-fitted surrogate and its embedding. While our method incurs higher time cost, our primary goal was to significantly improve membership inference performance in the *caption-free setting* – without relying on VLM-generated captions. To the best of our knowledge, ***our work is the first to demonstrate such strong performance in this setting***, and thus we intentionally prioritized accuracy over efficiency.
>
> That said, we fully agree that cost-efficiency is critical. We thus report VRAM usage and runtime when using more lightweight optimizers in the section below. *We will further quantify the efficiency–performance trade-off using early stopping in a future comment*.
>
> ---
>
> ### **(2) Lightweight optimizers**
>
> To address concerns about computational costs in our two-stage pipeline – (i) model-fitted surrogate alignment and (ii) embedding extraction – we perform ablation studies comparing lightweight optimizers (SGD, RMSProp, and LION \[1\]).
>
> | (a) Surrogate | VRAM (MB) | Runtime (sec) | (b) Embedding | VRAM (MB) | Runtime (sec) |
> | :---: | :---: | :---: | ----- | ----- | ----- |
> | Adam | 18053.79 | 333.24 | Adam (**Ours**) | 14799.11 | 38.28 |
> | SGD | 18052.38 | 333.19 | SGD | 14797.22 | 38.21 |
> | RMSProp | 18050.83 | 333.66 | RMSProp | 14803.43 | 38.15 |
> | LION | 18049.72 | 333.57 | LION | 14803.64 | 38.15 |
> | **Ours** | 20667.5 | 358.96 |  |  |  |
>
> Table 4\. Comparison of GPU VRAM usage and runtime overhead across optimizers during (a) surrogate optimization and (b) embedding extraction.
>
> **(i) Model-fitted surrogate optimization**
> When optimizing the surrogate, we follow prior methods \[2, 3\] and update the perturbation using a single-step, first‑order update that directly applies the sign of the loss gradient – i.e., sign(∇Lₓ) – to the input image. As evaluated in Table 4(a) for 1,000 steps (default setting), this design results in higher GPU consumption (up to 2,618 MB VRAM) and slightly longer runtime (up to 25.77 s) than optimizer-based alternatives. However, this single-step method avoids the need for multiple optimization trials of hyperparameter tuning (e.g., learning rate or momentum), which can be costly and dataset-dependent. We view this trade-off as a design choice left to the adversary, depending on their resource constraints and deployment scenario.
>
> **(ii) Surrogate-driven embedding extraction**
> For the embedding extraction step from the model-fitted surrogate, we employ the Adam optimizer by default. In Table 4(b), while the optimizers were evaluated using 200 optimization steps (default setting for SD v1.4 fine-tuned with Pokemon dataset), we find that all optimizers perform consistently, with only a subtle difference. Given its stability and wide applicability across diverse tasks, we find Adam to be a practical default.
>
> ---
>
> \[1\] Chen, Xiangning, et al. "Symbolic discovery of optimization algorithms." Advances in neural information processing systems 36 (2023): 49205-49233.
> \[2\] Xue, Haotian et al. “Toward effective protection against diffusion based mimicry through score distillation.” ArXiv abs/2311.12832 (2023).
> \[3\] Salman, Hadi, et al. "Raising the cost of malicious ai-powered image editing." arXiv preprint arXiv:2302.06588 (2023).

---

> ### Author Response · Authors · 2025-11-22
>
> ## **Comment C \- W3: Evaluation on Stable Diffusion v2 and v3**
>
> We appreciate the reviewer’s insightful suggestion to evaluate MoFit on a broader architecture of large-scale diffusion models beyond SD v1.5. In response, we extended our experiments to: (1) Stable Diffusion v2.1, which differs in the text encoder with SD v1 (switch from CLIP to OpenCLIP \[1\]) and (2) Stable Diffusion v3, which adopts a different architecture based on Transformers instead of U-Net.
>
> ---
>
> ### **(1) Evaluation on Stable Diffusion v2.1 (SD v2.1)**
>
> **Defining Member/Hold-out Splits for SDv2.1.** Unlike SDv1.5, which allows controlled evaluation via the curated LAION-mi split \[2\], evaluating membership inference on SDv2.1 \[3\] presents a unique challenge. As detailed in \[3\], SDv2.1 is trained on LAION-5B, a superset that contains both the member and hold-out images of LAION-mi. Thus, LAION-mi cannot be reused directly for testing generalization performance on SDv2.1.
>
> | LAION-mi member vs. | COCO | Flickr | CC3M | LAION-mi hold-out |
> | :---: | :---: | :---: | :---: | :---: |
> | FID | 53.6041 | 61.8270 | 16.0582 | 8.8673 |
>
> Table 5\. FID scores between LAION‑mi member set and several datasets
>
> To address this challenge, we construct a distributionally aligned evaluation split, following best practices in \[2\] that recommend closely matched real-world distributions for member and hold-out samples. We compute FID scores between the LAION-mi member set and candidate datasets (COCO, Flickr, CC3M) in Table 5, and find that CC3M yields the lowest FID (16.06), closely matching the intra‑LAION-mi FID (8.87). Accordingly, we use LAION‑mi members as the member set and CC3M as the hold-out set for SDv2.1 evaluation.
>
> **Evaluation on SDv2.1 using real-world split.** We perform membership inference attacks using 100 images from each split and evaluate MoFit alongside all baselines. In Table 2(a), under VLM‑generated captions – a realistic setting where the adversary lacks GT annotations – all baselines exhibit notable drops in ASR and AUC. In contrast, ***MoFit consistently outperforms them, achieving a \+2.34% ASR gain over the second-best method*** under these restricted conditions.
>
> ---
>
> ### **(2) Evaluation on Stable Diffusion v3 (SD v3)**
>
> **Defining Member/Hold-out Splits for SDv3.** We further evaluated MoFit on Stable Diffusion v3 (SDv3) to test its generalization to a different model architecture—specifically, from U-Net-based LDMs (e.g., SDv1.5, SDv2.1) to the Multimodal Diffusion Transformer (MMDiT) \[5\]. However, SDv3 provides no publicly available details about its training dataset aside from the number of training images \[6\], making it challenging to construct reliable member/hold-out splits.
>
> Empirically, we observed that COCO samples yield significantly lower conditional and unconditional loss values (Eq. 1 and 2 in the main paper) compared to LAION-mi or CC3M. Based on this, we selected COCO as the member set and CC3M as the hold-out set for evaluating MoFit on SDv3. (We will include a supporting figure visualizing the loss gap across datasets in the Appendix.)
>
> **Evaluation on Pre-trained SDv3.**  We conducted membership inference using 100 images from each split, running all methods at a fixed diffusion timestep (t \= 328.1250, i.e., step 140). As shown in Table 2(b), ***MoFit outperforms all baselines, including CLiD under ground-truth conditions.*** These results further confirm that MoFit can match or even exceed methods that rely on full image-caption supervision, consistent with our findings on SDv1.5 (see Table 2 of the main paper for MS-COCO).
>
> ---
>
> \[1\] Cherti, Mehdi, et al. "Reproducible scaling laws for contrastive language-image learning." Proceedings of the IEEE/CVF conference on computer vision and pattern recognition. 2023\.
> \[2\] Dubinski et al., Towards More Realistic Membership Inference Attacks on Large Diffusion Models, WACV 24\.
> \[3\] [https://github.com/Stability-AI/stablediffusion](https://github.com/Stability-AI/stablediffusion)
> \[4\] Webster et al., A reproducible extraction of training images from diffusion models, arxiv 23\.
> \[5\] Esser, Patrick, et al. "Scaling rectified flow transformers for high-resolution image synthesis." Forty-first international conference on machine learning. 2024\.
> \[6\] [https://huggingface.co/stabilityai/stable-diffusion-3-medium-diffusers](https://huggingface.co/stabilityai/stable-diffusion-3-medium-diffusers)

---

> ### Author Response · Authors · 2025-11-22
>
> ## **Comment D \- W4: Lack of failure case analysis (1/2)**
>
> We thank the reviewer for highlighting the importance of analyzing failure cases. In response, we now provide an extended analysis covering: (1) the underperformance of both MoFit and prior methods on LoRA-adapted LDMs, and (2) extreme cases of members and hold-outs under our method. We will include the full analysis and supporting tables in the revised manuscript.
>
> ---
>
> ### **(1) LoRA-adapted LDMs**
>
> We find that ***Low‑Rank Adaptation (LoRA) substantially reduces the effectiveness of MoFit.*** Specifically, replacing full U‑Net fine‑tuning with LoRA leads to a noticeable performance drop for our method.
>
> **Studies related to robustness of LoRA against MIA.** Recent studies in language modeling have evaluated membership inference under LoRA fine-tuning. While early works \[1\] suggest robustness, later studies \[2,3\] show that LoRA’s limited parameter updates significantly reduce attack effectiveness. For latent diffusion models, Luo et al. \[4\] suggest that LoRA-adapted LDMs may remain vulnerable, but their evaluation is limited to earlier MIA methods \[5,6\] (e.g., the “Loss” method in our paper), and does not cover more recent approaches.
>
> **Our experiment against LoRA.** In contrast, Table 3 shows that when evaluated on 100 samples of Pokemon training set used to fine-tune the LoRA-adapted SDv1.4 \[7\], ***performance drops to near-random levels for both MoFit and most of the baselines under ground-truth and VLM-generated captions***. We hypothesize that this robustness stems from LoRA’s minimal footprint, which preserves most of the original weights and reduces the model’s capacity to remember specific training samples \[2\].
>
> **Reason behind the robustness of PFAMI \[8\].** Baselines other than PFAMI rely on *cross-query* comparisons—typically based on the relative ranking of diffusion losses across different queries. The poor performance of these baselines under LoRA, as shown in Table 3, suggests that the loss values of member and non-member queries become increasingly similar. This makes it difficult for cross-query methods to distinguish between them based on relative loss ordering.
>
> However, PFAMI uses *within-query* scores by applying multiple augmentations to a single query image and computing the loss for each. The membership score is based on the relative variation within these losses. Since it does not rely on comparisons across different queries, PFAMI remains robust even when LoRA causes the overall loss values of member and non-member samples to become similar. The relative differences within a single query are preserved, making it less sensitive to such global shifts.
>
> ---
>
> \[1\] Wen, Rui, et al. "Last one standing: A comparative analysis of security and privacy of soft prompt tuning, lora, and in-context learning." arXiv preprint arXiv:2310.11397 (2023).
> \[2\] Amit, Guy, Abigail Goldsteen, and Ariel Farkash. "SoK: reducing the vulnerability of fine-tuned language models to membership inference attacks." arXiv preprint arXiv:2403.08481 (2024).
> \[3\] Liu, Ruixuan, et al. "Precurious: How innocent pre-trained language models turn into privacy traps”, arxiv 2024\.
> \[4\] Luo, Zihao, et al. "Privacy-Preserving Low-Rank Adaptation Against Membership Inference Attacks for Latent Diffusion Models." Proceedings of the AAAI Conference on Artificial Intelligence. Vol. 39\. No. 6\. 2025\.
> \[5\] Hu, Hailong, and Jun Pang. "Membership inference of diffusion models." arXiv preprint arXiv:2301.09956 (2023).
> \[6\] Matsumoto, Tomoya, Takayuki Miura, and Naoto Yanai. "Membership inference attacks against diffusion models." 2023 IEEE Security and Privacy Workshops (SPW). IEEE, 2023\.
> \[7\] [https://huggingface.co/sr5434/sd-pokemon-model-lora](https://huggingface.co/sr5434/sd-pokemon-model-lora)
> \[8\] Fu, Wenjie, et al. "A probabilistic fluctuation based membership inference attack for diffusion models." arXiv preprint arXiv:2308.12143 (2023).

---

> ### Author Response · Authors · 2025-11-22
>
> ## **Comment E \- W4: Lack of failure case analysis (2/2)**
>
> ### **(2) Analysis on extreme cases of members and hold-outs**
>
> | Dataset | MS-COCO | MS-COCO | MS-COCO | MS-COCO | Flickr | Flickr | Flickr | Flickr |
> | ----- | :---: | :---: | :---: | :---: | :---: | :---: | :---: | :---: |
> |  | TP | FN | FP | TN | TP | FN | FP | TN |
> | Colorfulness Score | 135.27 | 122.65 | 147.64 | 132.54 | 146.22 | 138.56 | 147.51 | 147.95 |
> | Entropy | 15.22 | 13.70 | 15.11 | 14.27 | 15.55 | 15.05 | 14.92 | 15.33 |
> | Keypoint Count | 500.00 | 471.95 | 499.25 | 492.20 | 500.00 | 494.55 | 499.65 | 497.75 |
>
> Table 6\. TP, FN, FP, and TN statistics using color diversity and image complexity metrics for two fine-tuned SD 1.4 models.
>
> We are grateful for the reviewer’s suggestion to investigate extreme cases, which we address through the following analysis. Using the two fine-tuned SD 1.4 models (MS-COCO and Flickr), we selected the top-20 samples from each of the following groups:
>
> - **TP** (True Positives): members predicted as members
> - **FN** (False Negatives): members predicted as hold-outs
> - **FP** (False Positives): hold-outs predicted as members
> - **TN** (True Negatives): hold-outs predicted as hold-outs
>
> To quantify color diversity, we used the colorfulness score \[1\], and to assess image complexity, we employed Shannon entropy \[2\] and ORB keypoint count \[3\]. Higher colorfulness indicates richer chromatic variation, while higher entropy and keypoint counts reflect greater structural complexity.
>
> Interestingly, as shown in Table 6, we observed that samples predicted as members (TP and FP) consistently exhibit higher values across all metrics compared to their hold-out counterparts (FN and TN). This suggests that the ***model is more likely to associate colorful and structurally rich images with training membership***. We believe this finding may offer an interesting direction for future work on understanding what types of images diffusion models are more likely to retain.
>
> ---
>
> \[1\] Hasler, David, and Sabine E. Suesstrunk. "Measuring colorfulness in natural images." Human vision and electronic imaging VIII. Vol. 5007\. SPIE, 2003\.
> \[2\] Shannon, Claude E. "A mathematical theory of communication." The Bell system technical journal 27.3 (1948): 379-423.
> \[3\] Rublee, Ethan, et al. "ORB: An efficient alternative to SIFT or SURF." 2011 International conference on computer vision. Ieee, 2011\.

---

> ### Author Response · Authors · 2025-11-22
>
> ## **Comment F: Clarification and Implementation Adjustment**
>
> ### **(1) Section 4.1 & 5**
>
> We would like to clarify that the optimization referred to as PGD in Section 4.1 was implemented as standard Gradient Descent (GD) without the projection (clipping) step. While we initially described the optimization as PGD, we acknowledge this mismatch in terminology and will revise the manuscript to reflect the exact implementation.
>
> Importantly, the numerical results throughout the paper remain unchanged. In practice, all reported experiments were conducted using GD with an initial noise level $\\varepsilon$ equivalent to the previously stated noise bound ($\\eta \= 0.3$). The step size $\\alpha$ is initialized to 0.15 and decays proportionally with the iteration count. Please refer to Eq. (10) in Appendix A.1 for the update equation.
>
> ---
>
> ### **(2) Section 5.5**
>
> In Section 5.5, we extend our evaluation of configuration (ii), where a random noise perturbation $\\delta$ is added to the query image. While the main paper reports results at a single perturbation level ($\\varepsilon \= 0.3$), we now investigate a broader range of noise levels, varying $\\varepsilon$ from 0.1 to 0.9. For each dataset, we identify the noise level that yields the strongest defensive effect and report the corresponding results in the second row of Table 4\.
> Notably, MoFit continues to exhibit strong and stable performance across all datasets, even under the best-performing perturbation level for each dataset. This demonstrates that our model-fitted surrogate $x^\*$ establishes a more robust and discriminative coupling with its optimized embedding than perturbed inputs such as $x \+ \\delta$ and other variants (*e.g.*, $x$ or $x \+ \\delta\_{\\text{max}}$).
>
> ---
>
> ### **(3) Figure 5**
>
> As our optimization uses GD without projection, we replace the noise bound (η) with the initial step size, which decays proportionally with iteration. Note that the experimental results remain unchanged; only the x-axis label is updated to reflect the corrected interpretation.

---

> ### Author Response · Authors · 2025-11-25
>
> ## **Additional Comment (1) \- W2: Early Stopping for Efficiency Improvement**
>
> | (a) Surrogate |  |  |  |  | (b) Embedding |  |  |  |  |  |  |  |  |  |
> | ----- | ----- | ----- | ----- | ----- | ----- | ----- | ----- | ----- | ----- | ----- | ----- | ----- | ----- | ----- |
> | **Threshold** | ASR | AUC | T@1F | Runtime (sec) | **Threshold** | ASR | AUC | T@1F | Runtime (sec) | **Threshold** | ASR | AUC | T@1F | Runtime (sec) |
> | 0.11 | 81.50 | 88.53 | 32.00 | 10.71 | 0.01 | 80.00 | 86.95 | 37.00 | 24.65 | 0.006 | 85.00 | 91.57 | 37.00 | 48.07 |
> | 0.10 | 82.50 | 88.99 | 25.00 | 12.80 | 0.009 | 81.00 | 88.17 | 35.00 | 29.11 | 0.005 | 86.00 | 92.97 | 43.00 | 58.19 |
> | 0.09 | 82.00 | 89.67 | 39.00 | 16.33 | 0.008 | 82.00 | 89.62 | 33.00 | 34.37 | 0.004 | 86.50 | 92.37 | 46.00 | 72.83 |
> | 0.08 | 84.00 | 90.15 | 42.00 | 22.32 | 0.007 | 84.00 | 90.50 | 31.00 | 40.45 | 0.003 | 85.79 | 91.77 | 47.96 | 88.81 |
> | 0.07 | 85.00 | 90.68 | 49.00 | 33.62 |  |  |  |  |  |  |  |  | Total Runtime | 116.38 |
> | 0.06 | 86.50 | 91.00 | 39.00 | 55.41 |  |  |  |  |  |  |  |  |  |  |
> |  |  |  | Total Runtime | 358.96 |  |  |  |  |  |  |  |  |  |  |
>
> Table 7\. Performance of MoFit under an early stopping regime applied to both surrogate optimization and embedding extraction.
>
> To investigate the reviewer’s insight regarding the importance of improving optimization efficiency, we conduct ***early stopping*** experiments that terminate the optimization process once a predefined loss threshold is reached. We apply this strategy to both (i) model-fitted surrogate optimization and (ii) surrogate-driven embedding extraction, using the SDv1.4 model fine-tuned on the MS-COCO dataset.
>
> **(i) Model-fitted surrogate optimization**
> We first average the final loss (Eq. (5)) from full optimization runs, which yield 0.02446 for members and 0.02259 for hold-outs. We then re-run the optimization and terminate early when the loss reaches a higher predefined threshold, choosing from {0.11, 0.1, 0.09, 0.08, 0.07, 0.06}. When a threshold is met, we save the corresponding surrogate and proceed to extract the embedding using the same setup as in the main experiments (300 iterations for MS-COCO, per Appendix A.1).
>
> In Table 7(a), we report membership inference results and average GPU runtime when applying early stopping to the surrogate optimization stage of MoFit. We evaluate on 100 member and 100 hold-out images from the MS-COCO dataset, using an NVIDIA RTX 4090\.
>
> As expected, performance improves as the loss threshold decreases, at the cost of increased runtime – demonstrating a clear trade-off between attack performance and computational efficiency.
>
> Notably, when compared to our baseline CLiD (83.50% ASR and 87.37 AUC in the same setting), MoFit reaches comparable or superior performance even when stopped early – for instance, at a loss threshold of 0.08, ***it achieves competitive results to CLiD while only taking 22.32 seconds per image*** (originally takes 358.96 sec). This suggests that an adversary can strategically balance efficiency and effectiveness by choosing an appropriate early stopping criterion.
>
> **(ii) Surrogate-driven embedding extraction**
> We follow the same procedure and compute the average final loss of Eq. (6) after embedding extraction, obtaining values of 0.00232 for members and 0.00229 for hold-outs. We then re-run full surrogate optimization (1,000 steps as default) for all samples, extract embeddings, and apply early stopping when the loss in Eq. (6) reaches a preset threshold: {0.01, 0.009, 0.008, 0.007, 0.006, 0.005, 0.004, 0.003}.
>
> As shown in Table 7(b), we again observe a trade-off between runtime and attack performance, mirroring the pattern in (i). Notably, MoFit reaches comparable performance to CLiD at a threshold of 0.007, while ***reducing runtime to 75.93 seconds per image***. This demonstrates that early-stopping can be effectively applied at both optimization and embedding stages to flexibly adjust the cost-performance balance of MoFit.

---

> > ### Comment · Reviewer_edeD · 2025-11-26
> >
> > Thanks for the thorough analysis and additional explanations and experiments. In general, as you already did, you can plug your changes and revisions inside the pdf itself to make sure it fits the length and level of writing. Currently, many of your explanations are in the OpenReview chat, and not in the pdf itself. I encourage you to resubmit the pdf with all of your additional experiments and explanations, to make sure everything is on place.
> >
> > I still think that the timing is very high, but as long as you conducted ablation on it and it can be further optimized in future work, it is good as starting point. The LoRa issue seems very interesting, so I think it is well appropriated to be in the main paper. When I pointed on failure cases, I mainly intended to interesting examples that illustrating something useful. In many cases large statistics help in get the large picture, and examples are good in pointing specific cases in a more understandable manner.

---

> ### Author Response · Authors · 2025-11-27
>
> We sincerely appreciate your acknowledgement of our discussions and contributions. As requested, we updated the revised version of our manuscript, and the main points discussed above can be found in the following sections, marked in blue for clarity:
>
> - LoRA – Section 5.6.2, Appendix A.7
> - Optimizer-specific VRAM usage and runtime – Appendix A.8.1
> - Early Stopping – Section 5.6.3, Appendix A.8.2
> - SD v1.5, SD v2, SD v3 – Appendix A.9
> - Failure cases – Appendix A.13
>
> We hope that our revisions have addressed your concerns, and we kindly request that you consider reflecting these changes in your updated review scores.
>
> ---
>
> ### **Visualization of extreme samples supporting Comment E**
>
> We are pleased that the additional experiments were found to be insightful. To further support Comment E and address the reviewer’s suggestion to include individual examples, we have added Fig. 13(a) in the revised manuscript, which visualizes representative extreme cases (TP, FN, FP, TN) of MS-COCO from our MoFit setup.
>
> Among *member* samples, **FN images tend to exhibit low visual complexity and monotonous color schemes** (*e.g.*, plain black-and-white scenes or sky/ocean backgrounds), whereas TP samples show vibrant colors (*e.g.*, green hues) and diverse semantic components (e.g., kitchen items, road scenes).
>
> A similar pattern is observed in *hold-out* samples: **FP images often include vivid colors** (e.g., green, red) **and rich details** (e.g., reflection of a tower, food on the table), while TNs are visually simpler (e.g., snowy landscapes, minimal objects).
>
> Focusing solely on image content regardless of the split, we find that samples with green grass backgrounds or crowded urban scenes (colorful and complex) are more likely to be retained by the model, whereas grayscale images or landscapes of desert and snow (relatively plain) are less likely to be retained.
>
> ### **Consistency under GT caption setting**
>
> We additionally include Fig. 13(b) to visualize representative extreme cases under the presence of GT captions, using samples extracted from the GT-captioned CLiD setup. While the extreme samples are not fully identical with Fig. 13(a), the same trend holds: FNs are visually plain, whereas FPs exhibit greater colorfulness and complexity. This consistency suggests that the observed behaviors of FPs and FNs may reflect a general property of latent diffusion models, offering a promising direction for future work on understanding which types of images are more likely to be retained or less trained in the model.

---

### Official Review · Reviewer_8oRv · 2025-10-31

**Soundness:** 2
**Presentation:** 3
**Contribution:** 2
**Rating:** 4
**Confidence:** 3

**Summary:**

This paper proposes a new framework, called MOFIT, for performing membership inference attacks on latent diffusion models without ground-truth captions. The paper first studies the differences in how member and non-member samples respond to mismatched conditioning during the denoising process. Based on this observation, the authors propose an efficient method to improve MIA in the caption-free setting.

**Strengths:**

- The problem being studied is practical. Privacy risks in generative diffusion models are an important topic. The work addresses a realistic limitation in existing MIAs by removing the dependency on ground-truth captions, making it applicable to real-world scenarios.

- Strong empirical performance: The results show consistent improvements over VLM-substituted baselines across multiple datasets and models.

- Comprehensive analysis: The paper provides detailed analysis and discussion of different components.

**Weaknesses:**

- Despite arguing that the proposed method has a more practical setting, it still relies on internal model signals (e.g., likelihoods or denoising trajectories required by CLiD) that may not translate to black-box APIs. While common in MIA research, this reduces practicality in real-world applications.


- The experiments focus on fine-tuned models on relatively small datasets (e.g., Pokémon with ~800 images), so generalization to large-scale, pre-trained models is unclear. It is important to conduct experiments on open-source pre-trained models with publicly available training sets. I am curious whether the proposed method still works on these models.


- It is also important to provide experiment on computational overhead. Surrogate optimization in high-dimensional conditioning spaces can be expensive and resource-intensive, especially for large models. Reporting runtime and scaling behavior would help assess practicality.


- Stability. The method relies on fixed noise bounds and timesteps for optimization, which might be sensitive to hyperparameters. The paper mentions using VLM initialization for stability but lacks ablation studies on failure cases or robustness to noise variations.

**Questions:**

Please see the weaknesses section

---

> ### Author Response · Authors · 2025-11-22
>
> ## **Overview of All Reviewer Comments (1/2)**
>
> To thoroughly address the reviewers’ questions and concerns, we conducted a comprehensive set of additional experiments. For clarity and convenience, we summarize all relevant tables below. We kindly invite you to review this summary and refer to the corresponding comments (Comment A to E) for detailed explanations.
> **Note.** T@1F in the table refers to  the TPR@1%FPR metric.
>
> ---
>
> ### **| W1. Limitation to black-box APIs**
>
> |  |  | Pokemon |  |  | MS-COCO |  |  | Flickr |  |  |
> | ----- | ----- | :---: | ----- | ----- | :---: | ----- | ----- | :---: | ----- | ----- |
> |  |  | ASR | AUC | T@1F | ASR | AUC | T@1F | ASR | AUC | T@1F |
> | GT | CLiD | 96.52 | 99.17 | 90.14 | 86.50 | 90.27 | 68.80 | 91.10 | 95.13 | 77.20 |
> | GT | **ReDiffuse** | 52.40 | 49.53 | 0.72 | 54.20 | 52.41 | 1.20 | 60.00 | 48.23 | 0.20 |
> | VLM | Loss | 72.27 | 78.99 | 4.81 | 63.70 | 67.88 | 4.80 | 61.60 | 64.24 | 5.40 |
> | VLM | SecMI | 78.51 | 6.22 | 6.97 | 57.30 | 58.07 | 4.20 | 54.00 | 52.38 | 2.00 |
> | VLM | PIA | 71.79 | 76.76 | 10.82 | 66.00 | 69.70 | 6.60 | 61.00 | 64.05 | 5.00 |
> | VLM | PFAMI | 74.43 | 81.25 | 6.01 | 80.40 | 87.50 | 29.40 | 76.90 | 84.99 | 24.80 |
> | VLM | CLiD | 77.55 | 83.43 | 19.23 | 80.90 | 86.53 | 50.80 | 79.00 | 85.16 | 40.60 |
> | VLM | **ReDiffuse** | 52.16 | 49.75 | 0.72 | 55.30 | 53.37 | 1.60 | 51.30 | 48.73 | 0.20 |
> | Emb. | Ours | 94.48 | 97.30 | 50.48 | 88.00 | 94.17 | 47.00 | 86.00 | 91.32 | 53.20 |
>
> Table 1\. Evaluation of the black-box MIA method ReDiffuse \[1\]. The results extend Table 2 from the main paper.
>
> We fully agree with the reviewer that methods relying on internal model scores – including MoFit – may encounter deployment challenges in black-box API settings. While a recent black-box MIA method, ReDiffuse, has been proposed, we emphasize that ***our work is the first to provide a viable alternative to the suboptimal performance of VLM-generated captions in the caption-free setting***. Moreover, our additional results in Table 1 show that ***ReDiffuse performs poorly in both the practical GT-caption scenario and the caption-free regime***, suggesting that progress in the white-box setting is a necessary first step before tackling fully black-box scenarios. We provide further details in **Comment A**, and we hope this helps convey the motivation and scope of our work.
>
> ---
>
> ### **| W2. Evaluation against large-scale pre-trained models**
>
> |  |  | (a) SD v1.5 |  |  | (b) SD v2.1 |  |  | (c) SD v3 |  |  |
> | ----- | ----- | :---: | ----- | ----- | :---: | ----- | ----- | :---: | ----- | ----- |
> | **Condition** | **Method** | ASR | AUC | T@1F | ASR | AUC | T@1F | ASR | AUC | T@1F |
> | GT | CLiD \+ GT | 60.00 | 58.13 | 1.00 | 58.00 | 55.82 | 2.00 | 67.50 | 71.64 | 5.00 |
> | VLM | Loss | 52.00 | 50.48 | 0.60 | 55.50 | 52.47 | 1.00 | 53.00 | 46.27 | 0.00 |
> | VLM | SecMI | 52.50 | 50.28 | 2.00 | 57.50 | 54.37 | 0.00 | 62.50 | 65.06 | 4.00 |
> | VLM | PIA | 51.50 | 49.75 | 1.60 | 59.00 | 57.53 | 0.00 | 55.00 | 50.84 | 0.00 |
> | VLM | PFAMI | 54.00 | 52.58 | 0.75 | 51.50 | 39.63 | 1.00 | 68.50 | 69.81 | 3.00 |
> | VLM | CLiD | 56.50 | 52.78 | 1.00 | 53.50 | 51.90 | 1.00 | 67.50 | 71.59 | 4.00 |
> | $\\phi^\*$ | **Ours** | 60.00 | 58.23 | 2.00 | 61.34 | 58.99 | 0.00 | 70.00 | 73.42 | 2.00 |
>
> Table 2\. Evaluation against three large-scale pre-trained models: SD v1.5 (LAION-mi train vs. test), SD v2.1 (LAION-mi train vs. CC3M), and  SD v3 (COCO vs. CC3M).
>
> In **Table 2**, we evaluate MoFit on Stable Diffusion v1.5 (SD v1.5) \[2\] using the challenging LAION-mi split \[3\]. We further extend our analysis to Stable Diffusion v2.1 (SD v2.1) \[4\] and the architecturally distinct SD v3 \[5\]. Across all models, ***MoFit consistently outperforms VLM-captioned baselines under the publicly available dataset splits*** and remains competitive with CLiD under GT captions. Please refer to **Comment B** and **C** for details.
>
> ---
>
> \[1\] Li, Jingwei, et al. "Towards black-box membership inference attack for diffusion models." arXiv preprint arXiv:2405.20771 (2024).
> \[2\] [https://huggingface.co/stable-diffusion-v1-5/stable-diffusion-v1-5](https://huggingface.co/stable-diffusion-v1-5/stable-diffusion-v1-5)
> \[3\] Dubiński, Jan, et al. "Towards more realistic membership inference attacks on large diffusion models." Proceedings of the IEEE/CVF Winter Conference on Applications of Computer Vision. 2024\.
> \[4\] [https://github.com/Stability-AI/stablediffusion](https://github.com/Stability-AI/stablediffusion)
> \[5\] [https://huggingface.co/stabilityai/stable-diffusion-3-medium-diffusers](https://huggingface.co/stabilityai/stable-diffusion-3-medium-diffusers)

---

> ### Author Response · Authors · 2025-11-22
>
> ## **Overview of All Reviewer Comments (2/2)**
>
> ### **| W3. VRAM usage and runtime report & Early stopping**
>
> We thank the reviewer for highlighting runtime concerns in our two-stage optimization. As noted in **Comment D**, we report optimizer-specific VRAM usage and runtime. We will further quantify the efficiency–performance trade-off using *early stopping* in a future comment.
>
> ---
>
> ### **| W4. Stability**
>
> We sincerely thank the reviewer for raising thoughtful concerns regarding fixed hyperparameters, including the choice of timesteps. To start, we kindly note that ***Figure 5 in the main paper already includes results across diverse timestep values***. Additionally, our method does not impose any noise bounds (see Comment E), as imperceptibility is not required in our setting.
>
> |  | ASR | AUC | T@1F |
> | ----- | :---: | :---: | :---: |
> | random noise 1 | 94.50 | 96.96 | 18.00 |
> | random noise 2 | 94.00 | 96.23 | 25.00 |
> | random noise 3 | 94.00 | 95.85 | 18.00 |
> | random noise 4 | 95.50 | 96.72 | 41.00 |
>
> Table 3\. MoFit performance under four random target noise settings.
>
> To further support the reviewer’s point, we conducted an additional experiment where the target noise $\\hat{\\epsilon}$ is randomized. Consistent results in Table 3 further support the stable effectiveness of MoFit under different noise targets. Further details are discussed in **Comment E**.
>
> ---
>
> ### **| Notes on Implementation Details and Revisions**
> We clarify that the optimization previously referred to as PGD was actually implemented as standard Gradient Descent (GD), with all reported results unaffected. Additionally, we extend the perturbation-level analysis in Section 5.5, further confirming MoFit’s superiority over alternative surrogate types. Please refer to **Comment E** for details.

---

> ### Author Response · Authors · 2025-11-22
>
> ## **Comment A \- W1: Necessity for White-box MIA in Caption-Free Setting**
>
> We thank the reviewer for highlighting the practical limitations of relying on internal model signals. We agree that black-box access (*e.g.*, via API) poses deployment challenges for white-box methods.
>
> However, our goal is to tackle a critical and underexplored problem: *caption-free membership inference*, where ground-truth captions are unavailable. Prior works assume access to GT captions or VLM-generated substitutes, which significantly degrade performance (see Tables 2 and 3 in the main paper). MoFit is the first to demonstrate strong results in this more realistic setting.
>
> Importantly, we view ***white-box exploration as a necessary first step***, as done in prior diffusion MIA works (*e.g*., SecMI, CLiD), before developing black-box extensions (*e.g.*, ReDiffuse \[1\]). In this context, MoFit establishes a strong caption-free baseline under white-box conditions.
>
> ---
>
> ### **A recent membership inference studies: ReDiffuse**
>
> We appreciate the reviewers for pointing out recent studies on membership inference against text-to-image diffusion models. One such work, ReDiffuse \[4\], proposes a black-box attack that infers membership based on the reconstruction error between a query image and multiple generated samples.
>
> While ReDiffuse is black-box with respect to model architecture and parameters, ***it assumes access to ground-truth (GT) captions during inference***. In more realistic settings without GT captions, it still relies on VLM-generated captions – similar to prior baselines.
>
> Additionally, while recent studies \[2, 3\] highlight the importance of evaluating in settings where members and hold-outs are drawn from the same distribution, ReDiffuse adopts a less challenging protocol where the two groups differ in distribution, making them easier to distinguish.
>
> We evaluated ReDiffuse under our more challenging, distributionally aligned setting. As shown in Table 1, its performance drops significantly under both GT and VLM-generated captions. This suggests that while ReDiffuse is effective when member and hold-out distributions are disjoint, ***its ability to infer membership deteriorates in more realistic scenarios where the distributions are matched***.
>
> ---
>
> \[1\] Li, Jingwei, et al. "Towards black-box membership inference attack for diffusion models." arXiv preprint arXiv:2405.20771 (2024).
> \[2\] Zhai, Shengfang, et al. "Membership inference on text-to-image diffusion models via conditional likelihood discrepancy." Advances in Neural Information Processing Systems 37 (2024): 74122-74146.
> \[3\] Dubiński, Jan, et al. "Towards more realistic membership inference attacks on large diffusion models." Proceedings of the IEEE/CVF Winter Conference on Applications of Computer Vision. 2024\.

---

> ### Author Response · Authors · 2025-11-22
>
> ## **Comment B \- W2: Evaluation on Large-Scale Models (1/2)**
>
> We thank the reviewer for highlighting the importance of evaluating publicly released, large-scale pre-trained models. To this end, we conducted additional experiments on three publicly available pre-trained diffusion models with increasing scale and diversity: (1) Stable Diffusion v1.5 (SDv1.5), (2) Stable Diffusion v2.1 (SDv2.1), and (3) Stable Diffusion v3 (SDv3). These models were evaluated using only publicly available training sets and under the same caption-free setting.
>
> ---
>
> ### **(1) Evaluation of MoFit on SD v1.5**
>
> To validate the transferability of MoFit, we evaluate on SD v1.5 using the LAION-mi split \[1\], a standard member/hold-out partition. Notably, Table 3 in the main paper already reports SD v1.5 results using a curated subset of verified memorized samples \[2\] to better expose performance differences across methods.
>
> For this new experiment, we follow the LAION-mi split directly and increase the optimization timestep from $t=140$ to $t=350$, which improves MoFit's performance on large-scale models. As shown in Table 2(a), although the task remains challenging, MoFit consistently outperforms VLM-captioned baselines and even remains competitive with CLiD under GT captions. These results confirm ***MoFit’s strong transferability to large-scale public models in realistic, caption-free settings***. We will include this experiment in the final paper.
>
> ---
>
> ### **(2) Evaluation of MoFit on Stable Diffusion v2.1 (SD v2.1)**
>
> **Defining Member/Hold-out Splits for SDv2.1.** Unlike SDv1.5, which allows controlled evaluation via the curated LAION-mi split, evaluating membership inference on SDv2.1 \[3\] presents a unique challenge. As detailed in \[3\], SDv2.1 is trained on LAION-5B, a superset that contains both the member and hold-out images of LAION-mi. Thus, LAION-mi cannot be reused directly for testing generalization performance on SDv2.1.
>
> | LAION-mi member vs. | COCO | Flickr | CC3M | LAION-mi hold-out |
> | :---: | :---: | :---: | :---: | :---: |
> | FID | 53.6041 | 61.8270 | 16.0582 | 8.8673 |
>
> Table 4\. FID scores between LAION‑mi member set and several datasets
>
> To address this challenge, we construct a distributionally aligned evaluation split, following best practices in \[1\] that recommend closely matched real-world distributions for member and hold-out samples. We compute FID scores between the LAION-mi member set and candidate datasets (COCO, Flickr, CC3M) in Table 4, and find that CC3M yields the lowest FID (16.0582), closely matching the intra‑LAION-mi FID (8.8673). Accordingly, we use LAION‑mi members as the member set and CC3M as the hold-out set for SDv2.1 evaluation.
>
> **Evaluation on SDv2.1 using real-world split.** We perform membership inference attacks using 100 images from each split and evaluate MoFit alongside all baselines. In Table 2(b), under VLM‑generated captions – a realistic setting where the adversary lacks GT annotations – all baselines exhibit notable drops in ASR and AUC. In contrast, ***MoFit consistently outperforms them, achieving a \+2.34% ASR gain over the second-best method*** under these restricted conditions.
>
> ---
>
> \[1\] Dubinski et al., Towards More Realistic Membership Inference Attacks on Large Diffusion Models, WACV 24\.
> \[2\] Webster et al., A reproducible extraction of training images from diffusion models, arxiv 23\.
> \[3\] [https://github.com/Stability-AI/stablediffusion](https://github.com/Stability-AI/stablediffusion)

---

> ### Author Response · Authors · 2025-11-22
>
> ## **Comment C \- W2: Evaluation on Large-Scale Models (2/2)**
>
> ### **(3) Evaluation of MoFit on LDMs of different architecture, SD v3**
>
> **Defining Member/Hold-out Splits for SDv3.** We further evaluated MoFit on Stable Diffusion v3 (SDv3) to test its generalization to a different model architecture – specifically, from U-Net-based LDMs (*e.g.*, SDv1.5, SDv2.1) to the Multimodal Diffusion Transformer (MMDiT) \[1\]. However, SDv3 provides no publicly available details about its training dataset aside from the number of training images \[2\], making it challenging to construct reliable member/hold-out splits.
>
> Empirically, we observed that COCO samples yield significantly lower conditional and unconditional loss values (Eq. 1 and 2 in the main paper) compared to LAION-mi or CC3M. Based on this, we selected COCO as the member set and CC3M as the hold-out set for evaluating MoFit on SDv3. (We will include a supporting figure visualizing the loss gap across datasets in the Appendix.)
>
> **Evaluation on Pre-trained SDv3.**  We conducted membership inference using 100 images from each split, running all methods at a fixed diffusion timestep (t \= 328.1250, *i.e*., step 140). As shown in Table 2(c), MoFit outperforms all baselines, including CLiD under ground-truth conditions. These results further confirm that MoFit can match or even exceed methods that rely on full image-caption supervision, consistent with our findings on SD v1.4 (see Table 2 of the main paper for MS-COCO).
>
> ---
>
> \[1\] Esser, Patrick, et al. "Scaling rectified flow transformers for high-resolution image synthesis." Forty-first international conference on machine learning. 2024\.
> \[2\] [https://huggingface.co/stabilityai/stable-diffusion-3-medium-diffusers](https://huggingface.co/stabilityai/stable-diffusion-3-medium-diffusers)

---

> ### Author Response · Authors · 2025-11-22
>
> ## **Comment D \- W3: Computational Overhead**
>
> ### **(1) Computational Cost and Real-World Applicability**
>
> We acknowledge the reviewer’s concern regarding the efficiency of our two-stage optimization, which sequentially optimizes the model-fitted surrogate and its embedding. While our method incurs higher time cost, our primary goal was to significantly improve membership inference performance in the *caption-free setting* – without relying on VLM-generated captions. To the best of our knowledge, ***our work is the first to demonstrate such strong performance in this setting***, and thus we intentionally prioritized accuracy over efficiency.
>
> That said, we fully agree that cost-efficiency is critical. We thus report VRAM usage and runtime when using more lightweight optimizers in the section below.
>
> ---
>
> ### **(2) Lightweight Optimizers**
>
> | (a) Surrogate | VRAM (MB) | Runtime (sec) | (b) Embedding | VRAM (MB) | Runtime (sec) | (c) Per Model Scale | VRAM (MB) | Surrogate (sec) | Embedding (sec) | Total Runtime |
> | :---: | :---: | :---: | ----- | ----- | ----- | ----- | ----- | ----- | ----- | ----- |
> | Adam | 18053.79 | 333.24 | Adam (**Ours**) | 14799.11 | 38.28 | SD v1.4 | 20667.50 | 361.75 | 115.61 | 477.36 |
> | SGD | 18052.38 | 333.19 | SGD | 14797.22 | 38.21 | SD v1.5 | 20662.47 | 357.71 | 115.73 | 473.44 |
> | RMSProp | 18050.83 | 333.66 | RMSProp | 14803.43 | 38.15 | SD v2.1 | 17816.31 | 338.21 | 106.70 | 444.92 |
> | LION | 18049.72 | 333.57 | LION | 14803.64 | 38.15 | SD v3 | 41367.13 | 906.13 | 335.30 | 1241.43 |
> | **Ours** | 20667.50 | 358.96 |  |  |  |  |  |  |  |  |
>
> Table 5\. Optimizer- and model-specific VRAM usage and runtime overhead.
>
> To address concerns about computational costs in our two-stage pipeline – (i) model-fitted surrogate alignment and (ii) embedding extraction – we perform ablation studies comparing lightweight optimizers (SGD, RMSProp, and LION \[1\]). All optimizers were evaluated using 1,000 optimization steps (default configuration in the main paper).
>
> **(i) Model-fitted surrogate optimization**
> When optimizing the surrogate, we follow prior methods \[2, 3\] and update the perturbation using a single-step, first‑order update that directly applies the sign of the loss gradient – *i.e.*, sign(∇Lₓ) – to the input image. As evaluated in Table 5(a) for 1,000 steps (default setting), this design results in higher GPU consumption (up to 2,618 MB VRAM) and slightly longer runtime (up to 25.77 s) than optimizer-based alternatives. However, this single-step method avoids the need for multiple optimization trials of hyperparameter tuning (*e.g.*, learning rate or momentum), which can be costly and dataset-dependent. We view this trade-off as a design choice left to the adversary, depending on their resource constraints and deployment scenario.
>
> **(ii) Surrogate-driven embedding extraction**
> For the embedding extraction step from the model-fitted surrogate, we employ the Adam optimizer by default. In Table 5(b), while the optimizers were evaluated using 200 optimization steps (default setting for SD v1.4 fine-tuned with Pokemon dataset), we find that all optimizers perform consistently, with only a subtle difference. Given its stability and wide applicability across diverse tasks, we find Adam to be a practical default.
>
> ---
>
> ### **(3) Runtime and VRAM usage for each target model**
>
> Throughout the paper and the discussion phase, we evaluate four large-scale text-to-image diffusion models: SD v1.4, v1.5, v2.1, and v3. For each, we report VRAM usage and runtime per query image, using Float32 precision. All experiments were run on an RTX 4090, except SD v3, which required an A100 due to its size.
>
> As shown in Table 5(c), SD v1.4 and v1.5 show similar resource usage, while SD v2.1 is faster and more light-weighted. SD v3, however, demands significantly more VRAM and runtime, suggesting that future generative models may require larger GPU resources.
>
> ---
>
> \[1\] Chen, Xiangning, et al. "Symbolic discovery of optimization algorithms." Advances in neural information processing systems 36 (2023): 49205-49233.
> \[2\] Xue, Haotian et al. “Toward effective protection against diffusion based mimicry through score distillation.” ArXiv abs/2311.12832 (2023).
> \[3\] Salman, Hadi, et al. "Raising the cost of malicious ai-powered image editing." arXiv preprint arXiv:2302.06588 (2023).

---

> ### Author Response · Authors · 2025-11-22
>
> ## **Comment E \- W4: Stability**
>
> **Robustness to different target noises.** To further support the reviewer’s point regarding the MoFit’s sensitivity to hyperparameters, we conducted an additional experiment where the target noise $\\hat{\\epsilon}$ is randomized. Specifically, we sample four random noise vectors from a standard normal distribution and set each as the target noise. This evaluation helps confirm that MoFit remains robust to the choice of target noise, suggesting that any noise drawn from a normal distribution is sufficient for our method. The experiment was conducted on SD v1.4 fine-tuned with the Pokemon dataset, using 100 member and 100 hold-out samples.
>
> As shown in Table 3, ***MoFit performs consistently across all metrics when implemented with random noise***. We hope this helps address the reviewer’s concern regarding the sensitivity of MoFit to hyperparameter choices.
>
> ---
>
> ## **Clarification and Implementation Adjustment**
>
> ### **(1) Section 4.1 & 5**
>
> We would like to clarify that the optimization referred to as PGD in Section 4.1 was implemented as standard Gradient Descent (GD) without the projection (clipping) step. While we initially described the optimization as PGD, we acknowledge this mismatch in terminology and will revise the manuscript to reflect the exact implementation.
>
> Importantly, the numerical results throughout the paper remain unchanged. In practice, all reported experiments were conducted using GD with an initial noise level $\\varepsilon$ equivalent to the previously stated noise bound ($\\eta \= 0.3$). The step size $\\alpha$ is initialized to 0.15 and decays proportionally with the iteration count. Please refer to Eq. (10) in Appendix A.1 for the update equation.
>
> ---
>
> ### **(2) Section 5.5**
>
> In Section 5.5, we extend our evaluation of configuration (ii), where a random noise perturbation $\\delta$ is added to the query image. While the main paper reports results at a single perturbation level ($\\varepsilon \= 0.3$), we now investigate a broader range of noise levels, varying $\\varepsilon$ from 0.1 to 0.9. For each dataset, we identify the noise level that yields the strongest defensive effect and report the corresponding results in the second row of Table 4\.
> Notably, MoFit continues to exhibit strong and stable performance across all datasets, even under the best-performing perturbation level for each dataset. This demonstrates that our model-fitted surrogate $x^\*$ establishes a more robust and discriminative coupling with its optimized embedding than perturbed inputs such as $x \+ \\delta$ and other variants (*e.g.*, $x$ or $x \+ \\delta\_{\\text{max}}$).
>
> ---
>
> ### **(3) Figure 5**
>
> As our optimization uses GD without projection, we replace the noise bound (η) with the initial step size, which decays proportionally with iteration. Note that the experimental results remain unchanged; only the x-axis label is updated to reflect the corrected interpretation.

---

> ### Author Response · Authors · 2025-11-25
>
> ## **Additional Comment (1) \- W3: Early Stopping for Efficiency Improvement**
>
> | (a) Surrogate |  |  |  |  | (b) Embedding |  |  |  |  |  |  |  |  |  |
> | ----- | ----- | ----- | ----- | ----- | ----- | ----- | ----- | ----- | ----- | ----- | ----- | ----- | ----- | ----- |
> | **Threshold** | ASR | AUC | T@1F | Runtime (sec) | **Threshold** | ASR | AUC | T@1F | Runtime (sec) | **Threshold** | ASR | AUC | T@1F | Runtime (sec) |
> | 0.11 | 81.50 | 88.53 | 32.00 | 10.71 | 0.01 | 80.00 | 86.95 | 37.00 | 24.65 | 0.006 | 85.00 | 91.57 | 37.00 | 48.07 |
> | 0.10 | 82.50 | 88.99 | 25.00 | 12.80 | 0.009 | 81.00 | 88.17 | 35.00 | 29.11 | 0.005 | 86.00 | 92.97 | 43.00 | 58.19 |
> | 0.09 | 82.00 | 89.67 | 39.00 | 16.33 | 0.008 | 82.00 | 89.62 | 33.00 | 34.37 | 0.004 | 86.50 | 92.37 | 46.00 | 72.83 |
> | 0.08 | 84.00 | 90.15 | 42.00 | 22.32 | 0.007 | 84.00 | 90.50 | 31.00 | 40.45 | 0.003 | 85.79 | 91.77 | 47.96 | 88.81 |
> | 0.07 | 85.00 | 90.68 | 49.00 | 33.62 |  |  |  |  |  |  |  |  | Total Runtime | 116.38 |
> | 0.06 | 86.50 | 91.00 | 39.00 | 55.41 |  |  |  |  |  |  |  |  |  |  |
> |  |  |  | Total Runtime | 358.96 |  |  |  |  |  |  |  |  |  |  |
>
> Table 6\. Performance of MoFit under an early stopping regime applied to both surrogate optimization and embedding extraction.
>
> To investigate the reviewer’s insight regarding the importance of improving optimization efficiency, we conduct ***early stopping*** experiments that terminate the optimization process once a predefined loss threshold is reached. We apply this strategy to both (i) model-fitted surrogate optimization and (ii) surrogate-driven embedding extraction, using the SDv1.4 model fine-tuned on the MS-COCO dataset.
>
> **(i) Model-fitted surrogate optimization**
> We first average the final loss (Eq. (5)) from full optimization runs, which yield 0.02446 for members and 0.02259 for hold-outs. We then re-run the optimization and terminate early when the loss reaches a higher predefined threshold, choosing from {0.11, 0.1, 0.09, 0.08, 0.07, 0.06}. When a threshold is met, we save the corresponding surrogate and proceed to extract the embedding using the same setup as in the main experiments (300 iterations for MS-COCO, per Appendix A.1).
>
> In Table 6(a), we report membership inference results and average GPU runtime when applying early stopping to the surrogate optimization stage of MoFit. We evaluate on 100 member and 100 hold-out images from the MS-COCO dataset, using an NVIDIA RTX 4090\.
>
> As expected, performance improves as the loss threshold decreases, at the cost of increased runtime – demonstrating a clear trade-off between attack performance and computational efficiency.
>
> Notably, when compared to our baseline CLiD (83.50% ASR and 87.37 AUC in the same setting), MoFit reaches comparable or superior performance even when stopped early – for instance, at a loss threshold of 0.08, ***it achieves competitive results to CLiD while only taking 22.32 seconds per image*** (originally takes 358.96 sec). This suggests that an adversary can strategically balance efficiency and effectiveness by choosing an appropriate early stopping criterion.
>
> **(ii) Surrogate-driven embedding extraction**
> We follow the same procedure and compute the average final loss of Eq. (6) after embedding extraction, obtaining values of 0.00232 for members and 0.00229 for hold-outs. We then re-run full surrogate optimization (1,000 steps as default) for all samples, extract embeddings, and apply early stopping when the loss in Eq. (6) reaches a preset threshold: {0.01, 0.009, 0.008, 0.007, 0.006, 0.005, 0.004, 0.003}.
>
> As shown in Table 6(b), we again observe a trade-off between runtime and attack performance, mirroring the pattern in (i). Notably, MoFit reaches comparable performance to CLiD at a threshold of 0.007, while ***reducing runtime to 75.93 seconds per image***. This demonstrates that early-stopping can be effectively applied at both optimization and embedding stages to flexibly adjust the cost-performance balance of MoFit.

---

> ### Author Response · Authors · 2025-11-27
>
> ### **Gentle Reminder and Follow-Up**
>
> We appreciate your time and effort for reviewing our work again. We have posted clarifications and additional evaluations results to address your raised concerns. It would be grateful to let us know if there are more questions and concerns that we can address.
>
> For your convenience, we summarize the main points and indicate where they are addressed in the revised manuscript below, marked in blue for clarity:
>
> - Optimizer-specific VRAM usage and runtime – Appendix A.8.1
> - Early Stopping – Section 5.6.3, Appendix A.8.2
> - SD v1.5, SD v2, SD v3 – Appendix A.9
> - Random target noise – Appendix A.12

---

### Official Review · Reviewer_awTs · 2025-10-31

**Soundness:** 2
**Presentation:** 2
**Contribution:** 2
**Rating:** 4
**Confidence:** 3

**Summary:**

Latent diffusion models (LDMs) have high text-to-image generation fidelity but risk training data memorization, audited by membership inference attacks (MIAs). Existing MIAs need ground-truth captions, ineffective with only images. This work proposes MOFIT, a caption-free MIA framework with two stages. Experiments show it outperforms VLM-conditioned baselines, even matching/surpassing caption-dependent methods.

**Strengths:**

1. Fills the "caption-free" gap, fits real scenarios with undisclosed training annotations, and has practical value.
2. Based on member samples’ higher sensitivity to mismatched conditions, its two-stage optimization is logically consistent.
3. Outperforms VLM-based baselines across datasets/models, and surpasses caption-dependent methods in some cases.

**Weaknesses:**

1. Its two-stage optimization takes 7-9 minutes per image on RTX 4090, far slower than VLM-based baselines, unable to handle large-scale data.
2. No tests on non-ideal images or small-sample LDMs; key parameters are only validated on Pokemon, lacking cross-scenario adaptability.
3. It excludes recent caption-free MIA studies, uses only 2 VLMs as baselines, and lacks theoretical explanation for member samples’ sensitivity.

**Questions:**

1. Optimize efficiency with adaptive stopping criteria and lightweight optimizers, and quantify efficiency-performance trade-offs for large-scale applicability.
2. Test on non-ideal images/small-sample LDMs, add more VLMs/recent studies as baselines to boost result robustness.
3. Derive member samples’ sensitivity from LDM mechanisms, analyze risks, and propose defense strategies and usage rules.

---

> ### Author Response · Authors · 2025-11-22
>
> ## **Overview of All Reviewer Comments (1/2)**
>
> To thoroughly address the reviewers’ questions and concerns, we conducted a comprehensive set of additional experiments. For clarity and convenience, we summarize all relevant tables below. We kindly invite you to review this summary and refer to the corresponding comments (Comment A to F) for detailed explanations.
> **Note.** T@1F in the table refers to  the TPR@1%FPR metric.
>
> ---
>
> ### **| W1Q1. Early stopping for improved efficiency**
> We thank the reviewer for highlighting runtime concerns in our two-stage optimization. As noted in **Comment A**, we report optimizer-specific VRAM usage and runtime. We will further quantify the efficiency–performance trade-off using early stopping in a future comment.
>
> ---
>
> ### **| W2Q2. Evaluation on LDMs fine-tuned on non-ideal or limited number of images**
>
> |  |  | (a) Gaus. Blur |  |  | (b) JPEG |  |  | (c) 500 images |  |  | (d) 1000 images |  |  |
> | ----- | ----- | :---: | ----- | ----- | :---: | ----- | ----- | :---: | :---: | ----- | ----- | :---: | ----- |
> | **Condition** | **Method** | ASR | AUC | T@1F | ASR | AUC | T@1F | ASR | AUC | T@1F | ASR | AUC | T@1F |
> | GT | CLiD | 89.10 | 92.27 | 70.40 | 85.50 | 89.59 | 61.00 | 80.50 | 83.35 | 55.00 | 83.00 | 86.76 | 60.00 |
> | VLM | Loss | 64.10 | 68.17 | 4.20 | 63.40 | 66.53 | 3.80 | 64.50 | 68.05 | 5.00 | 64.50 | 67.72 | 4.00 |
> | VLM | SecMI | 60.10 | 62.15 | 5.20 | 54.00 | 54.59 | 2.80 | 62.00 | 62.31 | 4.00 | 58.00 | 59.49 | 4.00 |
> | VLM | PIA | 63.10 | 65.28 | 4.40 | 64.90 | 67.18 | 5.00 | 66.50 | 70.71 | 7.00 | 69.50 | 71.83 | 9.00 |
> | VLM | PFAMI | 80.60 | 87.51 | 27.72 | 78.04 | 84.39 | 36.03 | 83.00 | 87.26 | 23.00 | 83.50 | 90.45 | 31.00 |
> | VLM | CLiD | 81.00 | 87.17 | 44.60 | 78.80 | 85.21 | 39.20 | 76.00 | 80.32 | 23.00 | 78.50 | 82.55 | 29.00 |
> | $\\phi^\*$ | **Ours** | 88.70 | 94.92 | 54.20 | 82.80 | 89.75 | 26.00 | 86.00 | 91.89 | 27.00 | 88.00 | 90.35 | 30.00 |
>
> Table 1\. Performance against LDMs fine-tuned with blur/compression and few training samples (500 or 1,000 images).
>
> To address the reviewer’s concern regarding robustness and cross-scenario applicability, we evaluate all baselines and MoFit against LDMs fine-tuned under non-ideal conditions and with limited training data. As shown in Table 1, ***MoFit consistently outperforms alternatives across all cross-scenario settings***, indicating strong applicability in practical cases where images are degraded or only few training samples are available. Further details are provided in **Comment B**.
>
> ---
>
> ### **| W3Q2. Evaluation on recent MIA study and VLMs**
>
> |  |  | Pokemon |  |  | MS-COCO |  |  | Flickr |  |  |
> | ----- | ----- | :---: | ----- | ----- | :---: | ----- | ----- | :---: | ----- | ----- |
> |  |  | ASR | AUC | T@1F | ASR | AUC | T@1F | ASR | AUC | T@1F |
> | GT | CLiD | 96.52 | 99.17 | 90.14 | 86.50 | 90.27 | 68.80 | 91.10 | 95.13 | 77.20 |
> | GT | **ReDiffuse** | 52.40 | 49.53 | 0.72 | 54.20 | 52.41 | 1.20 | 60.00 | 48.23 | 0.20 |
> | VLM | Loss | 72.27 | 78.99 | 4.81 | 63.70 | 67.88 | 4.80 | 61.60 | 64.24 | 5.40 |
> | VLM | SecMI | 78.51 | 6.22 | 6.97 | 57.30 | 58.07 | 4.20 | 54.00 | 52.38 | 2.00 |
> | VLM | PIA | 71.79 | 76.76 | 10.82 | 66.00 | 69.70 | 6.60 | 61.00 | 64.05 | 5.00 |
> | VLM | PFAMI | 74.43 | 81.25 | 6.01 | 80.40 | 87.50 | 29.40 | 76.90 | 84.99 | 24.80 |
> | VLM | CLiD | 77.55 | 83.43 | 19.23 | 80.90 | 86.53 | 50.80 | 79.00 | 85.16 | 40.60 |
> | VLM | **ReDiffuse** | 52.16 | 49.75 | 0.72 | 55.30 | 53.37 | 1.60 | 51.30 | 48.73 | 0.20 |
> | **MoonDream2** | CLiD | 74.91 | 80.20 | 13.22 | 77.50 | 83.41 | 31.00 | 75.20 | 81.62 | 33.40 |
> | **Molmo** | CLiD | 78.00 | 81.40 | 8.00 | 72.00 | 71.65 | 17.00 | 68.50 | 69.23 | 21.00 |
> | Emb. | Ours | 94.48 | 97.30 | 50.48 | 88.00 | 94.17 | 47.00 | 86.00 | 91.32 | 53.20 |
>
> Table 2\. Evaluation of the additional MIA method ReDiffuse \[1\] and two recent VLMs \[2,3\]. The results extend Table 2 from the main paper.
>
> In Table 2, we include additional experiments for the recent MIA method ReDiffusion \[1\] and two VLMs, MoonDream2 \[2\] and Molmo Flux captioner \[3\], to address the reviewer’s concern. We observe that ***ReDiffuse shows limited applicability in practical cases*** where members and hold-outs originate from the same distribution. Moreover, although recent VLMs produce more descriptive captions, ***they still underperform MoFit***, underscoring the importance of studying the caption-free setting. Further explanations regarding ReDiffusion and the VLMs are provided in **Comment C**.
>
> ---
>
> \[1\] Li, Jingwei, et al. "Towards black-box membership inference attack for diffusion models." arXiv preprint arXiv:2405.20771 (2024).
> \[2\] [https://moondream.ai/](https://moondream.ai/)
> \[3\] Deitke, M., et al. "Molmo and pixmo: Open weights and open data for state-of-the-art vision-language models, 2024." URL https://arxiv. org/abs/2409.17146.

---

> ### Author Response · Authors · 2025-11-22
>
> ## **Overview of All Reviewer Comments (2/2)**
>
> ### **| W3Q3. Sensitivity difference between members and hold-outs**
>
> We thank the reviewer for the opportunity to clarify the core motivation of our method – namely, that member samples exhibit greater sensitivity to condition changes than hold-outs. ***We provide a detailed explanation in*** ***Comment D***, ranging from the theoretical basis of this sensitivity difference to how it directly motivates our proposed approach. We hope this clarifies our design choices and assists the reviewer’s understanding.
>
> ---
>
> ### **| Q3. Potential defensive strategy, LoRA**
>
> |  |  | LoRA |  |  |
> | ----- | ----- | :---: | ----- | ----- |
> | **Condition** | **Method** | ASR | AUC | T@1F |
> | GT | CLiD | 59.00 | 52.05 | 1.00 |
> | VLM | Loss | 54.50 | 49.26 | 0.00 |
> | VLM | SecMI | 58.50 | 53.50 | 1.00 |
> | VLM | PIA | 58.50 | 53.50 | 0.00 |
> | VLM | PFAMI | 73.50 | 77.50 | 1.00 |
> | VLM | CLiD | 59.00 | 53.11 | 1.00 |
> | $\\phi^\*$ | **Ours** | 58.50 | 54.35 | 0.00 |
>
> Table 3\. Performance of the baselines and MoFit against a potentially critical defensive strategy: LoRA.
>
> We agree with the reviewer on the importance of exploring potential defensive strategies. As shown in Table 3, we observe that ***MoFit is highly ineffective against LoRA-adapted LDMs***. We attribute this robustness to LoRA’s limited parameter updates, which likely reduce the model’s capacity to remember training samples. A more detailed discussion is provided in **Comment E**.
>
> ---
>
> ### **| Notes on Implementation Details and Revisions**
> We clarify that the optimization previously referred to as PGD was actually implemented as standard Gradient Descent (GD), with all reported results unaffected. Additionally, we extend the perturbation-level analysis in Section 5.5, further confirming MoFit’s superiority over alternative surrogate types. Please refer to **Comment F** for details.

---

> ### Author Response · Authors · 2025-11-22
>
> ## **Comment A \- W1Q1: Large-Scale Applicability**
>
> ### **(1) Computational cost and real-world applicability**
>
> We acknowledge the reviewer’s concern regarding the runtime of our two-stage optimization, which sequentially optimizes the model-fitted surrogate and its embedding. While our method incurs higher time cost, our primary goal was to significantly improve membership inference performance in the *caption-free setting* – without relying on VLM-generated captions. To the best of our knowledge, ***our work is the first to demonstrate such strong performance in this setting***, and thus we intentionally prioritized accuracy over efficiency.
>
> That said, we fully agree that cost-efficiency is critical. We thus report VRAM usage and runtime when using more lightweight optimizers in the section below. *We will further quantify the efficiency–performance trade-off using early stopping in a future comment*.
>
> ---
>
> ### **(2) Lightweight optimizers**
>
> To address concerns about computational costs in our two-stage pipeline – (i) model-fitted surrogate alignment and (ii) embedding extraction – we perform ablation studies comparing lightweight optimizers (SGD, RMSProp, and LION \[1\]).
>
> | (a) Surrogate | VRAM (MB) | Runtime (sec) | (b) Embedding | VRAM (MB) | Runtime (sec) |
> | :---: | :---: | :---: | ----- | ----- | ----- |
> | Adam | 18053.79 | 333.24 | Adam (**Ours**) | 14799.11 | 38.28 |
> | SGD | 18052.38 | 333.19 | SGD | 14797.22 | 38.21 |
> | RMSProp | 18050.83 | 333.66 | RMSProp | 14803.43 | 38.15 |
> | LION | 18049.72 | 333.57 | LION | 14803.64 | 38.15 |
> | **Ours** | 20667.5 | 358.96 |  |  |  |
>
> Table 4\. Comparison of GPU VRAM usage and runtime overhead across optimizers during (a) surrogate optimization and (b) embedding extraction.
>
> **(i) Model-fitted surrogate optimization**
> When optimizing the surrogate, we follow prior methods \[2, 3\] and update the perturbation using a single-step, first‑order update that directly applies the sign of the loss gradient – *i.e.*, sign(∇Lₓ) – to the input image. As evaluated in Table 4(a) for 1,000 steps (default setting), this design results in higher GPU consumption (up to 2,618 MB VRAM) and slightly longer runtime (up to 25.77 s) than optimizer-based alternatives. However, this single-step method avoids the need for multiple optimization trials of hyperparameter tuning (*e.g.*, learning rate or momentum), which can be costly and dataset-dependent. We view this trade-off as a design choice left to the adversary, depending on their resource constraints and deployment scenario.
>
> **(ii) Surrogate-driven embedding extraction**
> For the embedding extraction step from the model-fitted surrogate, we employ the Adam optimizer by default. In Table 4(b), while the optimizers were evaluated using 200 optimization steps (default setting for SD v1.4 fine-tuned with Pokemon dataset), we find that all optimizers perform consistently, with only a subtle difference. Given its stability and wide applicability across diverse tasks, we find Adam to be a practical default.
>
> ---
>
> \[1\] Chen, Xiangning, et al. "Symbolic discovery of optimization algorithms." Advances in neural information processing systems 36 (2023): 49205-49233.
> \[2\] Xue, Haotian et al. “Toward effective protection against diffusion based mimicry through score distillation.” ArXiv abs/2311.12832 (2023).
> \[3\] Salman, Hadi, et al. "Raising the cost of malicious ai-powered image editing." arXiv preprint arXiv:2302.06588 (2023).

---

> ### Author Response · Authors · 2025-11-22
>
> ## **Comment B \- W2Q2: Robustness Evaluation**
>
> We appreciate the reviewer’s feedback regarding MoFit’s robustness under suboptimal fine-tuning conditions. To this end, we conduct additional experiments in two realistic scenarios: (1) fine-tuning on non-ideal training images (*e.g.*, blur, compression), and (2) fine-tuning with limited data. These settings reflect practical challenges where training data may be noisy or scarce.
>
> ---
>
> ### **(1) LDM fine-tuned on non-ideal images**
>
> To address the reviewer’s concern regarding the robustness of our method against non-ideally fine-tuned LDMs, we evaluate input randomization as a defense method. Following the standard SD v1.4 fine‑tuning protocol on MS‑COCO, we additionally apply:
> (a) **Gaussian blur** (3×3 kernel, σ ∈ \[0.1, 2.0\]) and (b) **JPEG compression** (quality \= 60).
>
> We fine‑tune separate LDMs under each augmentation and directly evaluate MoFit using embeddings crafted from the non‑augmented base model—reflecting a challenging setting where our model-fitted embedding is constructed without awareness of the input-space changes introduced during fine-tuning.
>
> Using 500 member and 500 hold‑out images, Table 1 (a) and (b) show that all methods experience comparable or slightly degraded performance under both augmentations. Importantly, two trends remain consistent:
> - (i) All baselines degrade significantly when shifting from GT to VLM‑generated captions
> - (ii) MoFit consistently outperforms all baselines under VLM‑generated captions across both ASR and AUC – ***even when input randomization is applied, achieving ASRs over 82.8%**.*
>
> These results suggest that the alignment between the model-fitted surrogate and its embedding remains strong, despite input perturbations during training – leading to significant misalignment at the inference-time that MoFit effectively leverages.
>
> ---
>
> ### **(2) LDM fine-tuned on small samples**
>
> We also address the concern regarding MoFit’s robustness when the target LDM is fine-tuned using a limited number of training images. To simulate this low-resource scenario, we fine-tune an LDM using only 500 or 1,000 images from the MS-COCO dataset over 30,000 or 60,000 steps, respectively – maintaining the same steps-to-image ratio used in our main experiments (2,500 images for 150,000 steps), and following the setting introduced in our baseline \[1\].
>
> Table 1 (c) and (d) report the membership inference results for all baseline methods and MoFit, evaluated on 100 member and 100 hold-out images. As expected, prior methods show a noticeable performance drop when moving from ground-truth captions to VLM-generated captions, highlighting the difficulty of caption-free MIA in this sparse data regime. In contrast, MoFit continues to outperform baselines even under the VLM-generated caption setting, demonstrating ***strong performance despite the reduced training data***.
>
> These results indicate that MoFit generalizes well to low-resource LDMs, suggesting broader applicability and robustness in practical scenarios where only a limited number of private images are available for fine-tuning.
>
> ---
>
> \[1\] Zhai, Shengfang, et al. "Membership inference on text-to-image diffusion models via conditional likelihood discrepancy." Advances in Neural Information Processing Systems 37 (2024): 74122-74146.

---

> ### Author Response · Authors · 2025-11-22
>
> ## **Comment C \- W3Q2: Recent MIA Method and VLMs**
>
> We thank the reviewer for pointing out recent MIA studies and would like to first clarify the positioning of our work in relation to other MIA studies.
>
> Among the baselines evaluated in our study – SecMI \[1\], PIA \[2\], and CLiD \[3\] – all consider the caption-free scenario, where ground-truth (GT) captions are unavailable during inference. However, we emphasize that ***these methods typically address this by leveraging captions generated by vision-language models (VLMs)***, which, as shown in Tables 2 and 3 in the main paper, results in significant performance degradation compared to GT captions.
>
> Given the notable performance degradation observed with VLM-generated captions, ***our work explores an alternative approach for membership inference in the caption-free setting***. Specifically, motivated by our observation that member samples exhibit greater sensitivity to condition perturbations than hold-outs, we design a surrogate-driven method that does not rely on VLM-generated captions. To the best of our knowledge, this work is the first work to improve performance in the caption-free setting by exploring alternatives to the commonly adopted – but often suboptimal – VLM-generated captions.
>
> ---
>
> ### **(1) A recent membership inference studies: ReDiffuse**
>
> We appreciate the reviewers for pointing out recent studies on membership inference against text-to-image diffusion models. One such work, ReDiffuse \[4\], proposes a black-box attack that infers membership based on the reconstruction error between a query image and multiple generated samples.
>
> While ReDiffuse is black-box with respect to model architecture and parameters, ***it assumes access to ground-truth (GT) captions during inference***. In more realistic settings without GT captions, it still relies on VLM-generated captions – similar to prior baselines.
>
> Additionally, while recent studies \[3, 5\] highlight the importance of evaluating in settings where members and hold-outs are drawn from the same distribution, ReDiffuse adopts a less challenging protocol where the two groups differ in distribution, making them easier to distinguish.
>
> We evaluated ReDiffuse under our more challenging, distributionally aligned setting. As shown in Table 2, its performance drops significantly under both GT and VLM-generated captions. This suggests that while ReDiffuse is effective when member and hold-out distributions are disjoint, ***its ability to infer membership deteriorates in more realistic scenarios where the distributions are matched***.
>
> ---
>
> ### **(2) Captions generated from recent VLM models**
>
> We appreciate the reviewer’s suggestion to explore caption generation using recent vision-language models (VLMs). In response, we generate captions using two new VLMs – *MoonDream2* ([https://moondream.ai/](https://moondream.ai/)) and *Molmo Flux captioner* \[6\] – and re-implement the baselines using the same protocol with these updated captions for fair comparison.
>
> As shown in Table 2, we condition CLiD with the VLM-generated captions and evaluate on three fine-tuned SD v1.4 models (Pokemon, MS-COCO, and Flickr). While our method consistently outperforms all alternatives, ***we still observe notable performance degradation when using VLM-generated captions***. These results highlight that even highly descriptive captions from powerful VLMs fail to recover the original image-caption supervision signal, underscoring the need for research in caption-free settings as our approach.
>
> ---
>
> \[1\] Duan, Jinhao, et al. "Are diffusion models vulnerable to membership inference attacks?." Proceedings of the 40th International Conference on Machine Learning. 2023\.
> \[2\] Kong, Fei, et al. "An efficient membership inference attack for the diffusion model by proximal initialization." arXiv preprint arXiv:2305.18355 (2023).
> \[3\] Zhai, Shengfang, et al. "Membership inference on text-to-image diffusion models via conditional likelihood discrepancy." Advances in Neural Information Processing Systems 37 (2024): 74122-74146.
> \[4\] Li, Jingwei, et al. "Towards black-box membership inference attack for diffusion models." arXiv preprint arXiv:2405.20771 (2024).
> \[5\] Dubiński, Jan, et al. "Towards more realistic membership inference attacks on large diffusion models." Proceedings of the IEEE/CVF Winter Conference on Applications of Computer Vision. 2024\.
> \[6\] Deitke, M., et al. "Molmo and pixmo: Open weights and open data for state-of-the-art vision-language models, 2024." URL https://arxiv.org/abs/2409.17146.

---

> ### Author Response · Authors · 2025-11-22
>
> ## **Comment D \- W3Q3: Explanation for Sensitivity Difference**
>
> We thank the reviewer for the opportunity to clarify the theoretical basis behind our key observation: *member samples exhibit higher sensitivity to condition change than hold-out samples.*
>
> ---
>
> ### **(1) Tight Coupling in Member-Caption Pairs.**
>
> In latent diffusion models (LDMs), *member samples* are trained with ***tightly coupled image-caption pairs*** ($x$, $c$) to minimize the condition-aware denoising loss $L\_{\\text{cond}}$ (Eq. 1). At each training step, the latent $z\_t$ of image $x$ is denoised by the U-Net $\\epsilon\_\\theta$ conditioned on its corresponding caption $c$,  forming a strong alignment. Replacing $c$ with an alternative $c’$ – even a descriptive VLM-generated caption – disrupts this alignment and increases the loss, as shown empirically in Section 3.3 of the main paper. Note that the model is simultaneously trained with the unconditional loss $L\_{\\text{uncond}}$ (Eq. 2), where the conditioning is dropped.
>
> In contrast, *hold-out* samples are never observed during training, and thus there does not exist any meaningful association with their captions. Consequently, altering the caption has minimal effect on $L\_{\\text{cond}}$, as all pairings remain unfamiliar to the model.
>
> This difference in condition sensitivity is empirically validated across Figures 1(d,e), 3(b), 6(b), and Table 1, for all domains (Pokemon, MS-COCO, Flickr). In every case, member samples exhibit larger increases in $L\_{\\text{cond}}$ under condition shifts, aligning with our explanation and supporting MoFit’s design.
>
> ---
>
> ### **(2) Leveraging Sensitivity Difference in MoFit.**
>
> Our method capitalizes on the observed difference in condition sensitivity between members and hold-outs as a discriminative signal in the caption-free setting. For *members*, the training-established coupling between image $x$ and caption $c$ makes them highly sensitive to condition changes (*i.e.*, replacing $c$ with $c’$ increases $L\_{\\text{cond}}$). *Hold-outs*, lacking such learned alignment, exhibit negligible sensitivity.
>
> ***MoFit amplifies this gap by constructing an even tighter surrogate pair*** **($x^\*$, $\\phi^\*$)***: it first generates a model-fitted surrogate* $x^\*$ from the query $x$, then derives a condition embedding $\\phi^\*$ aligned to $x^\*$. When conditioning the model on $\\phi^\*$ but evaluating on the original $x$, member samples incur a sharper rise in $L\_{\\text{cond}}$, while hold-outs remain largely unaffected – yielding a strong separability signal.
>
> ---
>
> ### **(3) Discriminative Signal of MoFit**.
>
> MoFit leverages a simple yet effective score: $L\_{\\text{MoFit}} \= L\_{\\text{cond}} \- L\_{\\text{uncond}}$, which amplifies two empirically observed asymmetries between member and hold-out samples.
>
> - **(i) Higher condition sensitivity in members**: Due to the strong ($x^\*$, $\\phi^\*$) coupling formed during our optimization, member samples exhibit larger increases in 𝐿cond when this strong coupling is mismatched – $\\phi^\*$ conditions the query image $x$, not its pair $x^\*$.
> - **(ii) Lower unconditional loss in members**: Since only members contribute to optimizing $L\_{\\text{uncond}}$ during training, they naturally achieve lower unconditional losses than unseen hold-outs.
>
> By subtracting $L\_{\\text{uncond}}$ from $L\_{\\text{cond}}$, MoFit amplifies this bidirectional contrast, yielding a stronger signal that distinctly separates members from hold-outs. This is summarized in the following table.
>
> | Sample | $L\_{\text{cond}}$ under $\\phi^\*$ | $L\_{\\text{uncond}}$ | $L\_{\\text{MoFit}} \= L\_{\\text{cond}} \- L\_{\\text{uncond}}$ |
> |:-----------:|:-----------------------------------:|:---------------------:|:----------------------------------------------------------------:|
> | Member      | High                                | Low                   | **High**                                                         |
> | Hold-out    | Low                                 | High                  | **Low**                                                          |

---

> ### Author Response · Authors · 2025-11-22
>
> ## **Comment  E \- Q3: Potential Defensive Strategy**
>
> We thank the reviewer for raising this important point. The original draft lacked sufficient discussion on potential defenses, which we now address. We will include this extended analysis and the corresponding tables in the appendix of the revised manuscript.
>
> To address the reviewer’s question regarding promising defenses, we find that ***Low‑Rank Adaptation (LoRA) substantially reduces the effectiveness of MoFit.*** Specifically, replacing full U‑Net fine‑tuning with LoRA leads to a noticeable performance drop for our method.
>
> ---
>
> ### **(1) Studies related to robustness of LoRA against MIA**
>
> Recent studies in language modeling have evaluated membership inference under LoRA fine-tuning. While early works \[1\] suggest robustness, later studies \[2,3\] show that LoRA’s limited parameter updates significantly reduce attack effectiveness. For latent diffusion models, Luo et al. \[4\] suggest that LoRA-adapted LDMs may remain vulnerable, but their evaluation is limited to earlier MIA methods \[5,6\] (*e.g.*, the “Loss” method in our paper), and does not cover more recent approaches.
>
> ---
>
> ### **(2) Our experiment against LoRA**
>
> In contrast, Table 3 shows that when evaluated on 100 samples of Pokemon training set used to fine-tune the LoRA-adapted SDv1.4 \[7\], ***performance drops to near-random levels for both MoFit and most of the baselines under ground-truth and VLM-generated captions***. We hypothesize that this robustness stems from LoRA’s minimal footprint, which preserves most of the original weights and reduces the model’s capacity to remember specific training samples \[2\].
>
> ---
>
> ### **(3) Reason behind the robustness of PFAMI \[8\]**
>
> Baselines other than PFAMI rely on *cross-query* comparisons – typically based on the relative ranking of diffusion losses across different queries. The poor performance of these baselines under LoRA, as shown in Table 3, suggests that the loss values of member and non-member queries become increasingly similar. This makes it difficult for cross-query methods to distinguish between them based on relative loss ordering.
>
> However, PFAMI uses *within-query* scores by applying multiple augmentations (*i.e.*, crop) to a single query image and computing the loss for each. The membership score is based on the relative variation within these losses. Since it does not rely on comparisons across different queries, PFAMI remains robust even when LoRA causes the overall loss values of member and non-member samples to become similar. The relative differences within a single query are preserved, making it less sensitive to such global shifts.
>
> ---
>
> \[1\] Wen, Rui, et al. "Last one standing: A comparative analysis of security and privacy of soft prompt tuning, lora, and in-context learning." arXiv preprint arXiv:2310.11397 (2023).
> \[2\] Amit, Guy, Abigail Goldsteen, and Ariel Farkash. "SoK: reducing the vulnerability of fine-tuned language models to membership inference attacks." arXiv preprint arXiv:2403.08481 (2024).
> \[3\] Liu, Ruixuan, et al. "Precurious: How innocent pre-trained language models turn into privacy traps”, arxiv 2024\.
> \[4\] Luo, Zihao, et al. "Privacy-Preserving Low-Rank Adaptation Against Membership Inference Attacks for Latent Diffusion Models." Proceedings of the AAAI Conference on Artificial Intelligence. Vol. 39\. No. 6\. 2025\.
> \[5\] Hu, Hailong, and Jun Pang. "Membership inference of diffusion models." arXiv preprint arXiv:2301.09956 (2023).
> \[6\] Matsumoto, Tomoya, Takayuki Miura, and Naoto Yanai. "Membership inference attacks against diffusion models." 2023 IEEE Security and Privacy Workshops (SPW). IEEE, 2023\.
> \[7\] [https://huggingface.co/sr5434/sd-pokemon-model-lora](https://huggingface.co/sr5434/sd-pokemon-model-lora)
> \[8\] Fu, Wenjie, et al. "A probabilistic fluctuation based membership inference attack for diffusion models." arXiv preprint arXiv:2308.12143 (2023).

---

> ### Author Response · Authors · 2025-11-22
>
> ## **Comment F: Clarification and Implementation Adjustment**
>
> ### **(1) Section 4.1 & 5**
>
> We would like to clarify that the optimization referred to as PGD in Section 4.1 was implemented as standard Gradient Descent (GD) without the projection (clipping) step. While we initially described the optimization as PGD, we acknowledge this mismatch in terminology and will revise the manuscript to reflect the exact implementation.
>
> Importantly, the numerical results throughout the paper remain unchanged. In practice, all reported experiments were conducted using GD with an initial noise level $\\varepsilon$ equivalent to the previously stated noise bound ($\\eta \= 0.3$). The step size $\\alpha$ is initialized to 0.15 and decays proportionally with the iteration count. Please refer to Eq. (10) in Appendix A.1 for the update equation.
>
> ---
>
> ### **(2) Section 5.5**
>
> In Section 5.5, we extend our evaluation of configuration (ii), where a random noise perturbation $\\delta$ is added to the query image. While the main paper reports results at a single perturbation level ($\\varepsilon \= 0.3$), we now investigate a broader range of noise levels, varying $\\varepsilon$ from 0.1 to 0.9. For each dataset, we identify the noise level that yields the strongest defensive effect and report the corresponding results in the second row of Table 4\.
> Notably, MoFit continues to exhibit strong and stable performance across all datasets, even under the best-performing perturbation level for each dataset. This demonstrates that our model-fitted surrogate $x^\*$ establishes a more robust and discriminative coupling with its optimized embedding than perturbed inputs such as $x \+ \\delta$ and other variants (*e.g.*, $x$ or $x \+ \\delta\_{\\text{max}}$).
>
> ---
>
> ### **(3) Figure 5**
>
> As our optimization uses GD without projection, we replace the noise bound (η) with the initial step size, which decays proportionally with iteration. Note that the experimental results remain unchanged; only the x-axis label is updated to reflect the corrected interpretation.

---

> ### Author Response · Authors · 2025-11-25
>
> ## **Additional Comment (1) \- W1Q1: Early Stopping for Efficiency Improvement**
>
> | (a) Surrogate |  |  |  |  | (b) Embedding |  |  |  |  |  |  |  |  |  |
> | ----- | ----- | ----- | ----- | ----- | ----- | ----- | ----- | ----- | ----- | ----- | ----- | ----- | ----- | ----- |
> | **Threshold** | ASR | AUC | T@1F | Runtime (sec) | **Threshold** | ASR | AUC | T@1F | Runtime (sec) | **Threshold** | ASR | AUC | T@1F | Runtime (sec) |
> | 0.11 | 81.50 | 88.53 | 32.00 | 10.71 | 0.01 | 80.00 | 86.95 | 37.00 | 24.65 | 0.006 | 85.00 | 91.57 | 37.00 | 48.07 |
> | 0.10 | 82.50 | 88.99 | 25.00 | 12.80 | 0.009 | 81.00 | 88.17 | 35.00 | 29.11 | 0.005 | 86.00 | 92.97 | 43.00 | 58.19 |
> | 0.09 | 82.00 | 89.67 | 39.00 | 16.33 | 0.008 | 82.00 | 89.62 | 33.00 | 34.37 | 0.004 | 86.50 | 92.37 | 46.00 | 72.83 |
> | 0.08 | 84.00 | 90.15 | 42.00 | 22.32 | 0.007 | 84.00 | 90.50 | 31.00 | 40.45 | 0.003 | 85.79 | 91.77 | 47.96 | 88.81 |
> | 0.07 | 85.00 | 90.68 | 49.00 | 33.62 |  |  |  |  |  |  |  |  | Total Runtime | 116.38 |
> | 0.06 | 86.50 | 91.00 | 39.00 | 55.41 |  |  |  |  |  |  |  |  |  |  |
> |  |  |  | Total Runtime | 358.96 |  |  |  |  |  |  |  |  |  |  |
>
> Table 5\. Performance of MoFit under an early stopping regime applied to both surrogate optimization and embedding extraction.
>
> To investigate the reviewer’s insight regarding the importance of improving optimization efficiency, we conduct ***early stopping*** experiments that terminate the optimization process once a predefined loss threshold is reached. We apply this strategy to both (i) model-fitted surrogate optimization and (ii) surrogate-driven embedding extraction, using the SDv1.4 model fine-tuned on the MS-COCO dataset.
>
> **(i) Model-fitted surrogate optimization**
> We first average the final loss (Eq. (5)) from full optimization runs, which yield 0.02446 for members and 0.02259 for hold-outs. We then re-run the optimization and terminate early when the loss reaches a higher predefined threshold, choosing from {0.11, 0.1, 0.09, 0.08, 0.07, 0.06}. When a threshold is met, we save the corresponding surrogate and proceed to extract the embedding using the same setup as in the main experiments (300 iterations for MS-COCO, per Appendix A.1).
>
> In Table 5(a), we report membership inference results and average GPU runtime when applying early stopping to the surrogate optimization stage of MoFit. We evaluate on 100 member and 100 hold-out images from the MS-COCO dataset, using an NVIDIA RTX 4090\.
>
> As expected, performance improves as the loss threshold decreases, at the cost of increased runtime – demonstrating a clear trade-off between attack performance and computational efficiency.
>
> Notably, when compared to our baseline CLiD (83.50% ASR and 87.37 AUC in the same setting), MoFit reaches comparable or superior performance even when stopped early – for instance, at a loss threshold of 0.08, ***it achieves competitive results to CLiD while only taking 22.32 seconds per image*** (originally takes 358.96 sec). This suggests that an adversary can strategically balance efficiency and effectiveness by choosing an appropriate early stopping criterion.
>
> **(ii) Surrogate-driven embedding extraction**
> We follow the same procedure and compute the average final loss of Eq. (6) after embedding extraction, obtaining values of 0.00232 for members and 0.00229 for hold-outs. We then re-run full surrogate optimization (1,000 steps as default) for all samples, extract embeddings, and apply early stopping when the loss in Eq. (6) reaches a preset threshold: {0.01, 0.009, 0.008, 0.007, 0.006, 0.005, 0.004, 0.003}.
>
> As shown in Table 5(b), we again observe a trade-off between runtime and attack performance, mirroring the pattern in (i). Notably, MoFit reaches comparable performance to CLiD at a threshold of 0.007, while ***reducing runtime to 75.93 seconds per image***. This demonstrates that early-stopping can be effectively applied at both optimization and embedding stages to flexibly adjust the cost-performance balance of MoFit.

---

> ### Author Response · Authors · 2025-11-27
>
> ### **Gentle Reminder and Follow-Up**
>
> We appreciate your time and effort for reviewing our work again. We have posted clarifications and additional evaluations results to address your raised concerns. It would be grateful to let us know if there are more questions and concerns that we can address.
>
> For your convenience, we summarize the main points and indicate where they are addressed in the revised manuscript below, marked in blue for clarity:
>
> - Input Randomization – Section 5.6.2
> - LoRA – Section 5.6.2, Appendix A.7
> - Optimizer-specific VRAM usage and runtime – Appendix A.8.1
> - Early Stopping – Section 5.6.3, Appendix A.8.2
> - Evaluation with recent VLMs – Appendix A.11
> - Limited number of images – Appendix A.14

---

### Official Review · Reviewer_exJi · 2025-11-10

**Soundness:** 3
**Presentation:** 3
**Contribution:** 3
**Rating:** 6
**Confidence:** 2

**Summary:**

The paper presents new methods for membership inference attacks on diffusion models, focuses on generative/representation learning, and addresses privacy.

**Strengths:**

By tackling the MIA problem in the absence of ground-truth captions, the work fills a clear gap in current literature, reflecting realistic adversarial constraints faced in practice.

**Weaknesses:**

1. Limited diversity in experiments with large models: While SD v1.5 is included, the exploration of truly large-scale public models (e.g., with more challenging real-world splits or harder negative scenarios) is limited. The LAION-mi split required special curation and so does not test MoFit “in the wild” under extreme generalization.


2. The paper focuses heavily on the positive results of MoFit, but does not sufficiently examine its limitations, potential false-positive drivers (e.g., for out-of-distribution or near-duplicate samples), or how defender-side strategies (e.g., differential privacy or input randomization) could mitigate its efficacy.

3. No significant discussion of defense strategies: The paper states findings are for diagnostic/privacy audit purposes, but does not offer suggestions or preliminary results for possible countermeasures or design recommendations for more robust LDMs.

**Questions:**

1. How does MoFit’s performance transfer to other classes of generative models or different LDM architectures? Could the authors provide results (or analysis) for other types of datasets (e.g., medical or highly out-of-distribution images) not covered here?

2. What, in the authors' view, would be the most promising defense for LDMs against MoFit, aside from differential privacy? Are there results for even naïve defense schemes

3. Can the authors provide distributions or examples of hard false positives/negatives at very low FPRs? Are there query images or image properties where MoFit’s advantage is diminished, and if so, what characteristics drive this behavior?

---

> ### Author Response · Authors · 2025-11-22
>
> ## **Overview of All Reviewer Comments**
>
> To thoroughly address the reviewers’ questions and concerns, we conducted a comprehensive set of additional experiments. For clarity and convenience, we summarize all relevant tables below. We kindly invite you to review this summary and refer to the corresponding comments (Comment A to D) for detailed explanations.
> **Note.** T@1F in the table refers to  the TPR@1%FPR metric.
>
> ---
>
> ### **| W1Q1. Evaluation to other large-scale public models.**
>
> |  |  | (a) SD v2.1 |  |  | (b) SD v3 |  |  |
> | ----- | ----- | :---: | ----- | ----- | :---: | ----- | ----- |
> | **Condition** | **Method** | ASR | AUC | T@1F | ASR | AUC | T@1F |
> | GT | CLiD | 58.00 | 55.82 | 2.00 | 67.50 | 71.64 | 5.00 |
> | VLM | Loss | 55.50 | 52.47 | 1.00 | 53.00 | 46.27 | 0.00 |
> | VLM | SecMI | 57.50 | 54.37 | 0.00 | 62.50 | 65.06 | 4.00 |
> | VLM | PIA | 59.00 | 57.53 | 0.00 | 55.00 | 50.84 | 0.00 |
> | VLM | PFAMI | 51.50 | 39.63 | 1.00 | 68.50 | 69.81 | 3.00 |
> | VLM | CLiD | 53.50 | 51.90 | 1.00 | 67.50 | 71.59 | 4.00 |
> | $\\phi^\*$ | **Ours** | 61.34 | 58.99 | 0.00 | 70.00 | 73.42 | 2.00 |
>
> Table 1\. Evaluation against two settings of SD v2.1 (LAION-mi train vs. CC3M) and (b) SD v3 (COCO vs. CC3M).
>
> In Table 1, we evaluate MoFit on two large-scale models with architectural differences from Stable Diffusion v1.5 \[1\]: SD v2.1 \[2\], which differs in the text encoder, and SD v3 \[3\], which introduces a new noise predictor. Across both models, ***MoFit consistently outperforms VLM-captioned baselines across different models***, demonstrating robustness across real-world splits and different architecture designs. Please refer to **Comment A** for further details.
>
> ---
>
> ### **| W2W3Q2. Possible defensive strategies**
>
> |  |  | (a) Gaus. Blur |  |  | (b) JPEG |  |  | (c) LoRA |  |  |
> | ----- | ----- | :---: | ----- | ----- | :---: | ----- | ----- | :---: | ----- | ----- |
> | **Condition** | **Method** | ASR | AUC | T@1F | ASR | AUC | T@1F | ASR | AUC | T@1F |
> | GT | CLiD | 89.10 | 92.27 | 70.40 | 85.50 | 89.59 | 61.00 | 59.00 | 52.05 | 1.00 |
> | VLM | Loss | 64.10 | 68.17 | 4.20 | 63.40 | 66.53 | 3.80 | 54.50 | 49.26 | 0.00 |
> | VLM | SecMI | 60.10 | 62.15 | 5.20 | 54.00 | 54.59 | 2.80 | 58.50 | 53.50 | 1.00 |
> | VLM | PIA | 63.10 | 65.28 | 4.40 | 64.90 | 67.18 | 5.00 | 58.50 | 53.50 | 0.00 |
> | VLM | PFAMI | 80.60 | 87.51 | 27.72 | 78.04 | 84.39 | 36.03 | 73.50 | 77.50 | 1.00 |
> | VLM | CLiD | 81.00 | 87.17 | 44.60 | 78.80 | 85.21 | 39.20 | 59.00 | 53.11 | 1.00 |
> | $\\phi^\*$ | **Ours** | 88.70 | 94.92 | 54.20 | 82.80 | 89.75 | 26.00 | 58.50 | 54.35 | 0.00 |
>
> Table 2\. Evaluation against potential defensive strategies: Gaussian blur, JPEG compression, and LoRA.
>
> We thank the reviewer for emphasizing the importance of defender-side strategies. In **Table 2**, we evaluate all methods under two input randomization defenses. ***MoFit exhibits the highest robustness***, outperforming all VLM-conditioned baselines. Furthermore, we observe that ***LoRA may serve as a promising defensive strategy for MoFit***, as its performance substantially degrades when LDMs are trained with LoRA. Please refer to **Comment B** for details.
>
> ---
>
> ### **| W2Q3. Analysis on failure cases**
>
> We appreciate the reviewer for the opportunity to share our analysis of extreme cases in our approach. We examine both color diversity and image complexity across TP, FN, FP, and TN samples. Interestingly, we found that ***samples predicted as members** (TP and FP) **tend to exhibit higher color diversity and greater structural complexity than their hold-out counterparts** (FN and TN, respectively)*. Please refer to **Comment C** for the experiment and its details.
>
> ---
>
> ### **| Notes on Implementation Details and Revisions**
>
> We clarify that the optimization previously referred to as PGD was actually implemented as standard Gradient Descent (GD), with all reported results unaffected. Additionally, we extend the perturbation-level analysis in Section 5.5, further confirming MoFit’s superiority over alternative surrogate types. Please refer to **Comment D** for details.
>
> ---
>
> \[1\] [https://huggingface.co/stable-diffusion-v1-5/stable-diffusion-v1-5](https://huggingface.co/stable-diffusion-v1-5/stable-diffusion-v1-5)
> \[2\] [https://github.com/Stability-AI/stablediffusion](https://github.com/Stability-AI/stablediffusion)
> \[3\] [https://huggingface.co/stabilityai/stable-diffusion-3-medium-diffusers](https://huggingface.co/stabilityai/stable-diffusion-3-medium-diffusers)

---

> ### Author Response · Authors · 2025-11-22
>
> ## **Comment A \- W1Q1: Evaluation on different large-scale LDMs**
>
> We appreciate the reviewer’s insightful suggestion to evaluate MoFit on a broader architecture of large-scale diffusion models and datasets beyond SD v1.5. In response, we extended our experiments to: (1) Stable Diffusion v2.1, which differs in the text encoder with SD v1 (switch from CLIP to OpenCLIP \[1\]) and (2) Stable Diffusion v3, which adopts a different architecture based on Transformers instead of U-Net.
>
> ---
>
> ### **(1) Evaluation on Stable Diffusion v2.1 (SD v2.1)**
>
> **Defining Member/Hold-out Splits for SD v2.1.** Unlike SD v1.5, which allows controlled evaluation via the curated LAION-mi split \[2\], evaluating membership inference on SD v2.1 \[3\] presents a unique challenge. As detailed in \[3\], SD v2.1 is trained on LAION-5B, a superset that contains both the member and hold-out images of LAION-mi. Thus, LAION-mi cannot be reused directly for testing generalization performance on SD v2.1.
>
> | LAION-mi member vs. | COCO | Flickr | CC3M | LAION-mi hold-out |
> | :---: | :---: | :---: | :---: | :---: |
> | FID | 53.6041 | 61.8270 | 16.0582 | 8.8673 |
>
> Table 3\. FID scores between LAION‑mi member set and different datasets.
>
> To address this challenge, we construct a distributionally aligned evaluation split, following best practices in \[2\] that recommend closely matched distributions for member and hold-out samples. We compute FID scores between the LAION-mi member set and candidate datasets (COCO, Flickr, CC3M), and find that CC3M yields the lowest FID (16.06), closely matching the intra‑LAION-mi FID (8.87) (see Table 3). Accordingly, we use LAION‑mi members as the member set and CC3M as the hold-out set for SD v2.1 evaluation.
>
> **Evaluation on SD v2.1 using real-world split.** We perform membership inference attacks using 100 images from each split and evaluate MoFit alongside all baselines. In Table 1(a), under VLM‑generated captions – a realistic setting where the adversary lacks GT annotations – most baselines exhibit notable drops in ASR and AUC. In contrast, ***MoFit consistently outperforms them, achieving a \+2.34% ASR gain over the second-best method*** under these restricted conditions.
>
> ---
>
> ### **(2) Evaluation on Stable Diffusion v3 (SD v3)**
>
> **Defining Member/Hold-out Splits for SD v3.** We further evaluated MoFit on Stable Diffusion v3 (SD v3) to test its generalization to a different model architecture – specifically, from U-Net-based LDMs (*e.g.*, SD v1.4, v1.5, v2.1) to the Multimodal Diffusion Transformer (MMDiT) \[5\]. However, SD v3 provides no publicly available details about its training dataset aside from the number of training images \[6\], making it challenging to construct reliable member/hold-out splits.
>
> Empirically, we observed that COCO samples yield significantly lower conditional and unconditional loss values (Eq. 1 and 2 in the main paper) compared to LAION-mi or CC3M. Based on this, we selected COCO as the member set and CC3M as the hold-out set for evaluating MoFit on SD v3. (We will include a supporting figure visualizing the loss gap across datasets in the Appendix.)
>
> **Evaluation on Pre-trained SD v3.**  We conducted membership inference using 100 images from each split, running all methods at a fixed diffusion timestep (*i.e.*, 140-th timestep of the default schedule). As shown in Table 1(b), ***MoFit outperforms all baselines, including CLiD under ground-truth conditions.*** These results further confirm that MoFit can match or even exceed methods that rely on full image-caption supervision, consistent with our findings on SD v1.4 (see Table 2 of the main paper for MS-COCO).
>
> ---
>
> \[1\] Cherti, Mehdi, et al. "Reproducible scaling laws for contrastive language-image learning." Proceedings of the IEEE/CVF conference on computer vision and pattern recognition. 2023\.
> \[2\] Dubiński, Jan, et al. "Towards more realistic membership inference attacks on large diffusion models." Proceedings of the IEEE/CVF Winter Conference on Applications of Computer Vision. 2024\.
> \[3\] [https://github.com/Stability-AI/stablediffusion](https://github.com/Stability-AI/stablediffusion)
> \[5\] Esser, Patrick, et al. "Scaling rectified flow transformers for high-resolution image synthesis." Forty-first international conference on machine learning. 2024\.
> \[6\] [https://huggingface.co/stabilityai/stable-diffusion-3-medium-diffusers](https://huggingface.co/stabilityai/stable-diffusion-3-medium-diffusers)

---

> ### Author Response · Authors · 2025-11-22
>
> ## **Comment B \- W2 & W3Q2: Defensive strategies**
>
> We appreciate the reviewer for emphasizing the need to consider defender‑side strategies. (1) We extended our evaluation to **input‑randomization defenses** and analyze their effects on MoFit. (2) We also found that **Low‑Rank Adaptation (LoRA) with a limited number of parameters can act as an effective countermeasure**. We agree that the original paper lacked sufficient discussion on defenses and will incorporate below discussions into the revised paper.
>
> ---
>
> ### **(1) Input Randomization**
>
> We evaluate input randomization as a defense method. Following the standard SD v1.4 fine‑tuning protocol on MS‑COCO, we additionally apply: (a) **Gaussian blur** (3×3 kernel, σ ∈ \[0.1, 2.0\]) and (b) **JPEG compression** (quality \= 60).
>
> We fine‑tune separate LDMs under each augmentation and directly evaluate MoFit using embeddings crafted from the non‑augmented base model – reflecting a challenging setting where our model-fitted embedding is constructed without awareness of the input-space changes introduced during fine-tuning.
>
> Using 500 member and 500 hold‑out images, Table 2 (a) and (b) show that all methods experience comparable or slightly degraded performance under both augmentations. Importantly, two trends remain consistent:
> - (i) All baselines degrade significantly when shifting from GT to VLM‑generated captions
> - (ii) MoFit consistently outperforms all baselines under VLM‑generated captions across both ASR and AUC – ***even when input randomization is applied, achieving ASRs over 82.8%**.*
>
> These results suggest that the alignment between the model-fitted surrogate and its embedding remains strong, despite input perturbations during training – leading to significant misalignment at the inference-time that MoFit effectively leverages.
>
> ---
>
> ### **(2) Potential defensive strategy: LoRA**
>
> To address the reviewer’s question regarding promising defenses against MoFit, we find that ***Low‑Rank Adaptation (LoRA) substantially reduces the effectiveness**.* Specifically, replacing full U‑Net fine‑tuning with LoRA leads to a noticeable performance drop for our method.
>
> **Studies related to robustness of LoRA against MIA.** Recent studies in language modeling have evaluated membership inference under LoRA fine-tuning. While early works \[1\] suggest robustness, later studies \[2,3\] show that LoRA’s limited parameter updates significantly reduce attack effectiveness. For latent diffusion models, Luo et al. \[4\] suggest that LoRA-adapted LDMs may remain vulnerable, but their evaluation is limited to earlier MIA methods \[5,6\] (*e.g.*, the “Loss” method in our paper), and does not cover more recent approaches.
>
> **Our experiment against LoRA.** In contrast, Table 2(c) shows that when evaluated on 100 samples of Pokemon training set used to fine-tune the LoRA-adapted SDv1.4 \[7\], ***performance drops to near-random levels for both MoFit and most of the baselines under ground-truth and VLM-generated captions***. We hypothesize that this robustness stems from LoRA’s minimal footprint, which preserves most of the original weights and reduces the model’s capacity to remember specific training samples \[2\].
>
> **Reason behind the robustness of PFAMI \[8\].** Baselines other than PFAMI rely on *cross-query* comparisons – typically based on the relative ranking of diffusion losses across different queries. The poor performance of these baselines under LoRA, as shown in Table 2(c), suggests that the loss values of member and non-member queries become increasingly similar. This makes it difficult for cross-query methods to distinguish between them based on relative loss ordering.
>
> However, PFAMI uses *within-query* scores by applying multiple augmentations (*i.e.*, crop) to a single query image and computing the loss for each. The membership score is based on the relative variation within these losses. Since it does not rely on comparisons across different queries, PFAMI remains robust even when LoRA causes the overall loss values of member and non-member samples to become similar. The relative differences within a single query are preserved, making it less sensitive to such global shifts.
>
> ---
>
> \[1\] Wen et al., “Last one standing: A comparative analysis of security and privacy of soft prompt tuning, lora, and in-context learning”, arXiv 2023\.
> \[2\] Amit et al., “SoK: reducing the vulnerability of fine-tuned language models to membership inference attacks”, arXiv 2024\.
> \[3\] Liu et al., “Precurious: How innocent pre-trained language models turn into privacy traps”, arxiv 2024\.
> \[4\] Luo et al., "Privacy-Preserving Low-Rank Adaptation Against Membership Inference Attacks for Latent Diffusion Models", AAAI 2025\.
> \[5\] Hu et al., "Membership inference of diffusion models", arXiv 2023\.
> \[6\] Matsumoto et al., "Membership inference attacks against diffusion models", SPW 2023\.
> \[7\] https://huggingface.co/sr5434/sd-pokemon-model-lora

---

> ### Author Response · Authors · 2025-11-22
>
> ## **Comment C – W2Q3: Analysis on extreme cases**
>
>
> | Dataset | MS-COCO | MS-COCO | MS-COCO | MS-COCO | Flickr | Flickr | Flickr | Flickr |
> | ----- | :---: | :---: | :---: | :---: | :---: | :---: | :---: | :---: |
> |  | TP | FN | FP | TN | TP | FN | FP | TN |
> | Colorfulness Score | 135.27 | 122.65 | 147.64 | 132.54 | 146.22 | 138.56 | 147.51 | 147.95 |
> | Entropy | 15.22 | 13.70 | 15.11 | 14.27 | 15.55 | 15.05 | 14.92 | 15.33 |
> | Keypoint Count | 500.00 | 471.95 | 499.25 | 492.20 | 500.00 | 494.55 | 499.65 | 497.75 |
>
> Table 4\. TP, FN, FP, and TN statistics using color diversity and image complexity metrics for two fine-tuned SD 1.4 models.
>
>
> We are grateful for the reviewer’s suggestion to investigate extreme cases, which we address through the following analysis. Using the two fine-tuned SD 1.4 models (MS-COCO and Flickr), we selected the top-20 samples from each of the following groups:
>
> - **TP** (True Positives): members predicted as members
> - **FN** (False Negatives): members predicted as hold-outs
> - **FP** (False Positives): hold-outs predicted as members
> - **TN** (True Negatives): hold-outs predicted as hold-outs
>
> To quantify *color diversity*, we used the colorfulness score \[1\], and to assess *image complexity*, we employed Shannon entropy \[2\] and ORB keypoint count \[3\]. Higher colorfulness indicates richer chromatic variation, while higher entropy and keypoint counts reflect greater structural complexity.
>
> Interestingly, as shown in Table 4, we observed that samples predicted as members (TP and FP) consistently exhibit higher values across all metrics compared to their hold-out counterparts (FN and TN). This suggests that the ***model is more likely to associate colorful and structurally rich images with training membership***. We believe this finding may offer an interesting direction for future work on understanding what types of images diffusion models are more likely to retain.
>
> ---
>
> \[1\] Hasler, David, and Sabine E. Suesstrunk. "Measuring colorfulness in natural images." Human vision and electronic imaging VIII. Vol. 5007\. SPIE, 2003\.
> \[2\] Shannon, Claude E. "A mathematical theory of communication." The Bell system technical journal 27.3 (1948): 379-423.
> \[3\] Rublee, Ethan, et al. "ORB: An efficient alternative to SIFT or SURF." 2011 International conference on computer vision. Ieee, 2011\.

---

> ### Author Response · Authors · 2025-11-22
>
> ## **Comment D: Clarification and Implementation Adjustment**
>
> ### **(1) Section 4.1 & 5**
>
> We would like to clarify that the optimization referred to as PGD in Section 4.1 was implemented as standard Gradient Descent (GD) without the projection (clipping) step. While we initially described the optimization as PGD, we acknowledge this mismatch in terminology and will revise the manuscript to reflect the exact implementation.
>
> Importantly, the numerical results throughout the paper remain unchanged. In practice, all reported experiments were conducted using GD with an initial noise level $\\varepsilon$ equivalent to the previously stated noise bound ($\\eta \= 0.3$). The step size $\\alpha$ is initialized to 0.15 and decays proportionally with the iteration count. Please refer to Eq. (10) in Appendix A.1 for the update equation.
>
> ---
>
> ### **(2) Section 5.5**
>
> In Section 5.5, we extend our evaluation of configuration (ii), where a random noise perturbation $\\delta$ is added to the query image. While the main paper reports results at a single perturbation level ($\\varepsilon \= 0.3$), we now investigate a broader range of noise levels, varying $\\varepsilon$ from 0.1 to 0.9. For each dataset, we identify the noise level that yields the strongest defensive effect and report the corresponding results in the second row of Table 4\.
> Notably, MoFit continues to exhibit strong and stable performance across all datasets, even under the best-performing perturbation level for each dataset. This demonstrates that our model-fitted surrogate $x^\*$ establishes a more robust and discriminative coupling with its optimized embedding than perturbed inputs such as $x \+ \\delta$ and other variants (*e.g.*, $x$ or $x \+ \\delta\_{\\text{max}}$).
>
> ---
>
> ### **(3) Figure 5**
>
> As our optimization uses GD without projection, we replace the noise bound (η) with the initial step size, which decays proportionally with iteration. Note that the experimental results remain unchanged; only the x-axis label is updated to reflect the corrected interpretation.

---

> ### Author Response · Authors · 2025-11-25
>
> ## **Additional Comment (1) \- W1Q1: Medical Dataset**
>
> |  |  | ASR | AUC | TPR@1%FPR |
> | ----- | ----- | :---: | :---: | :---: |
> | GT | CLiD | 60.50 | 60.30 | 0.00 |
> | VLM | Loss | 54.00 | 47.27 | 0.00 |
> | VLM | SecMI | 53.00 | 46.20 | 0.00 |
> | VLM | PIA | 51.50 | 44.25 | 0.00 |
> | VLM | PFAMI | 53.00 | 50.22 | 1.00 |
> | VLM | CLiD | 55.00 | 49.78 | 1.00 |
> | Emb | **Ours** | 57.00 | 54.44 | 2.00 |
>
> Table 5. Evaluation against Prompt2MedImage\[1\], SD v1.4 fine-tuned on ROCO dataset.
>
> As suggested by the reviewer, we additionally evaluated MoFit on a domain-specific Latent Diffusion Model (LDM) trained on medical data. Specifically, we used Prompt2MedImage \[1\], a Stable Diffusion v1.4 model fine-tuned on the ROCO dataset \[2\], which contains radiology image–caption pairs. The training and validation splits were used as member and hold-out sets, respectively. Additionally, because general‑purpose VLMs cannot reliably caption domain‑specific medical images, we employ a BLIP model fine‑tuned on the ROCO dataset \[3\] to generate the required captions in the caption-free setting.
>
> As shown in Table 5, ***MoFit consistently outperforms all VLM-based baselines across all metrics***, successfully distinguishing member samples from hold-outs even in this specialized medical domain. These results confirm that ***MoFit generalizes across domains***, highlighting its effectiveness on in-the-wild LDMs beyond general-purpose text-to-image settings.
>
> \[1\] [https://huggingface.co/Nihirc/Prompt2MedImage](https://huggingface.co/Nihirc/Prompt2MedImage)
> \[2\] Pelka, Obioma & Koitka, Sven & Rückert, Johannes & Nensa, Felix & Friedrich, Christoph. (2018). Radiology Objects in COntext (ROCO): A Multimodal Image Dataset: 7th Joint International Workshop, CVII-STENT 2018 and Third International Workshop, LABELS 2018, Held in Conjunction with MICCAI 2018, Granada, Spain, September 16, 2018, Proceedings. 10.1007/978-3-030-01364-6\_20.
> \[3\] [https://huggingface.co/Siddartha01/blip-medical-captioning-roco](https://huggingface.co/Siddartha01/blip-medical-captioning-roco)

---

> ### Author Response · Authors · 2025-11-27
>
> ### **Gentle Reminder and Follow-Up**
>
> We appreciate your time and effort for reviewing our work again. We have posted clarifications and additional evaluations results to address your raised concerns. It would be grateful to let us know if there are more questions and concerns that we can address.
>
> For your convenience, we summarize the main points and indicate where they are addressed in the revised manuscript below, marked in blue for clarity:
>
> - Input Randomization – Section 5.6.2
> - LoRA – Section 5.6.2, Appendix A.7
> - SD v1.5, SD v2, SD v3 – Appendix A.9
> - Failure cases – Appendix A.13
> - Medical dataset – Appendix A.10

---

### Author Response · Authors · 2025-12-03

Dear Area Chair,

Thank you for your time and effort during the review process. Below, we provide a brief summary of our manuscript and how we addressed the reviewers’ concerns and questions.

---

### **1\. Motivation**

We propose **MoFit**, a novel membership inference method for text-to-image diffusion models that operates without access to training captions – a challenging and underexplored setting in prior works.

**What is membership inference attack (MIA)?**

- A membership inference attack (MIA) is designed to decide whether a given query image is used to train the target model.

**Why *caption-free* setting?**

- Prior works assume access to ground-truth (GT) captions for each query image.
- However, this is unrealistic in practice – *e.g.*, artists rarely know the original captions used in training.
- VLM-generated captions have been proposed as an alternative, but suffer significant performance drop.
- We propose the first method that effectively addresses membership inference in the caption-free setting (`exJi`, `awTs`, and `8oRv` highlighted our novelty; `edeD` found the motivation intuitive and easy to follow).

---

### **2\. Contributions**

- Our analysis reveals a novel insight: *training samples* exhibit relatively higher sensitivity to condition changes (*e.g.*, replacing GT captions with alternatives) than *test samples* – offering a strong signal for membership inference in the caption-free setting (Sec. 3.3) (recognized by `awTs`, `edeD`).
- We propose **MoFit** which further amplifies this sensitivity difference by constructing a **mo**del-**fit**ted embedding tailored to each query image (Sec. 4).
- MoFit consistently outperforms prior VLM-captioned baselines and even achieves performance comparable to GT-captioned methods across diverse Stable Diffusion variants (v1–v3) and datasets (Sec. 5\) (recognized by `awTs`, `8oRv`, `edeD`).

---

### **3\. Discussions**

During the discussion period, we addressed all reviewer comments. Below, we summarize their feedback, our responses, and the corresponding revisions in the manuscript.

| Comments | `exJi` | `awTs` | `8oRv` | `edeD` | Manuscript | Our Responses |
| :---- | ----- | ----- | ----- | ----- | --- | ----- |
| Applying to SD v2.1, v3 | ✔ |  | ✔ | ✔ | A.9 | A |
| Recent method: ReDiffuse |  | ✔ | ✔ | ✔ | A.15 | B |
| Runtime & Early Stopping |  | ✔ | ✔ | ✔ | Sec. 5.6.3, A.8 | C |
| Potential Defense | ✔ | ✔ |  | ✔ | Sec. 5.6.2, A.7 | D |
| Analysis on FP, FN | ✔ |  |  | ✔ | A.13 | E |
| Medical dataset | ✔ |  |  |  | A.10 | F |
| Recent VLMs |  | ✔ |  |  | A.11 | G |

| Responses | Location | Summary |
| ---- | ---- | ---- |
| A | Tab.1 (`exJi`) | To address questions about MoFit’s robustness against different model architectures, we evaluated MoFit on **SD v2.1** and **v3**, which differ from v1 in their text encoder and denoising model, respectively. MoFit consistently outperforms VLM-conditioned baselines, demonstrating strong resilience to major architectural differences. |
| B | Tab. 2 (`awTs`) | In response to the mention of **ReDiffuse**, a recent black-box MIA method, we note that it still relies on VLM captions in our caption-free setting. MoFit outperforms it under VLM conditions, with ≥32.7% gain in ASR. |
| C | Tab.4,5 (`awTs`) | In response to comments regarding computational efficiency, we propose an **early stopping** strategy that reduces per-image runtime from 358.96 s to 22.32 s, while still outperforming the best-performing baseline. We also report **alternative optimizers** that significantly lower both memory usage and runtime. |
| D | Tab.2 (`exJi`) | To address reviewers’ request on potential defenses, we identify **LoRA** as a promising defense against both MoFit and prior MIA methods (`edeD` expressed interest). We further evaluate MoFit on SD models fine-tuned with JPEG- and blur-augmented samples, where it consistently outperforms baselines under VLM captions despite **input randomization**. |
| E | Tab.4 (`exJi`) | While **analyzing failure cases** (FN and FP) as reviewers requested, we observed that SD models tend to retain colorful and structurally rich images more than plain and monotonous ones. As `edeD` requested, we include visualizations of these examples in Fig. 13\. |
| F | Tab.5 (`exJi`) | To address feedback on domain transferability, we evaluated MoFit against SD fine-tuned with **radiology images**.  MoFit outperforms baselines under VLM captions, showing its generalizability beyond natural images. |
| G | Tab.2 (`awTs`) | To address inquiries on MoFit’s performance with **recent VLMs**, we evaluate using MoonDream2 and Molmo. Despite richer captions, MoFit still outperforms baselines by ≥10.5% in ASR. |

We also address reviewers’ questions on MoFit’s robustness to hyperparameter choices (`8oRv`) and performance in low-resource scenarios (`awTs`). Please see Tab. 3 of `8oRv` (or A.12) and Tab. 1 of `awTs` (or A.14) for the corresponding experiments and responses, respectively.

---

### Meta-Review · Area_Chair_ruqH · 2026-01-07

**Summary:**

The AC carefully read the paper and the full discussion. The submission received mixed initial reviews (scores: 6, 4, 4, 6). Reviewers generally acknowledged that the work addresses a meaningful “caption-free” gap, aligns well with real-world settings where training annotations may be unavailable or undisclosed, and is supported by extensive experiments and practical motivation. However, they also raised concerns about performance on other large-scale diffusion models, the long runtime, and omissions in related work coverage. In the rebuttal, the authors responded to the major issues from the initial review stage, and the updated aggregated scores now lean toward acceptance. Accordingly, I am inclined to recommend acceptance.

**Reviewer Concerns:**

The two most significant concerns—(1) the long runtime and computational cost raised by Reviewers awTs, 8oR, and vedeD, and (2) the lack of results on other large-scale image generation models raised by Reviewers exJi, 8oRv, and edeD—have been well addressed.

In addition, several secondary issues have also been addressed, including missing related work (awTs), potential defense concerns (exJi), domain transferability (exJi), and the absence of results with more recent VLMs (awTs).

**Reviewer Scores:**

Reviewer exJi  a is likely to keep the positive score (6), since the concerns about MoFit’s generalization, potential defenses beyond differential privacy, and low-FPR failure cases have been addressed.

Reviewer awTs is likely to raise the score (6), since the requested improvements on efficiency, robustness/baselines, and risk/defense analysis have been addressed.

Reviewer 8oRv is likely to raise the score(6), since concerns about practicality (black-box settings), generalization to large-scale pretrained models, computational overhead, and stability/robustness have been addressed.


Reviewer edeD will keep the postive score (6).

---

### Decision · Program_Chairs · 2026-01-26

Accept (Poster)